# Brainbuilder: a software pipeline for 3D reconstruction of cortical maps from multi-modal 2D data sets
Thomas Funck[1,2], Konrad Wagstyl[3], Claude Lepage[4], Mona Omidyeganeh[4], Paule-Joanne Toussaint [4], Ting Xu [1], Katrin Amunts [2,5], Alexander Thiel[6,7], Nicola Palomero-Gallagher [2,5] & Alan C. Evans [4]

Mesoscale maps of brain architecture are important tools for characterizing the chemo- and cytoarchitectonic organization of the brain. These maps are essential for advancing our understanding of normal and pathologic brain function because they provide a bridge between neuron-level micro-scale imagining and macro-level population brain imaging. Here we introduce a method and software package called BrainBuilder for reconstructing 3-dimensional cortical maps from data sets of 2-dimensional post-mortem serial brain sections processed for the visualization of multiple different biological features. This pipeline can be applied to the brains from different species, without the strict need for a corresponding reference volume from the brain donor. As a proof of principle, we reconstruct data showing the distribution of multiple neurotransmitter receptor binding sites, and cell and myelin stained sections in the human and macaque brain. We show that BrainBuilder can serve as the basis for the development of future mesoscale 3D atlases.

Post-mortem brain imaging of 2-dimensional (2D) brain sections, using modalities such as autoradiography and histology, enables the resolution of features at the scale of "mesoscale" brain structures such as architectonic cortical layers, cortical columns, and sub-nuclei. This approach allows the measurement of biological features that may remain inaccessible through in vivo imaging approaches, such as RNA transcriptomics. Mesoscale atlases[1–5] play a pivotal role in elucidating the intricate architecture of the brain, shedding light on the fine-grained details of its structure that are essential for advancing our understanding of brain function and pathology.

Imaging the complex anatomy of the brain using 2D sections has inherent limitations that can only be overcome by restoring the original embedding of these sections through 3D reconstruction. Notably, the laminar structure of the cortex is poorly represented when the section's cutting angle is not perpendicular to the curvature of the cortical surface. It is also difficult to use 2D sections with other brain imaging data sets, e.g., stereotaxic 3D atlases, because it is challenging to map the 2D sections to other data. Accurate 3D reconstruction is therefore essential to creating 3D mesoscale brain atlases.

Existing mesoscale atlases have focused on mapping cell density or myelination. Though macroscopic atlases capable of resolving cortical areas on the scale ≥2.4 mm have been created for neurotransmitter receptor distributions using in vivo positron emission tomography[6,7], mesoscale atlases of neurotransmitter receptor distributions are lacking. These macroscopic receptor atlases suffer from partial-volume effects that limit their quantitative accuracy and cannot resolve laminar receptor distribution patterns. Given that cortical layers differ in their connectivity patterns and neurotransmitter receptors underpin all information processing in the brain, both their regional and laminar distribution patterns are key to understanding how mesoscale brain structure links to brain function.

The creation of mesoscale neurotransmitter receptor atlases is particularly challenging because of the obstacles involved in reconstructing autoradiographs, i.e., images of sections processed by in vitro receptor autoradiography, the gold standard method for imaging neurotransmitter receptors. We have created a reconstruction pipeline, BrainBuilder, that, while initially designed to reconstruct mesoscale neurotransmitter receptor atlases, in fact generalizes to virtually any post-mortem 2D sections where the cortical gray matter (GM) can be visualized. This flexibility is thanks in part to a deep learning network that is trained exclusively on synthetic data to segment the cortical GM. The numerous individual challenges involved in the reconstruction of 2D autoradiography are, respectively, frequently

[1]Child Mind Institute, New York, NY, USA. [2]Institute of Neuroscience and Medicine (INM-1), Research Centre Jülich, Jülich, Germany. [3]Wellcome Center for Human Neuroimaging, University College London, London, UK. [4]Montreal Neurological Institute, McGill University, Montreal, QC, Canada. [5]C. and O. Vogt Institute of Brain Research, Medical Faculty, University Hospital Düsseldorf, Heinrich-Heine-University Düsseldorf, Düsseldorf, Germany. [6]Lady Davis Institute, Jewish General Hospital, McGill University, Montreal, QC, Canada. [7]Department of Neurology and Neurosurgery, McGill University, Montreal, QC, Canada. ✉e-mail: n.palomero-gallagher@fz-juelich.de

encountered in other post-mortem 2D imaging data sets. These include sparse sampling of sections, severe non-linear deformations of the brain prior to sectioning, and multi-modal serial data acquisition. Thus, by solving these problems for the particular application of 2D autoradiography, we have created a flexible, robust reconstruction pipeline that we hope will help generate mesoscale atlases from numerous other modalities.

Several challenges must be overcome before 2D brain sections can be reconstructed into 3D brain atlases that can be used for automated quantitative analyses across the acquired sections and in conjunction with existing 3D brain templates and annotations. These include sparse sampling of the sections, variable pixel intensities from different acquisitions, and non-linear 2D and 3D deformations in the brain sections. In the following, we describe some of the most frequent challenges that our pipeline aims to overcome.

Before proceeding, we will briefly clarify some terminology. A binding site refers to the molecular target to which a ligand binds. Different ligands can bind to the same receptor, e.g., $GABA_A$, but on different binding sites, e.g., benzodiazepine versus GABA binding sites, and hence map different spatial distributions. Acquisition is defined as a within-modality method for measuring a given biological feature, e.g., autoradiographs acquired with different ligands measure different receptor binding sites and hence are referred to as different acquisitions. "Section" refers generically to the 2D image produced by slicing the brain across a given plane and using an imaging modality to visualize some biological feature in the sliced tissue, such as cell density or receptor binding site.

A major obstacle to 3D reconstruction is 3D deformations of the donor brain prior to sectioning. These include shrinkage from chemical fixation or deformations occurring during immersion fixation. Deformations are especially severe when the acquisition method, e.g., in vitro autoradiography, requires the use of fresh tissue and where chemical fixation cannot be used (Fig. 1A). The result is that though the coronal sections may appear as 2D coronal images, they in fact come from a more complex 3D spatial embedding in the brain which must be recovered during reconstruction.

Sections may also be acquired from slabs of brain tissue that have not been cut along a single parallel axis (Fig. 1B). This results in 2D sections with varying cutting angles and which are therefore not in the same plane. The sections acquired across these tissue slabs cannot be naively concatenated into a single stack that could then be reconstructed into 3D. Instead, the simplest approach is to first reconstruct the sections to the slabs of tissue from which they were cut and only then can sections from each slab be combined into a single 3D reconstruction.

Another important obstacle to reconstruction is the use of multiple different types of image acquisitions, e.g., different radioligands or histological staining methods, from different sections within the same brain. While the use of several methods of acquisition is highly desirable to measure multiple biological features from within the same subject, it results in images with heterogeneous pixel intensity distributions (Fig. 1C). These sections may be difficult to align with one another and will produce 3D volumes with extremely varied pixel intensity distributions when concatenated together.

The use of serial acquisition of multiple types of biological features within the same brain (e.g., receptor density, cell body and white matter (WM) staining) implies that there will be gaps between acquired sections of a given modality. This problem is compounded by the occasional loss of sections due to mechanical processing or by poor image acquisition. The sparse sampling of sections necessitates a method for estimating the distribution of pixel intensities in missing sections for a particular type of section.

To illustrate the sparsity of the data used here, only ~37% of the donor brain was sampled with complete sections, with an average distance of 1.03 mm and a maximum distance of 1.42 mm between sections measuring the same biological feature. Sections were either lost due to mechanical processing errors or excluded because the quality of the acquisition was poor (e.g., binding artefacts or blurred histological staining) or because the sections contained only a fragment of brain tissue, frequently as a result of

slicing the sections close to the border of the tissue slabs. In the macaque, the sections covered ~24% of the brain with a sampling of 1.32 mm between acquired sections for the same biological feature. The macaque brain was deliberately sampled sparsely in repeats of sections along the coronal axis to allow sampling through the brain while limiting the cost of data acquisition.

The challenge posed by missing sections is further aggravated because the cutting angle of the section is virtually impossible to be completely perpendicular to the curvature of the pial surface across the entire cortical ribbon. As a consequence, the laminar distributions of certain biological features, such as cell body or receptor density, may be misrepresented across the cortical depth. Hence, an interpolation method is required for estimating missing sections that will not propagate artefacts resulting from the cutting angle of the sections.

Methods have been developed to attempt to account for each of the above mentioned challenges individually, including the use of fiducial markers in the brain prior to sectioning, block-face imaging, and the use of structural reference volumes from the donor, e.g., typically a T1 weighted (T1w) MRI. Furthermore, many reconstruction algorithms that leverage these techniques to perform 3D reconstruction have been developed (see Dubois[8] and Pichat et al.[9] for reviews). Although these specialized pipelines are able to solve many of the challenges presented above, none of them can address and solve the *combination of all* these challenges. E.g., semi-automated 2D reconstruction methods have been proposed using fiducial markers implanted in the brain prior to sectioning[10–12] and using block-face imaging[13–15]. Another semi-automated approach to 2D image reconstruction was to manually identify anatomic landmarks on adjacent sections[16,17]. Automated reconstruction can be performed with only the 2D sections themselves using principal-axes transforms[18], intensity or frequency-based cross-correlation[19,20], sum of squared error[21], discrepancy matching optical flow[21–23], or edge-based point matching[21–23]. Methods have also been developed to perform more robust alignment between sections and to maximize the smoothness of the 3D reconstruction[24–27]. A recent and particularly innovative approach to reconstruction involved using Bayesian estimation to simultaneously align the histological sections and corresponding MRI sections while also transforming the pixel intensities of the former to resemble the latter[28,29].

Finally, an iterative strategy for reconstructing 2D *unimodal* sections using an accompanying structural brain image was proposed by Malandain et al.[30] and later adapted by Yang[31] and Amunts et al.[1]. This scheme uses densely sampled sections for a single kind of histological acquisition, e.g., cell-density, and iterates between two steps at progressively higher spatial resolutions. First, the reference structural brain image is aligned in 3D to a stack of 2D sections, and then the 2D sections are linearly aligned in 2D to the structural brain image. By beginning at a coarse spatial resolution and progressively refining the resolution, these pipelines converge to an accurate alignment between the 2D sections and the 3D structural reference volume.

Summarizing, despite the existence of many reconstruction pipelines, none of them is designed to account for all of the challenges described above, and which are associated with the 3D reconstruction of 2D autoradiographs coding for the distribution patterns of 20 different neurotransmitter binding sites. Hence, the need for the development of a pipeline that is capable of working with this highly challenging multimodal data set.

Our goal is to create an automated 3D reconstruction pipeline that makes minimal assumptions about the 2D post-mortem sections, such that it can be used to create mesoscale brain atlases from a wide variety of 2D acquisitions. In particular, our pipeline works: 1) with multimodal data sets imaged with multiple serial acquisitions such as multi-receptor autoradiographs, 2) with sparsely sampled sections, 3) when sections are acquired from slabs of tissue within the whole brain, 4) without the strict requirement of a corresponding structural reference volume from the brain donor and 5) across multiple species, specifically humans and macaques. To facilitate the reconstruction of 2D images with heterogeneous pixel intensity distributions, we implemented a 2D U-Net model to segment the cortex. A recent tool for modality-agnostic 3D MRI segmentation was created through training a *U*-Net deep learning architecture using synthetic data

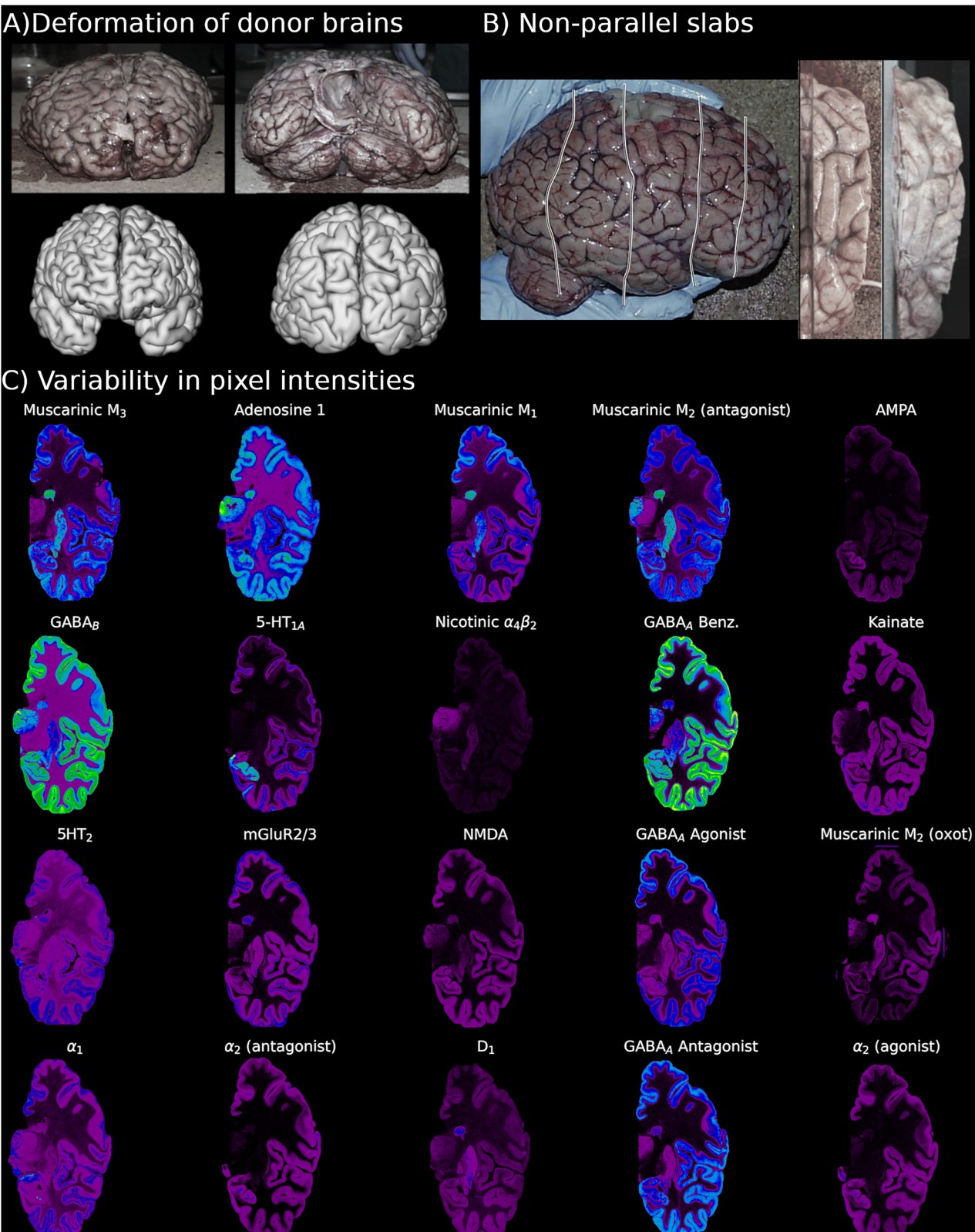

**Fig. 1 | Challenges of reconstructing multi-modal data from unfixed brains. A** Anterior and posterior view of donor brains. The brains were not chemically fixed and hence exhibited severe deformations when compared to Freesurfer8 cortical surface representations derived from the donor MRI. **B** Tissue slabs (~1.7–2.9 cm in thickness) from the human brains were not cut perfectly parallel to one another (see the lines for an example of cuts) and, as illustrated in the depicted slab, each slab of fresh tissue was subject to its own deformation prior to shock freezing. Sections sliced from ends of slabs frequently contained small amounts of brain tissue when sliced by the flat interface of the microtome and were lost or discarded, creating gaps between the slabs. **C** Exemplary autoradiographs from the human data set showing each of the 20 neurotransmitter binding sites, and illustrating the substantial heterogeneity in pixel intensities between the autoradiographs.

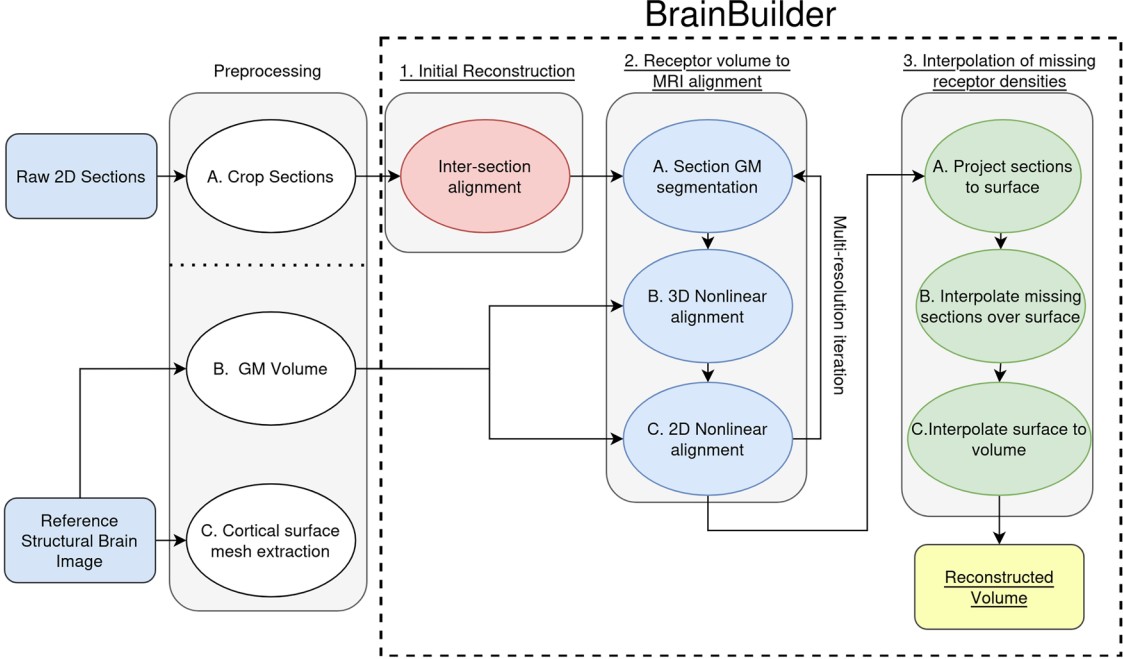

**Fig. 2 | Overview of BrainBuilder.** The pipeline contains 3 major processing stages: 1) inter-section 2D alignment, 2) iterative multi-resolution 3D volumetric registration followed by 2D section-wise alignment of section to the reference structural brain image, 3) surface-based interpolation of receptor binding densities.

with randomly generated tissue contrasts[32]. Here, expand on this approach and develop a neural network trained for 2D images based on synthetic data.

We here apply it to two autoradiography data sets collected by the Zilles labs at Heinrich-Heine-University Düsseldorf and the Research Center Jülich over a period of more than two decades: one covering the entire human brain, the other that of the macaque monkey (for overviews, see, Zilles et al.[33]; Palomero-Gallagher and Zilles[33,34]; Palomero-Gallagher et al.[35]). This is a unique resource because it samples the distribution patterns of different receptors for the classical neurotransmitters glutamate, GABA, acetylcholine, noradrenaline, serotonin and dopamine and for the neuromodulator adenosine (20 binding sites in the human and 14 in the macaque brain), as well as cell bodies and of myelinated fibers. The data are acquired for both species and at high spatial resolution: ~50 μm in-plane, ~20 μm through-plane full-width at half maximum (FWHM) resolution. The challenges present in these two data sets have prevented them from being reconstructed until now.

Specifically, as proof-of-principle we have demonstrated the application of BrainBuilder on the human brain for the entire receptor autoradiographic dataset of 20 neurotransmitter receptors from one hemisphere and on the macaque brain for a selection of 14 receptors for the classical neurotransmitter systems from one hemisphere. Sections stained for cell body and myelin density were also reconstructed from the macaque data. Furthermore, a reference structural MRI dataset was available for the human but not for the macaque donor brain. We show that BrainBuilder can be used on a wide variety of 2D images and can therefore be used as the basis for the creation of 3D mesoscale atlases of the brain across multiple species and acquisition modalities.

## Results
### Pipeline overview
We created a pipeline enabling the 3D reconstruction of 2D sections coding for the distribution of 20 different neurotransmitter binding sites in the human brain, and for which a volumetric reference was available. We also used the pipeline for the 3D reconstruction of histologically processed sections, and implemented the option to use a stereotaxic template brain, and not a structural volume of the donor brain, as the reference structural template. As a proof of concept, we reconstructed a dataset from the macaque monkey brain encompassing sections visualizing 14 receptor types

as well as cell bodies and myelinated fibers, and for which no structural template was available. The pipeline consists of three major processing stages: 1) an inter-section 2D alignment, 2) an iterative multi-resolution 3D volumetric registration followed by 2D section-wise alignment of section to the reference structural brain image, 3) and a surface-based interpolation of receptor binding densities.

Briefly, BrainBuilder (Fig. 2) is composed of three major processing stages:

1) An initial volume is created by rigid 2D inter-section alignment of acquired sections (Fig. 2.1).
2) An iterative multi-resolution alignment scheme that alternates between 3D volumetric followed by 2D section-wise alignment of the sections to the reference structural brain image (e.g., donor's T1w MRI; Fig. 2.2). The alignment between the reconstructed volume and the structural reference volume is performed using segmented GM volumes derived from each of these data sets, respectively. The problem of aligning a volume composed of heterogeneous pixel intensities to a reference volume with an entirely different pixel intensity distribution is thus simplified to mono-modal alignment between GM segmentation volumes.
3) Morphologically informed surface-based interpolation is used to estimate missing pixel intensities for locations where a type of section was not acquired (Fig. 2.3).

### Brainbuilder usage
BrainBuilder is designed to be flexible and simple to use. It can be run through a Python script or on the command line (Supplementary Fig. 1A). The essential information required for the reconstruction is stored in.csv files, which can easily be generated by users without experience in programming (Supplementary Fig. 1B). The code is openly available on GitHub: https://github.com/tfunck/brainbuilder. Additionally, BrainBuilder can be run through a Docker container: https://github.com/tfunck/brainbuilder.

### Visualization of 3D reconstruction
The multi-resolution algorithm for aligning sections to the structural reference volume produced increasingly accurate alignments, as shown exemplarily in Fig. 3 for the reconstruction of the most rostral tissue slab

**Fig. 3 | Illustration of hierarchical multi-resolution alignment to reference volume.** A hierarchical multi-resolution scheme was used to align the slab volumes to the reference structural volume. Sagittal images of the first tissue slab for each step of the alignment scheme show progressively finer alignment between the 2D coronal sections and structural reference volume. The edges of reference volume are shown in orange. For each iteration, all 2D GM segmentations are downsampled to the current resolution in the hierarchy and hence only contain morphological information at that resolution. **1** 3D segmentation volumes are created by applying 2D transformations to the segmented 2D images. In the first iteration, 4mm, only rigid transformations are used to create an initial 3D GM segmentation volume, but on subsequent iterations the non-linear transformations calculated from the previous iteration are used to create the 3D segmentation volumes. **2** The reference volume (the outline of which is shown in the second and third row in orange) is then 3D non-linearly aligned to the 3D segmentation volume. **3** The alignment is then refined using 2D non-linear transformations between the original 2D segmentations and their corresponding coronal section in the aligned structural reference volume.

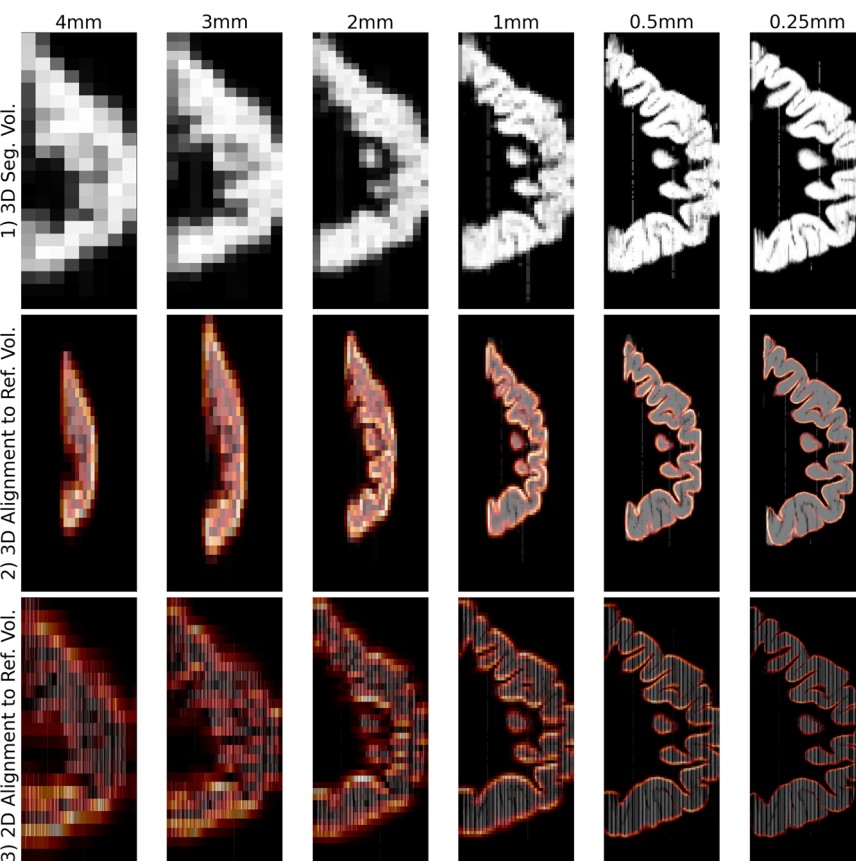

from a human hemisphere, where mismatches are clearly visible when the alignment was performed at 1 mm resolution, versus that at 0.5 mm Fig. 3.

To enable a greater flexibility of the pipeline, we implemented the option to reconstruct 2D images for which no structural reference volume was available. The alignment of sections acquired from a human donor and a macaque brain is shown in Fig. 4. Whereas the reconstruction of the human sections was performed with a T1w MRI as structural reference volume, the macaque sections were reconstructed with the MEBRAINS[36] stereotaxic template. In Fig. 4 the non-linearly aligned 2D sections are shown in a sagittal and axial view with WM and GM cortical surfaces overlaid on top. Visual inspection shows that the reconstruction performed similarly between the human reconstruction with a donor MRI and the macaque reconstruction using a reference template brain.

BrainBuilder was not only able to reconstruct volumes for multiple neurotransmitter receptor binding sites from the human (Fig. 5A) and macaque brains, but also for cell body and myelin stained sections from the latter species (Fig. 5B). BrainBuilder was able to accurately align human sections coding for receptor distribution patterns to their corresponding MRI volume, as well as macaque sections coding for receptor, cell body and myelin distribution patterns to a reference template brain MRI.

### U-Net GM segmentation

A U-Net[37] was trained to segment the GM in 2D sections using exclusively synthetic 2D images generated from a segmentation of the BigBrain.The segmentation accuracy of the *U*-Net was evaluated by calculating the Dice score[38] between GM segmentation predicted by the *U*-Net from real autoradiographs versus manually drawn GM labels for these sections. For comparison, the Dice scores were also calculated for segmentation produced with Otsu[38,39] thresholding. The results showed an average Dice score of $0.93 \pm 0.04$ for 40 human sections and $0.91 \pm 0.03$ for 20 macaque sections,

indicating that the network generalized reliably from human to macaque brain sections. The Dice score obtained with Otsu thresholding was only $0.85 \pm 0.08$ in the human sections and $0.8 \pm 0.14$ in macaque sections, showing a significantly more accurate and precise GM segmentation with the *U*-Net trained on synthetic data.

### Alignment of 2D Section to Reference Volume

The alignment accuracy of the reconstruction was quantified by using two Dice scores (Fig. 6). The first dice score is calculated between aligned 2D sections and their corresponding reference sections from the structural reference volume. The global average accuracy of the alignment was $0.95 \pm 0.03$ (Table 1) in the human brain and $0.93 \pm 0.08$ in the macaque.

A second Dice score was used to quantify the smoothness of the reconstruction between adjacent sections. Here the Dice score was also calculated between a given section and the adjacent sections in the anterior and posterior direction. Here, the average intersection Dice score was $0.93 \pm 0.05$ in the human and $0.90 \pm 0.1$ in the macaque.

Slabs closer to the rostral and caudal poles of the brain had higher average Dice scores than those closer to the center of the brain. This is likely because the anterior and posterior poles provide clear boundaries versus which to position the tissue slabs. Additionally, slabs 2–4 contain the temporal lobe which, because the brains were unfixed, is pressed against the dorsal lobes of the brain in the 2D images and requires significant non-linear deformation to correct.

The alignment of the sections was also validated against an external source. Average regional receptor densities were obtained for each of the 20 receptor binding sites from 29 paired regions (see section 5.4 for details) in a) manually defined cytoarchitectonic areas in the raw autoradiographs[33] and b) from the 3D reconstructed receptor volumes using the 3D Julich Brain Atlas[4] of cytoarchitectonic regions. The

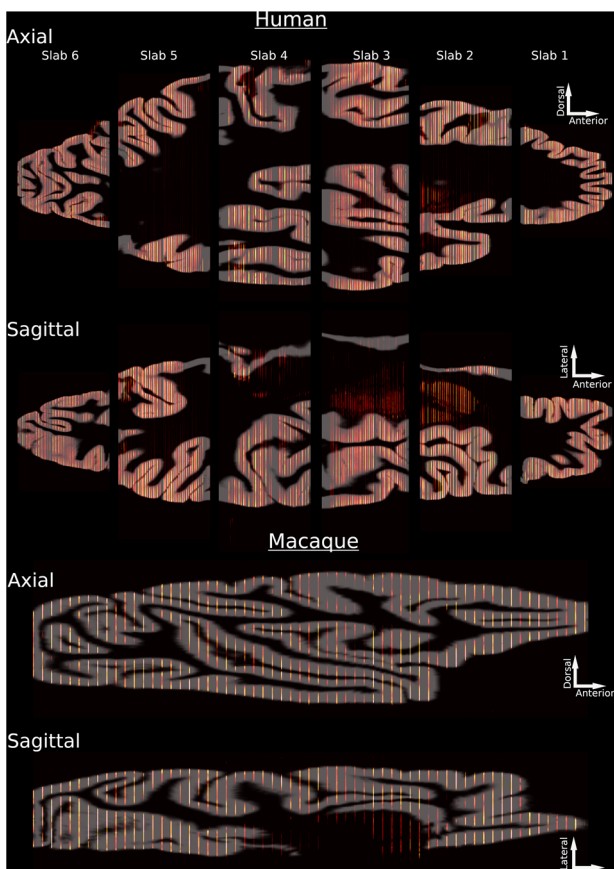

**Fig. 4 | Final alignment of sections to reference volume.** 2D coronal sections were aligned to 3D reference volumes and are shown along exemplary sagittal and axial cuts for the human and macaque reconstructions, respectively. Coronal sections are almost all correctly aligned to the cortical surfaces. Some less well aligned regions are visible in the dorsal portion of slab 4 and in slab 2 in the human, indicating that additional refinements may further improve the alignment.

manually defined regions consisted of one-dimensional profiles though the cortical depth but did not span the entire cytoarchitectonic area. Direct areal comparison of the 1D profiles to the 3D Julich Brain Atlas was therefore not feasible and regional densities between the corresponding regions were compared instead. The regional densities measured from the manually defined 2D regions were regressed onto the densities obtained in the 3D reconstruction. This produced an $r^2$ of 0.95 (Fig. 7), indicating a close correspondence between cytoarchitectonic areas defined in the raw sections and those defined in an independent 3D atlas.

### Surface-based estimation of missing pixel intensities

A surface-based interpolation algorithm was used to estimate missing pixel intensities between acquired autoradiographs. The surface-based interpolation was validated by applying it within randomly selected patches of vertices within acquired autoradiograph sections ($n = 10,000$). For each ligand, the overall correlation between true and interpolated pixel intensities was $r^2 = 0.97$ ($p < 0.001$) (Fig. 8). The distances between the vertices with known pixel intensities and the vertices to be estimated spanned 0.05–1.2 mm.

### Interpolation accuracy

The reconstruction pipeline was validated to quantify the amount of error induced in the pixel or voxel intensities due to resampling and interpolation. The accuracy of the measured pixel intensities was calculated after 2D non-linear alignment and after full 3D reconstruction. The accuracy after 2D non-linear alignment was ~97% and ~92% after 3D reconstruction (Fig. 9).

The standard deviation of the accuracy was 4% after 2D alignment and 6% after 3D reconstruction.

### Reconstructed values versus in vivo PET

The reconstructed GABA$_A$-benzodiazepine receptor volume exhibited a high degree of similarity to 10 PET-derived GABA$_A$-benzodiazepine distributions from healthy controls. Spearman's $\rho$ correlations between the reconstructed volume and PET scans ($\rho=0.71 \pm 0.06$) were comparable to those observed among PET scans themselves ($\rho=0.72 \pm .05$). Permutation testing ($n = 10,000$) revealed that the mean difference in correlation was not statistically significant ($p = 0.54$), indicating that the reconstructed receptor volume was as similar to the PET-derived receptor distributions as the PET scans were to one another. These results suggest that the reconstructed autoradiography-based receptor distribution closely aligns with in vivo PET-derived receptor mapping.

Receptor densities observed in the reconstructed GABA$_A$-benzodiazepine volumes were also not statistically different from the distribution of benzodiazepine receptor densities measured from in vivo PET scans in healthy participants, indicating accurate reconstruction of the autoradiographs.

### Discussion

We have created and validated BrainBuilder, a versatile pipeline that can successfully reconstruct 3D volumes from 2D multimodal datasets from different species, and without the constraint of requiring a structural volume of the donor brain. As a proof of principle, we have demonstrated this flexibility by reconstructing a set of receptor autoradiographs coding for the distribution of 20 neurotransmitter receptor binding sites in a human brain for which a T1w volume was available, and a dataset from a macaque monkey brain encompassing the distribution of 14 receptors as well as classical histological cell body and myelin stains, and using the MEBRAINS[36] stereotaxic template as a reference volume. In brief, all sections were aligned to one another using rigid transformations to create an initial 3D volume. An iterative multiresolution scheme was then used to non-linearly align the volume of 2D sections to the reference structural brain image. Finally, for each kind of section, a surface-based interpolation algorithm was used to create volumetric maps that represent the distribution of the biological feature measured by each respective 2D acquisition method throughout the cortical ribbon.

Visual and quantitative validation demonstrated that BrainBuilder effectively reconstructs the 3D anatomy of the reference volume, as shown in Fig. 4. The average Dice scores ranged from 0.93 to 0.95 across two alignment metrics in both human and macaque brains. Lower Dice scores were observed in slabs located farther from the rostral or caudal poles (Table 1), suggesting that the slab's position and anatomy influence the reconstruction's accuracy, likely because the rostral and caudal regions provide more easily identifiable landmarks. Additionally, receptor densities measured from manually defined cytoarchitectonic regions in the autoradiographs showed a close correlation ($r^2 = 0.95$) with those measured in the 3D reconstructed volumes using an external 3D cytoarchitectonic atlas[4]. The surface-based interpolation also displayed high accuracy, with a correlation of $r^2 = 0.97$ between true and estimated pixel intensities, confirming reliable estimation of missing pixel intensities across the cortical surface. Receptor densities observed in the reconstructed GABA$_A$-benzodiazepine volumes were also not statistically different from the distribution of benzodiazepine receptor densities measured from in vivo PET scans in healthy participants, indicating accurate reconstruction of the autoradiographs.

The reconstruction of the macaque brain was performed without an individual MRI, using the MEBRAINS[36] stereotaxic atlas instead. Our method provides, therefore, the flexibility to reconstruct data sets of 2D sections where no corresponding 3D structural image has been acquired. An important caveat is that, while still presenting a conspicuous gyrification pattern, macaque brain morphology is much simpler than that of humans. Therefore, while it may not be necessary to have a corresponding 3D structural image when reconstructing sections from animals with a lower

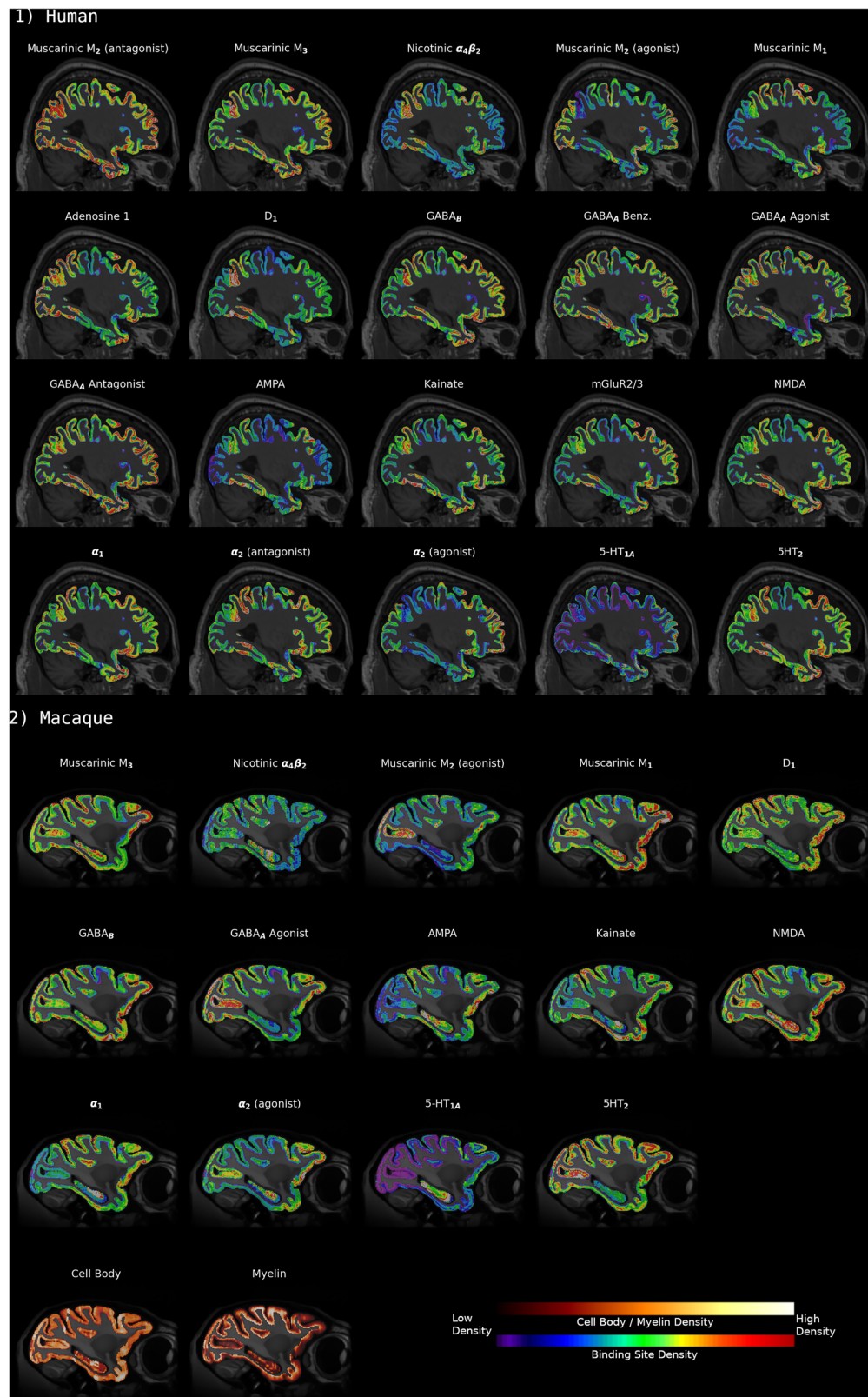

**Fig. 5 | Complete reconstructed cortical volumes for each acquisition modality.** Reconstructed sagittal sections show the binding sites for a **1** human brain and **2** a macaque brain. The macaque brain also includes sections indicating cell body and myelin density. Because each acquired section can only be visualized for a single biological feature, e.g., a given binding site, there are necessarily gaps between any two sections for a given acquisition and thus creates missing pixel intensities for that acquisition. The problem of missing pixel intensities is further aggravated by the occasional tissue section that has been lost due to mechanical processing errors. Missing pixel intensities are estimated using surface-based linear interpolation.

**Fig. 6 | Quantitative validation of alignment to reference and inter-section alignment. A** Dice scores for two different validations are shown. Reference: Dice score between aligned sections versus corresponding coronal section from reference volume, Inter-section: Average Dice score between aligned section versus neighbors in posterior and anterior direction along coronal axis. Overall Dice score were high 0.89–0.95 but with some sections that appear to be poorly aligned. **B** Examples of sections that were either very poorly (left) or well (right) aligned. The grey images indicate the non-linearly aligned 2D segmentations and the hot (orange-red) lines are the edges of the target 2D section from the structural reference volume. The poorly aligned sections show that the source of misalignment is frequently 2D sections that are either very small (Slab 1,2,6), poorly segmented (Slab 4), or incomplete sections (Slab 2,3). The sections that have been well aligned consistently demonstrate a Dice score of 0.97, indicating an upper ceiling for BrainBuilder.

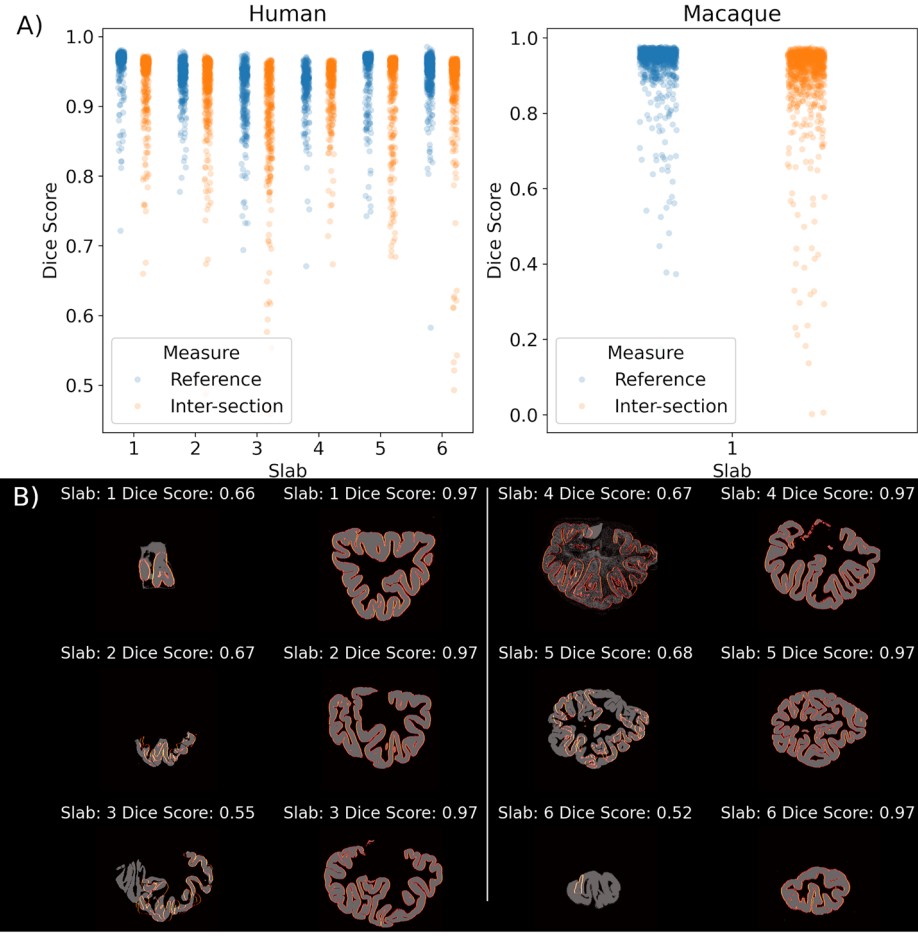

## Table 1 | The accuracy of the alignment was measured using two different Dice scores

| Slab | 1 (Anterior Pole) | 2 | Human 3 | 4 | 5 | 6 (Posterior Pole) | Macaque Whole Brain |
|---|---|---|---|---|---|---|---|
| Inter-section Dice Score | 0.94 ± 0.04 | 0.93 ± 0.05 | 0.89 ± 0.07 | 0.92 ± 0.04 | 0.93 ± 0.06 | 0.94 ± 0.06 | 0.90 ± 0.1 |
| Section to Reference Dice Score | 0.96 ± 0.06 | 0.94 ± 0.03 | 0.93 ± 0.04 | 0.93 ± 0.04 | 0.95 ± 0.05 | 0.95 ± 0.02 | 0.93 ± 0.08 |

The Dice score of the non-linearly aligned section versus the corresponding reference section from the structural volume ("Reference") and the Dice score between a given section and its immediate neighbors ("Inter-section"). Both Dice scores were consistently high across all tissue slabs and in both human and macaque reconstructions. In the multi-slab human reconstruction, the middle slabs had slightly lower Dice scores (0.89–0.92) versus at the poles. These lower scores likely reflecting both the added difficulty of identifying the anterior and posterior bounds of these slabs on the reference volume and the more varied set of morphological features in these middle slabs (e.g., basal ganglia and temporal lobe).

degree of gyrification, it may be required to accurately reconstruct sections acquired from human brains.

We chose not to apply existing semi-automated reconstruction methods based on manual identification of anatomic landmarks[16,17] because they are dependent on rater subjectivity and not easily reproducible. For the human data, the bounds of the tissue slabs on the reference structural image were manually identified, but these points serve only to constrain the 3D alignment and do not drive the reconstruction process. It should also be noted that no manually selected points are used in the reconstruction of the macaque data because the data were acquired within a single tissue slab. Hence, BrainBuilder can be used in a fully automated fashion when the sections have been acquired within a single slab of tissue spanning the entire brain. We were not able to apply 2D reconstruction methods using fiducial markers[10–12] or relying on block-face imaging[13–15], because neither fiducial markers nor block-face images were used in the acquisition of our data.

A similar approach to ours was used by both Malandain et al.[30] and Amunts et al.[1] to reconstruct histological volumes in 3D that iterates between two steps: first the donor MRI is aligned in 3D to a stack of sections and then, in the second step, the sections are linearly aligned in 2D to the transformed MRI. Amunts et al.[1] improved on the 3D-2D reconstruction approach by using a multi-resolution non-linear warping schema. However, this 3D-2D approach cannot be used for multimodal datasets due to their heterogeneous pixel intensity distributions. To address these problems, we transformed all acquired sections to binary GM masks and recalculated a continuous 3D GM volume between acquired sections at each resolution. The progressively smoother 3D representation of the GM improves alignment to the structural reference volume and allows for the reconstruction of even very sparsely sampled sections.

Our approach differs fundamentally from existing methods through the use of morphologically informed estimation of missing pixel intensities using cortical surface-based interpolation. The advantage of this approach is that cortical surfaces and geodesic distances better represent the actual morphology of the cortex than Euclidean distances because the cortex is organized into layers over a folded manifold.

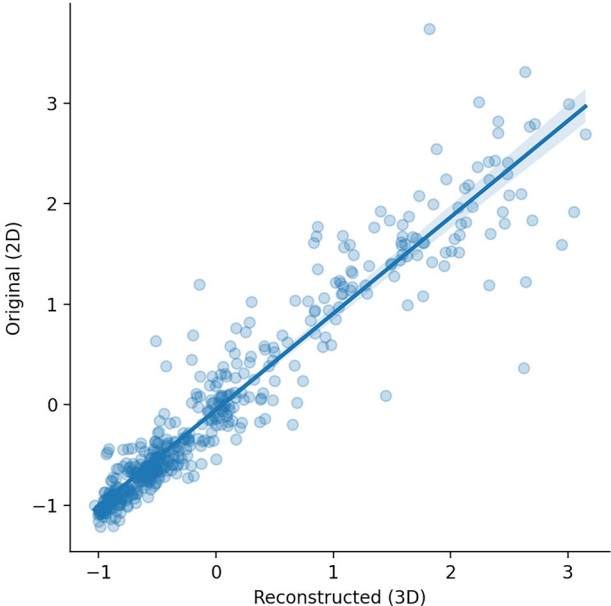

**Fig. 7 | Comparison of receptor densities measured with 2D versus corresponding 3D ROIs.** The strong correlation between regional receptor densities measured from areas manually defined on 2D autoradiograph versus the same regional densities defined in the same regions on the Julich Brain Atlas4 indicate that BrainBuilder produces accurate alignments at the scale of cytoarchitectonic areas.

Many methods have been proposed to create smooth and continuous 3D volumes from 2D sections without the use of external references[25–27,29], such as fiduciary markers, block face images, or structural reference volumes. In principle, these methods would be helpful for creating the initial reconstruction. However, these methods cannot be directly applied here due to the fact that our dataset consists of sparse, irregularly sampled sections with heterogeneous pixel intensities that cannot be naively stacked and aligned in 3D to a reference volume. Fortunately, because the multi-resolution schema begins at a very low spatial resolution of 4mm³, it only requires that the sections be grossly aligned to start with.

Another promising approach has been proposed by Iglesias et al.[28] to align sections to corresponding sections from MRI volumes. Their method uses Bayesian inference to estimate both physical transformations to the sections and the transformation of pixel intensities to align sections acquired with different intensity distributions. A key assumption of this approach is that the pixel intensities in sections are a warped version of the corresponding MRI. This assumption is sensible given that both MRI signal and histological signal derive directly, in the case of histology, or indirectly, in the case of MRI, from the density of cells and myelin in the brain tissue. It is not clear if this assumption would hold if it were to be applied to modalities whose distribution of pixel intensities is independent of cyto- and myeloarchitecture, as are the laminar distributions of receptor binding sites[34].

Our method, by contrast, converts all sections to binary GM segmentations and thereby transforms the problem of multi-modal alignment to one uni-modal alignment that can be solved with traditional intensity-based alignment algorithms, such as ANTS[40]. Our approach, therefore, only assumes that the GM can be accurately segmented within the sections and does not assume an intensity-based relationship between the acquired sections and the structural reference volume. This not only means that BrainBuilder can be used with a wide range of 2D ex vivo imaging modalities but also that our approach can in principle be applied with non-MR based structural reference volumes, e.g., with computed tomography, provided the latter have sufficient resolution to support accurate reconstruction.

Future work will apply the Iglesias, et al.[28] joint registration and synthesis approach to evaluate whether it provides enhanced 2D alignment to the methods presented here. If so, their approach can be integrated into the overall BrainBuilder framework to provide even more precise 3D reconstruction.

BrainBuilder is subject to some limitations, many of which will be the subject of future work to improve the reconstruction pipeline. The surface-based interpolation scheme used here performs linear interpolation between vertices and therefore assumes that pixel intensities measured from sections change linearly between the acquired sections and the missing section. Linear interpolation was chosen because it will only produce values within the range of those observed in the acquired sections and because it provides easily interpretable results. However, this approach is not strictly biologically valid because there are sharp boundaries between receptor architectonic areas[33,34], which would be obscured by this interpolation method. It does not appear possible to devise an interpolation method that could reproduce such sharp regional boundaries without additional anatomic information. More work is required to establish which interpolation scheme, e.g., linear, spline, AI-based in-painting, Gaussian process regression, produces the most reliable estimation of missing pixel intensities.

A potential limitation of our reconstruction method is that it relies on the segmentation of accurate GM images from 2D sections, whether of receptor density or cell and myelin-stained sections. It is possible that there are imaging modalities where this estimation of cortical GM may be more challenging. However, the network used in this work is trained using only synthetic data, which can, in principle, be extended to include different tissue contrasts, more tissue classes, and additional imaging artefacts[41]. This means that if there is a particular type of section that is not well segmented by the current network, the synthetic data set can be augmented to reflect the particularities of the problematic sections and hence improve the segmentation of these sections. Future work will expand the segmentation to include not only the cortical but also the subcortical GM.

Currently, BrainBuilder uses linear interpolation to fill missing segmented images between sections to create continuous 3D segmentation volumes. However, at higher spatial resolution, this may produce jagged 3D volumes. Hence, for higher resolution reconstructions, we will implement higher-order interpolation algorithms to create smoother 3D segmentation volumes for alignment to the structural reference volume.

The reconstruction process is computationally expensive due to the large number of sections and the high spatial resolution of the reconstructed volumes. The 3D nonlinear alignment step using ANTs is the primary bottleneck in terms of memory usage; however, if sufficient RAM is available for this step, then the rest of the pipeline can run without additional constraints. For the reconstruction of a human hemisphere at 250 μm resolution, RAM usage peaked between 64 and 128 GB. Extrapolating from this, a 50 μm reconstruction would require ~1 TB of RAM, which exceeds the capacity of standard workstations and requires access to high-performance computing clusters.

To address these limitations, future versions of BrainBuilder will implement a piecemeal 3D volumetric alignment approach—aligning sub-volumes independently—to reduce the memory burden for very high-resolution reconstructions. This strategy will allow for more efficient scaling while maintaining the accuracy of the reconstructed volumes.

BrainBuilder relies on sequential processing stages where errors in earlier stages may be propagated to the end result. While the validation tests aim to ensure the correct functioning of each step and the accuracy of the final reconstructed values, errors remain a possibility especially with new datasets. Several measures are implemented to facilitate the identification of errors. Quality control images are generated for all downsampled, segmented images, and for the alignment between 2D sections versus corresponding sections in the target reference volume. Additionally, Dice scores are calculated for all 2D alignments to facilitate plotting and identification of outlier scores that may indicate poor alignment.

BrainBuilder presently only reconstructs the cortex and the hippocampus and does not include the subcortex. Nonetheless, the sagittal cross-section of the human reconstruction in Fig. 4 shows that the subcortex is surprisingly well aligned given that this region was not included in the reference volume used for reconstruction. This will be improved by the

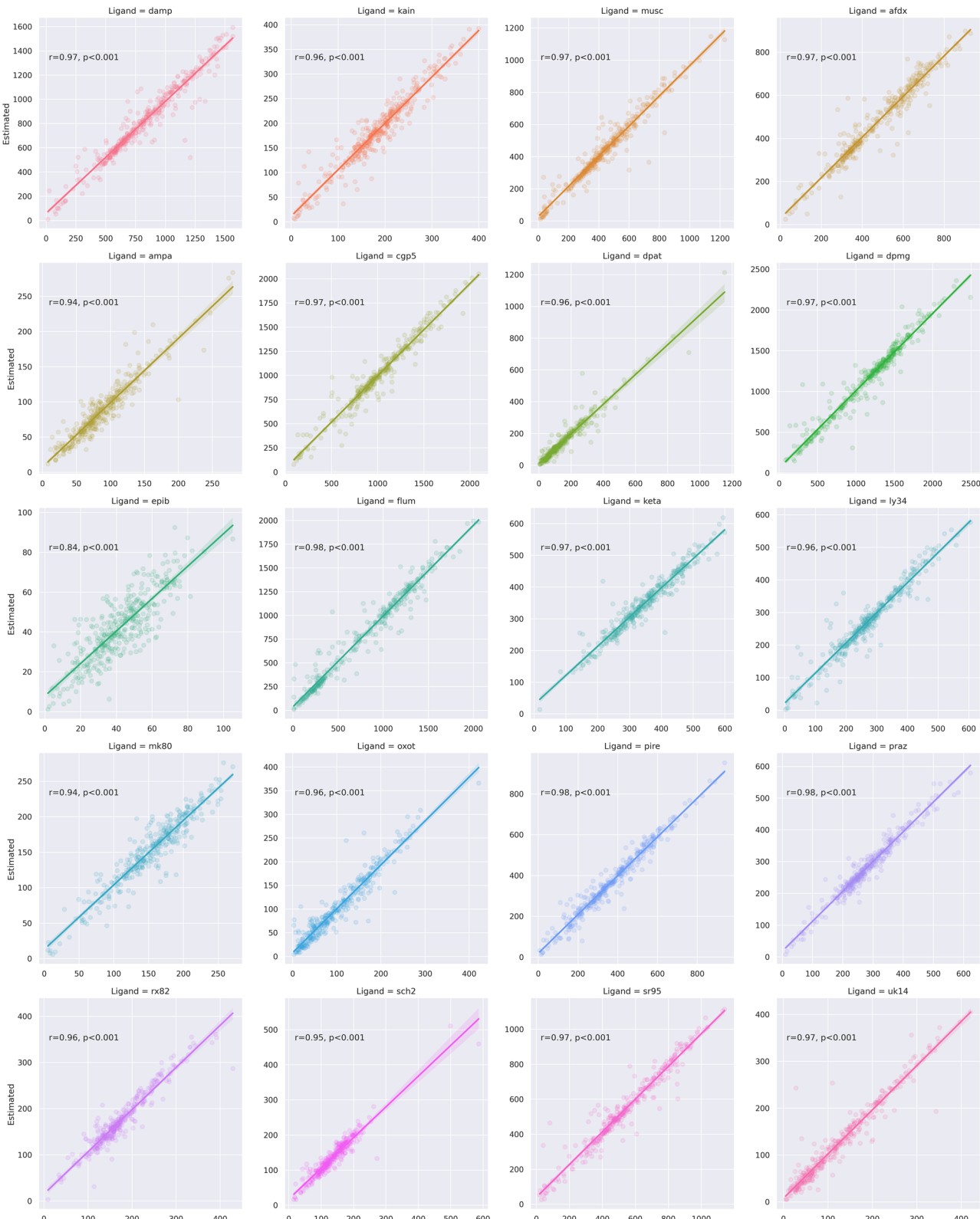

**Fig. 8 | Validation of surface-based interpolation within acquired sections.** The surface-based interpolation algorithm was evaluated within the human autoradiograph sections and demonstrated a high correlation between interpolated and true pixel intensities.

inclusion of the subcortex in the U-Net segmentation network and in the reference volume. Future work will expand BrainBuilder to also include the subcortex by implementing morphologically informed interpolation of missing sections.

We have created an image processing pipeline for reconstructing 2D sections into 3D volumes. The results here serve as a proof-of-principle that BrainBuilder can reconstruct images of 2D sections processed for the visualization of receptor, cell body or myelin density accurately at high

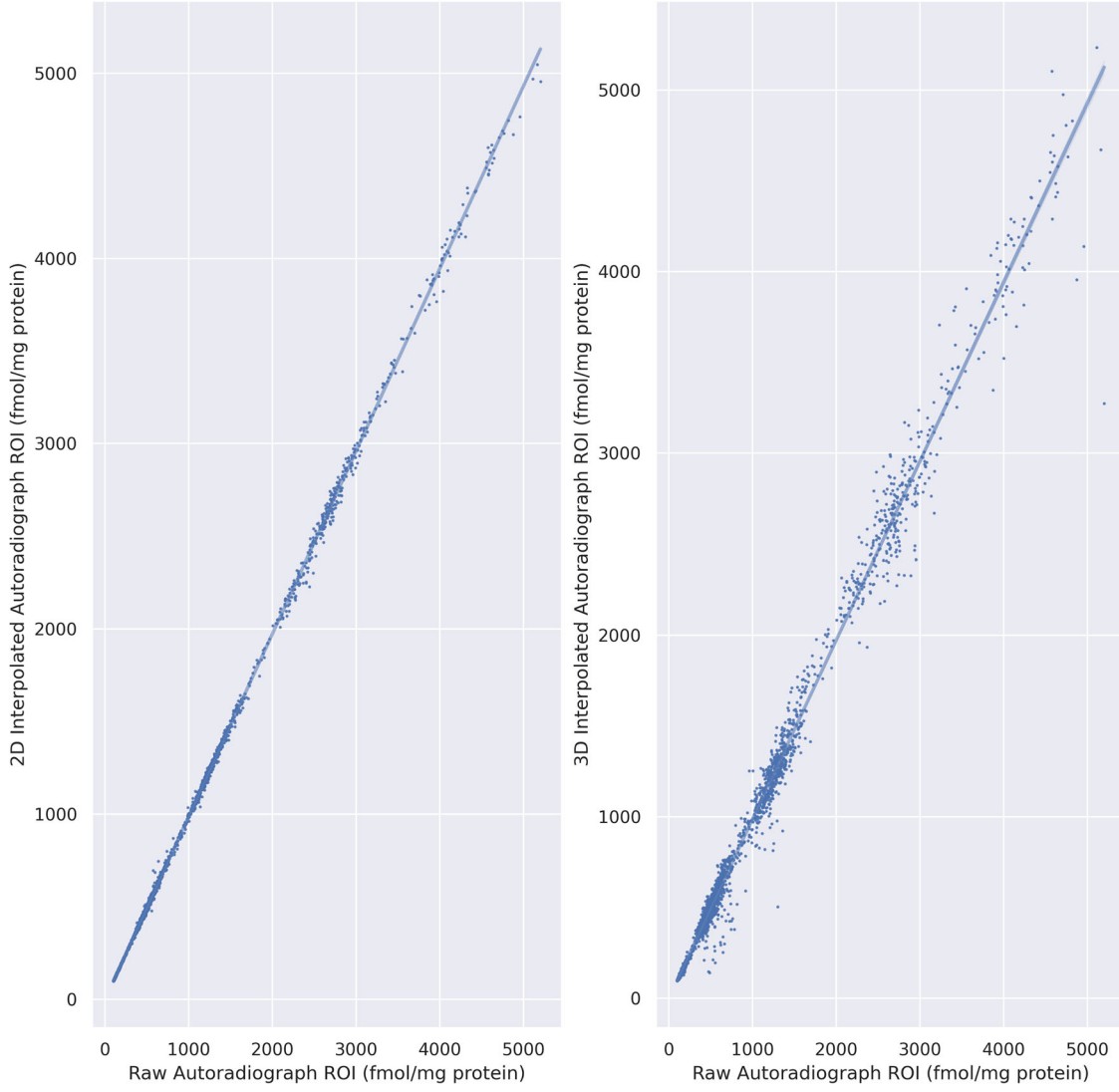

**Fig. 9 | Measurement of interpolation error between raw and reconstructed autoradiograph sections.** Binding densities were measured from human autoradiographs after applying 2D (left) and then 3D (middle) non-linear transformations to the autoradiographs and compared to binding densities measured from the raw autoradiographs. While the mean accuracy is similar after 2D alignment and 3D reconstruction, the variance of the accuracy increases for smaller sections after 3D reconstruction.

resolution in both the human and the macaque brain. We have also demonstrated that this can be done even when no MRI volume is available for the 2D data to be reconstructed. The work presented here will allow for the creation of an unparalleled data set of 20 receptor binding site volumes at 20 μm for three human brains and four hemispheres from macaque brains. Future work will focus on extending BrainBuilder to also reconstruct subcortical structures and to evaluate its use on other species.

## Methods

### Data acquisition and preprocessing

The macaque specimen was obtained from Covance (now Labcorp Drug Development), Munster, where they were used as control animals for pharmaceutical studies performed in compliance with legal requirements. The specimen was male, ~6 years old. All experimental protocols were in accordance with the guidelines of the European laws for the care and use of animals for scientific purposes. We have complied with all relevant ethical regulations for animal use.

The human participant had given written consent before death and/or had been included in the body donor program of the Department of Anatomy, University of Dusseldorf, Germany. The human donor was a 78 year old male; who died from non-neurological causes. All ethical regulations relevant to human research participants were followed. While these brains are part of a larger data set consisting of three complete human brains and four macaque brain hemispheres, only a single human and macaque hemisphere was used for proof-of-principle of the proposed reconstruction method.

The pipeline was initially developed for the 3D reconstruction of 2D serial sections through the human brain, which had been processed by in vitro receptor autoradiography for the visualization of multiple neurotransmitter binding site densities, and for which a structural reference volume of the donor brain was available[33,34]. The pipeline was then adapted to enable reconstruction of a comparable 2D multimodal dataset obtained from the macaque brain, but which also included sections stained for cell bodies and myelin fibers, and for which no structural reference volume of the donor brain was available.

For both humans and macaques, brains were removed from the skull at autopsy, the hemispheres were separated and the cerebellum removed. Human hemispheres were cut into 6 coronal slabs, each ~2–3 cm thick, macaque hemispheres were cut in the coronal plane into a rostral and a caudal block. Brain slabs were shock frozen and serially sectioned in the

coronal plane. ~1402 sections were acquired for the macaque hemisphere and 3142 sections in the human hemisphere. Alternating sections were processed by in vitro receptor autoradiography for the visualization of multiple types of receptor binding sites (20 in the human and 14 in the macaque brain), or histologically for the staining of cell bodies and of myelin[33,34]. The radioactively labeled sections were exposed against tritium-sensitive films and the ensuing autoradiographs digitized as 8 bit images with an in-plane spatial resolution of 20 μm resolution for the human and macaque sections, respectively by means of a CCD camera attached to the image acquisition and processing software Axiovision (Zeiss, Germany). Images of the cell-body and myelin stained sections were acquired with a TISSUEscope™ Huron Scanner (Huron, Canada) as 8 bit images with an in-plane spatial resolution of 1 μm per pixel.

For simplicity, in the following steps we will use the term "section" to refer to the digitized images of both the receptor autoradiographs and the cell-body or myelin stained sections.

Sections were quality controlled and sections that were severely damaged, had clear binding artefacts, were visualized for non-specific binding, or were missing more than 25% of the cortex were excluded. In the human data 693 sections failed quality control, most of which were at the ends of slabs where only small pieces of cortex were sliced from the frozen tissue slabs, and 384 for the macaque.

For both species, sections were preprocessed to isolate the target piece of brain tissue from each image and remove extraneous tissue and visual cues (for details, see Funck, et al.[42]).

For the human brain, a binary MRI GM volume was derived from the donor's T1w MRI using a mesh representation of the cortical surface. The contrast between the GM and surrounding tissue in the sections facilitates alignment to the GM in the structural reference volume. Cortical surface meshes were obtained from the MRI using the CIVET pipeline[43]. A super-resolution cortical GM mask at 250 μm was obtained from these cortical surface meshes by sampling points between the inner white-matter and outer GM surface meshes[44].

For the macaque reconstruction, the MEBRAINS template was used as reference structural template[36]. Hence for the macaque reconstruction, the structural reference volume did not come from the donor brain. The MEBRAINS data release provides cortical surfaces derived with Freesurfer[45,46] and these were used for the reconstruction.

[18F]Flumazenil PET scans were acquired for ten healthy subjects (age 61 ± 10 years, 10 males)[44]. PET scans for all subjects with an ECAT HRRT PET scanner in list mode (Siemens Medical Solutions, Knoxville, TN, USA)[47], with a spatial resolution of ~2.4 mm FWHM. After a transmission scan for attenuation correction (137Cs-source), ~370 MBq [18 F] FMZ were injected intravenously as a slow bolus over 60 s. The list mode data were acquired for 60 min after injection and were subsequently binned into 2209 sinograms (each of size 256 radial bins × 288 azimuthal bins) using span nine compression for a total of 17 time frames (40 s, 20 s, 2 × 30 s, 360 s, 4 × 50 s, 3 × 300 s, and 3 × 600 s), resulting in images with a voxel size of $1.22 \times 1.22 \times 1.22$ mm³. Fully 3D FBP by 3D reprojection was carried out with a Hamming windowed Colsher filter (alpha = 0.5, cut off at the Nyquist frequency).

For each scan, normalized standardized uptake value ratio (SUVR) images were calculated by dividing the PET images by the mean radio-activity concentration in the WM.

## BrainBuilder: initial inter-section alignment

The initial step of 3D reconstruction involves aligning the sections from all available modalities to one another using 2D rigid body transformations. For simplicity, the ensuing volume is designated as the "initial volume." Low contrast sections are more likely to be misaligned due to the difficulty in resolving all anatomic structures. It is therefore desirable to first align sections with high contrast before proceeding to lower contrast sections. To this end, sections are first ranked automatically by the Michelson contrast[48] of pixel intensity produced by each acquisition method.

Within each tissue slab, the central section is designated as "fixed" and serves as the reference to which all subsequent sections are aligned. Moving outwards from the central section, sections visualizing the same biological structure (e.g., a given receptor type) are aligned to their nearest "fixed" neighbor towards the center of the slab. Once aligned, each newly added section becomes "fixed" in place.

After aligning the highest contrast sections, they collectively serve as a reference against which to align subsequent lower contrast sections. This process iterates with progressively lower contrast sections until all sections have been aligned.

For the reconstruction of the human brain, 6 initial volumes (one for each of the slabs into which the hemisphere was cut) were produced by rigid alignment of the section. For the macaque brain, the sections were treated as belonging to only a single slab and hence a single initial volume was produced for the acquired hemisphere.

All alignments were calculated with ANTs. For details, see Supplementary Information 5.1.1.

## Brainbuilder: GM segmentation

Binary GM sections are generated from the sections using a deep neural network. A multi-tissue segmentation of BigBrain at 200 micron isotropic resolution was used to generate 10,000 2D training images (Supplementary Fig. 3). To better synthesize intact in vivo MRIs, a skull segmentation of the MNI152 atlas was coregistered to the BigBrain and used in 50% of training examples. To generate synthetic training examples, random affine transformations (with random scaling ~ uniform(0.9,1.1) and rotations ~ uniform(0,15°)) were first used to create randomly transformed versions of the segmented BigBrain volume. From each of these transformed volumes, 500 2D sections were extracted and used as a training example. To synthesize heterogenous laminar structure, the cortex was segmented into a random number of cortical layers between 0 and 11, with random proportional thicknesses[49]. The considered tissue classes included: WM, subcortical GM, cortical GM layers, cerebellar GM, within- and outside- brain background, and skull. For each training example, randomly generated values were assigned to each tissue class in the 2D section from Gaussian distributions. To introduce additional variation in the synthetic images, random smoothing, noise, filtering, scaling, and cropping were applied to training examples (see Supplementary Information 5.1).

The nnUNet (v.1) software package[50] was used to automatically identify optimal hyperparameters and train a U-Net[37]. The U-Net was trained to segment synthetic sections into cortex, background and WM, with a second task to identify the pixels at the boundaries between these image classes.

While all images in the present study were successfully segmented with the U-Net, in the eventuality that the network fails to provide a segmentation, i.e., returns an empty image, a segmentation method based on Otsu[39] histogram thresholding is used.

## Brainbuilder: alignment of 2D section to reference volume

The alignment of the initial volume to the structural reference volume was done within a multiresolution hierarchical framework. If the sections were sliced from a whole brain, then the structural reference volume comprises a whole brain. If, as is the case of the human data in this work, the sections were sliced from tissue slabs, then the structural reference volume is also a slab extracted from a whole brain. The steps of the BrainBuilder pipeline described in this section are repeated for each resolution in the hierarchy. The resolutions in the hierarchy were 4.0, 3.0, 2.0, 1.0, 0.5 mm and finally, at 0.25 mm. This resolution schedule is specified by the user at run-time and can be modified to suit the user's particular dataset.

3D reconstruction of a multimodal 2D dataset results in a single volume composed of extremely heterogeneous pixel intensities that are discontinuous between neighboring sections. This makes it impossible to perform volumetric alignment using either cross-correlation or an information theoretic cost function. We simplified the problem of aligning the heterogeneous initial volume to the structural reference volume by creating

binary masks representing cortical pixels in both the initial and the reference volumes.

At each resolution in the multiresolution hierarchy, 2D GM sections segmented with the U-Net are transformed using the best available transformations to align them together into a single 3D GM volume. At the initial resolution, the rigid body transformations calculated in the initial inter-section alignment serve as the best available transformations. At subsequent resolutions, the best alignments are the non-linear 2D transformations calculated for the previous resolution in the hierarchy (described below). Therefore, after the first step of the multiresolution hierarchy, the 2D GM sections are warped so that they better correspond to the actual anatomy of the structural reference volume to which they were aligned at the previous resolution of the hierarchy.

The aligned 2D GM sections contain gaps along the coronal axis where 1) no sections were acquired for a particular modality and 2) due to sections lost during acquisition. When reconstructing complete volumes for a particular modality, the missing pixel intensities produced by these gaps must be filled to enable a continuous representation of the cortical GM. We used linear interpolation between aligned 2D GM sections to estimate the morphology of the cortex where no sections could be acquired.

When more than one slab of tissue is reconstructed, as in the human data set, a significant challenge is to identify which portion of the structural reference volume corresponds to each slab of brain tissue. Due to the deformation of the tissue slabs prior to freezing and loss of sections between slabs, the total width of the brain slabs along the coronal axis was less than that of the brain in the MRI volume, hence the slabs could not simply be placed adjacent to one another.

We manually identified the anterior and posterior most points of each slab on the structural reference volume and extracted a corresponding slab of tissue from the reference volume. The points were identified by an expert neuroanatomist who localized structures that were easily and uniquely identifiable in both the MR images and both autoradiographs and photographs of the slabs before and after their shock freezing. These structures included the fundus of sulci, tips of gyri and the edges of dimples. We also identified the most rostral and/or caudal portions of the corpus callosum and of subcortical structures such as the caudate nucleus or the pulvinar nucleus. Importantly, we did not restrict identification of microanatomical landmarks to the most rostral and caudal portions of the slab, i.e., to the portion corresponding to the interface between slabs. Rather, we set landmarks throughout the entire lateral and medial surfaces of each slab. The alignment of the receptor slab was therefore limited to a manually defined portion of the corresponding structural reference volume. This was only necessary for the human data, whose acquisition within non-parallel tissue slabs is idiosyncratic. For most data sets, e.g., as was effectively the case for the macaque data set, step may be omitted if sections sampled from a single tissue slab spanning a whole hemisphere are being reconstructed.

For each slab, the reconstructed GM volume is non-linearly aligned with ANTs[40] to the portion of the structural reference volume. For details, see Supplementary Information 5.3.2.

After the initial 3D alignment of the reconstructed GM volume to the structural reference volume, the alignment is refined by aligning the sections in 2D to their corresponding coronal sections in the structural reference volume. This 2D alignment between corresponding coronal sections is possible because the 3D alignment in the previous stage produces a structural reference volume that has been transformed into the coordinate space of the reconstructed GM volumes for each tissue slab. The alignment is performed with ANTs[40], for details see Supplementary Information 5.3.2.

### Brainbuilder: Interpolation of missing pixel intensities

Intermediate cortical surfaces are generated by evenly subdividing the WM-GM and pial-GM border (Supplementary Fig. 3.A). Each cortical mesh is then supersampled in BrainBuilder such that the maximum distance between any two neighboring vertices is less than or equal to the final resolution of the reconstruction, i.e., 500 μm for the human reconstruction and 1 mm for the macaque reconstruction (Supplementary Fig. 3.C).

Specifically, for each triangle of three vertices ($A$,$B$,$C$) on the surface mesh, two vectors are defined (**AB, AC**). Points inside the surface triangle, $P$, are defined by the linear combination of the two vectors by scaling factors $\alpha$ and $\beta$, where $\alpha$ and $\beta$ are defined as increments of one half the reconstruction resolution, $r$:

$$P = A + \alpha \mathbf{AB} + \beta \mathbf{AC}; i = \mathbf{AB}/(0.5r); \alpha \in [0,1], \beta \in [0,1], \alpha + \beta \in [0,1]$$

This upsampling step produces a set of points such that there is at least one vertex per voxel in sections where sections have been acquired. For a reconstruction of 0.25 mm the upsampled mesh in the humans was 12,129,727 vertices in the human and 1,022,442 vertices in the macaque.

For the human brain, 18 intermediate cortical surfaces were generated, yielding a total of 20 cortical meshes spanning the depth of the cortex between the WM-GM and GM-pial border (Supplementary Fig. 3.B). In the macaque brain, eight intermediate cortical surface meshes are generated between the WM-GM and GM-pial border, yielding a total of ten cortical surface meshes. The number of surfaces would be increased for higher resolution reconstructions such that at least one intermediate surface would intersect every pixel between the WM-GM and GM-pial border.

The surface meshes are transformed with ANTs[40] from the coordinate space of the structural reference volume to the coordinate space of each of the slab volumes, respectively, by applying the inverse 3D linear transformations and non-linear deformation fields that align the reconstructed volume to the reference volume. The pixel intensities in sections are projected onto the surfaces in the native coordinate space of the slab with nearest neighbor interpolation (Supplementary Fig. 3.D).

When pixel intensities from acquired sections are projected onto the cortical mesh, the gaps between acquired sections from a given modality result in vertices with missing pixel intensities. These missing vertex values are estimated for vertices at which no pixel intensities were measured using a surface-based approach. All of the meshes along the depth of the native cortical surfaces are inflated to spheres (Supplementary Fig. 3.E) using the Freesurfer's mris_sphere (iterations = 100) and mris_inflate[46]. The inflated spherical meshes are then upsampled to the same number of vertices as the already upsampled cortical surfaces (Supplementary Fig. 3.F) so that the vertex values can be estimated at a sufficient spatial resolution for the desired reconstruction resolution.

The missing pixel intensities for each cortical surface mesh are interpolated by applying linear interpolation over each of the corresponding upsampled inflated spherical meshes (Supplementary Fig. 3.G) using the SSRFPACK algorithm[51] implemented in the stripy[52] Python package. The interpolation was performed on the inflated sphere instead of directly on the cortical surfaces, because it is computationally simpler to calculate distances between vertices on a simple geometric object like a sphere than on the complex surface of the cortex. Furthermore, given that the surface inflation process preserves relative distances between vertices[46], the interpolated densities on the inflated sphere are equivalent to those which would have been calculated directly on the cortical surface.

Finally, we implemented an algorithm to project intensity values from the surfaces spanning the cortex into a volume (Supplementary Fig. 3.H). This was done for each voxel within the cortical ribbon by averaging the intensities of the vertices located within the volume of each respective voxel. This method may leave gaps in the reconstructed cortex where no surface vertices are located within the volume of a voxel. The voxel intensities of these empty voxels are estimated by linear interpolation based on the values of the neighboring voxels with pixel intensities.

### Statistics and reproducibility

Statistics were calculated in the context of validating various aspects of the Brainbuilder pipeline. Statistical computations were performed using Python (NumPy, SciPy). Sample sizes, statistical metrics, and validation protocols are specified for each experiment in the relevant methods subsections.

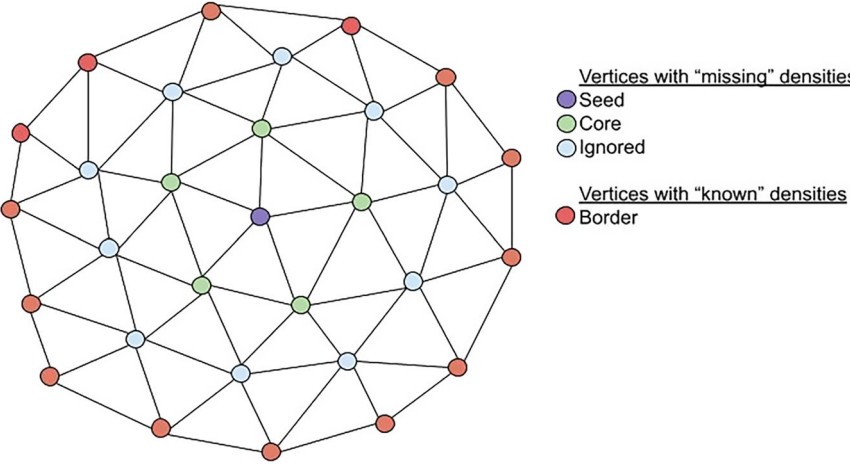

**Fig. 10 | Illustration of validation method for surface-based interpolation.** A toy example of a mesh patch used to validate the surface-based linear interpolation for estimating missing pixel intensities. Here a seed vertex, purple, with a neighbourhood of core vertices, m = 1, is estimated using known border vertices, red, n=3.

### Validation: U-NET GM segmentation

The prediction accuracy of the U-Net was validated using manually segmented GM images from randomly selected autoradiographs from the human data ($n = 39$) and macaque data ($n = 10$). The manual GM segmentations were drawn on the raw images using the GNU Image Processing (GIMP) software. The raw and segmented images were downsampled to images of $415 \times 558$ pixels to reflect the image size used to train the network. For comparison, segmented GM images were also created using Otsu thresholding[39]. The Dice score[38] between the predicted GM, both using U-Net and Otsu thresholding, and the manually segmented GM was used to validate the accuracy of the segmentation.

### Validation: alignment accuracy of brain sections

To demonstrate the efficacy of BrainBuilder, it was applied to sections acquired from human and macaque brains. The final reconstruction for the human and macaque data was generated at 250 µm isotropic voxel resolution. This resolution was chosen because it was sufficient to demonstrate the accuracy of the reconstruction. The alignment accuracy was evaluated in three ways. First, we calculated the Dice score between the aligned GM segmentation of the sections and the corresponding sections in the GM structural reference volume. Next, to quantify the accuracy of the inter-section alignment, we calculated the Dice score between adjacent sections in the anterior and posterior directions along the coronal axis, averaging these values. Finally, we compared average receptor densities measured in cytoarchitectonic areas defined on the raw autoradiographs to average densities measured from the 3D reconstructed volumes using an independent 3D cytoarchitectonic atlas[4].

A challenge in quantifying the accuracy of the alignment is that several of the tissue sections were damaged during acquisition and are missing pieces of tissue. Hence, even if the remaining tissue from a damaged section is perfectly aligned to the corresponding section from the structural reference volume, the resulting Dice score[53] will be low and not reflect the accuracy of the alignment with the available tissue. To avoid this problem quantitative validation was performed by calculating the dice score within a $5 \times 5$ moving window between the aligned GM segmentation of the sections and the corresponding sections in the GM structural reference volume. These local Dice scores were only calculated where tissue was available in the GM tissue segmentations. The local Dice scores were then averaged to produce a global Dice score.

Finally, to validate the alignment using independently defined cortical regions, we compared the neurotransmitter receptor densities measured in specific cytoarchitectonic areas, manually defined on the raw 2D images from previous work by expert neuroanatomists[33,34], to the densities in the

same cytoarchitectonic areas as defined on the Julich Brain cytoarchitectonic atlas[4]. The donor brain was non-linearly aligned using ANTs[40], and the Julich Brain parcellations were transformed onto the donor brain. We selected 29 regions that were defined both in the manual 2D annotations of the autoradiographs and on the Julich Brain atlas. The regions included the visual, primary sensory, motor, cingulum, parietal, and orbitofrontal regions (See Supplementary Information 5.3 for the list of regions). The receptor densities in the 2D annotations were measured in previous work, while the JuBrain regions were used to measure receptor densities in the 3D reconstructed volumes. To assess the accuracy of the reconstruction, we performed a regression analysis of the 2D regional receptor densities onto the corresponding receptor densities measured in the 3D reconstructed volumes using the JuBrain atlas.

### Validation: surface-based estimation of missing pixel intensities

For each type of binding site, respectively, 10,000 vertices that intersected acquired autoradiographs were selected at random. For each of these "seed" vertices, neighbors were identified within $n$ steps along the surface mesh within the same plane as the acquired autoradiograph (all vertices in Fig. 10), where $n$ follows a uniform probability distribution $n \sim U(2,6)$. For the purpose of validation, the vertices from the seed vertex to the n-1 neighbor (purple, green, and blue vertices in Fig. 10) were treated as though the pixel intensities at these locations were missing, though in fact all pixel intensities were known because the vertices all intersect an acquired autoradiograph. A subset of $m$ neighbors around the seed vertex, where $m \sim U(1,5)$, were then identified (purple and green vertices in Fig. 10). These form a core patch of vertices whose average pixel intensity was estimated.

The vertices that were $n$ edges away from the seed vertex (red vertices in Fig. 10) were considered to have known pixel intensities. The surface-based linear interpolation algorithm was then used to estimate pixel intensities for vertices within the seed and core patch of vertices given the vertices with "known" intensities. Pixel intensities were estimated for each vertex individually and then averaged together.

In the human data, the average sampling distance between sections of the same type was ~570 µm. The value of $n$ was chosen such that the interpolated vertices were 0.05–1.2 mm away from vertices with known pixel intensities.

### Validation: Interpolation accuracy

To ensure that the reconstruction pipeline did not affect the measured pixel intensities, regions in the unprocessed autoradiographs

were compared to the corresponding regions in the reconstructed volumes. Regions of interest (ROIs) were generated based on similar patterns of pixel intensity distribution, with a mean area of $21 \pm 11$ mm$^2$ (see Supplementary Fig. 4 for illustration). These ROI were not intended to be biologically meaningful per se, but rather to reflect the spatial characteristics of real parcellation schemes, e.g., ROI from atlases such as the Julich Brain[4], which we anticipate will be used to analyze the reconstructed data.

The parcellation of ROIs was produced using K-Means clustering with the "slic" function from Skimage[54]. The autoradiograph parcellations were transformed using the same 2D and 3D transformations as were applied to their corresponding autoradiographs during reconstruction (see Supplementary Fig. 4 for illustration). The accuracy of the transformed pixel intensities was quantified by calculating the absolute error between true and interpolated values, divided by the true intensities.

### Validation: reconstructed values versus in vivo PET

To assess the validity of the reconstructed GABA$_A$ -benzodiazepine receptor volume, we compared its regional receptor density measurements to those obtained from [18-F]-flumazenil PET scans acquired from nine healthy controls. We quantified the similarity between the reconstructed receptor volume and WM normalized SUVR images derived from the PET scan and, for comparison, measured the similarity among SUVR images themselves. SUVR images were used because they partially control for the amount of non-specific ligand binding versus specific binding to the receptor. This approach tested whether reconstructed receptor volumes exhibit a level of similarity to PET-derived receptor distributions comparable to the similarity observed among PET scans themselves.

Both the benzodiazepine receptor autoradiographs and PET scans were acquired using the [18-F]-flumazenil ligand, making them ideal for direct comparison. Regional receptor densities were extracted from the reconstructed flumazenil volume using the Julich Brain Atlas[4].

Each PET scan was linearly aligned to the corresponding participant's MRI with a rigid transformation. The MRI scans of the healthy controls and the brain donor were then non-linearly aligned to the MNI152 (2009c) template[55] using ANTs. PET scans and the reconstructed benzodiazepine receptor volumes were subsequently transformed into MNI152 space, and regional receptor densities were extracted using the Julich Brain Atlas[4].

To quantify similarity, we calculated the Spearman's $\rho$ correlation between regional receptor densities from the reconstructed and PET-derived volumes. To evaluate whether the reconstructed receptor volume was as similar to PET-derived receptor distributions as PET scans were to one another, we compared the mean Spearman's $\rho$ from reconstructed vs. PET comparisons to the mean Spearman's $\rho$ from PET vs. PET comparisons.

### Reporting summary

Further information on research design is available in the Nature Portfolio Reporting Summary linked to this article.

### Data availability

Derived metrics used for validation of Brainbuilder are publicly available (https://doi.org/10.6084/m9.figshare.28934441)[56].

### Code availability

The code used in this manuscript is available at https://github.com/tfunck/brainbuilder.

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

## Acknowledgements

The present work is supported by the following funding sources. European Union's Horizon 2020 - 945539 (Human Brain Project SGA3): Katrin Amunts, Nicola Palomero-Gallagher. Federal Ministry of Education and Research (BMBF) - 01GQ1902: Nicola Palomero-Gallagher. Helmholtz Association's Initiative and Networking Fund - InterLabs-0015 : Katrin Amunts, Alan Evans. National Institute of Health - RF1MH128696 : Ting Xu. National Institute of Health - R01MH139349 : Ting Xu. Open Access publication costs are funded by the Deutsche Forschungsgemeinschaft (DFG, German Research Foundation) – 491111487. The funders had no role in study design, data collection and interpretation, or the decision to submit the work for publication.

## Author contributions

T.F.: writing, methodology, validation, formal analysis, software, and conceptualization. K.W.: writing, methodology, software, and conceptualization. C.L.: writing and methodology. M.O.: methodology. P.-J.T.: writing and methodology. T.X.: funding acquisition, writing, an methodology. K.A.: writing and conceptualization. A.T.: funding acquisition, supervision, writing, resources, data curation, and conceptualization. N.P.-G.: funding acquisition, supervision, writing, resource, data curation, methodology, and conceptualization. A.C.E.: supervision, writing, resources, and conceptualization.

## Funding

## Competing interests

The authors all declare no competing interests.
