## [Transparent Peer Review file · Communications Biology]

BrainBuilder: A Software Pipeline for 3D Reconstruction of Cortical Maps from Multi-modal 2D Data Sets

Corresponding Author: Dr Thomas Funck

This manuscript has been previously submitted to another journal. This document only contains information relating to versions considered at Communications Biology.

Version 0:

Reviewer comments:

Reviewer #1

(Remarks to the Author)

Reviewer #2

(Remarks to the Author)

The paper describes a new open-source software repository (BrainBuilder), which is applied to two separate types of datasets: one with autoradiographs and whole-brain MRI data from a human subject, and another from a macaque, but the MRI is from a template. The authors describe a convincing strategy for reconstructing sparsely and alternately sampled 2D sections into a 3D volume that bears strict anatomical correspondence with the MRI data. Furthermore, the authors claim that their method is generalisable to other datasets besides their own.

In consideration for publication in Nature Communications Biology, I was asked to review the manuscript as part of a transparent peer review on the basis of the following criteria:

- The results are novel: yes
- The paper provides strong evidence for its conclusions: further evidence required
- The data are technically sound: yes
- The manuscript is important to scientists in the specific sub-field of biology: yes

With respect to the above, I have included detailed commentary in the attached PDF file, and hereby I just would like to reiterate my major critical observations:

- 1) The 3D receptor density volumes, which are considered the final output of the BrainBuilder pipeline, seem to be a result of four consecutive interpolations, which raises a significant concern about the integrity of the results. A quantitative comparison with the reconstructed volumetric autoradiograph data is suggested, and based on the results, the modification of the pipeline may be necessary.
- 2) The authors' claim of the registration accuracy between the autoradiographs and the MRI data are seemingly substantiated by Dice score calculations, however, the evaluation does not take segmentation errors into account, which could have propagated into the registration as well. Therefore, at the very least, representative visualisations of the registered autoradiographs with the white matter surface overlay should be provided.
- 3) The description of the methods at several points are incomplete. For example, too little information is given about the neural-network-based segmentation of cortical ribbons in autoradiographs, despite this being mentioned as one of the key technical novelties of the pipeline.

Based on the above, I would recommend this manuscript for major revisions at this time.

Version 1:

Reviewer comments:

Reviewer #1

(Remarks to the Author)
See attached document

Reviewer #2

(Remarks to the Author)

Upon reviewing the authors' response to my previous comments, as well as my fellow reviewer's comments, I found that the authors provided accurate and detailed justifications for previously unsubstantiated claims in their manuscript. All requested experiments have been conducted with sufficient scientific rigor, and limitations (where applicable) were adequately stated. The manuscript in its revised form adheres to a commendable standard, therefore I recommend it for publishing without further need for modification.

Should the authors have further questions about specific points that were raised in the review process, I would encourage them to contact me (Istvan N Huszar) directly.

Version 2:

Reviewer comments:

Reviewer #1

(Remarks to the Author)

The authors have thoroughly addressed all my comments in their rebuttal letter. They have strengthened the manuscript with additional validation using in vivo PET data, clarified methodological choices, and added discussions on scalability, error propagation, and applicability to non-cortical structures. The revisions improve both the scientific rigor and clarity of the work. I find the manuscript suitable for publication in its current form.

Reviewer 1

General Comments

Comment #1.1. *One notable concern revolves around the assumption underlying BrainBuilder's methodology. The pipeline relies heavily on structural reference volumes and intricate alignment processes, implicitly assuming that these references accurately capture the complex anatomy of the brain. This assumption may introduce biases or inaccuracies, particularly if the reference volumes do not adequately represent the variability across different brains or species.*

Reply: We agree that the use of a stereotaxic template does indeed make important assumptions about the correspondence between the donor brain and the template. This is why in the present work the human reconstruction was conducted with T1w MRI from the same donor brain as was used to acquire the 2D sections. A stereotaxic template was only used for the macaque reconstruction because no MRI was acquired in this case. The macaque has a much simpler cortical morphology than the human with more regular folding patterns, and hence is easier to align versus a standard template.

Comment #1.2. *Furthermore, the interpolation of missing sections within the pipeline raises concerns about the accuracy of reconstructed volumes, particularly in regions where sharp boundaries between biological features exist. The linear interpolation method used between vertices may oversimplify the complexity of tissue morphology and potentially obscure crucial distinctions between different receptor architectonic areas, which may lead to inaccuracies in the reconstructed volumes.*

Reply: This is a very valid criticism. Unfortunately it is impossible to overcome with sparsely sampled data because the ground truth as to the locations of sharp boundaries is unknown (beyond those already identified on the 2D autoradiographs). However, if users are concerned about the validity of the interpolation they can limit their analysis to the volume of acquired 2D sections transformed onto the reference volume without any surface-based interpolation.

Comment #1.3. *Additionally, missing sections are referenced multiple times, but how much are we talking about (1%, 10%, or more?)*

Reply: In the human data approximately 37% of the brain was sampled and 24% in the macaque. This has been clarified along with the spacing in Section 1.2 par. 7 as follows.

To illustrate the sparsity of the data used here, only approximately 37% of the donor brain was sampled with complete sections with an average distance of 1.03mm and a maximum distance of 1.42mm between sections measuring the same biological feature. Sections were either lost due to mechanical processing errors or excluded because the quality of the acquisition was

poor (e.g., binding artefacts or blurred histological staining) or because the sections contained only a fragment of brain tissue, frequently as a result of slicing the sections close to the border of the tissue slabs. In the macaque the sections covered approximately 24% of the brain with a sampling of 1.32mm between acquired sections for the same biological feature. The macaque brain was deliberately sampled sparsely in repeats of sections along the coronal axis to allow sampling through the brain while limiting the cost of data acquisition.

Comment #1.4. *Another area of concern lies in the segmentation of cortical gray matter from 2D sections, which forms a critical step in the reconstruction process. While BrainBuilder employs neural networks for segmentation, the effectiveness of these models may vary depending on the imaging modality and tissue contrasts involved. The reliance on accurate segmentation raises questions about the pipeline's robustness and generalizability across different datasets and experimental conditions.*

Reply: To address concerns about the quality of the segmentation we have added a more detailed description of methods (section 4.2.3.1 par. (1-2); 4.3.1 par. 1) and results (section 2.3 par. 1) for neural network segmentation. In our validation tests, the network achieves high dice scores when tested on real 2D brain images from human (Dice Score = ~0.95) and macaque (Dice Score = ~0.93). If for some reason the neural network fails on a given section, Otsu thresholding is used as a backup so that the whole reconstruction does not fail. Otsu thresholding yielded a Dice score of ~0.8-0.85 on the manual labels. We have added the following section in the revised manuscript.

2.3.1 U-Net GM segmentation

(1) A U-Net³⁸ was trained to segment the GM in 2D sections using exclusively synthetic 2D images generated from a segmentation of the BigBrain. The segmentation accuracy of the U-Net was evaluated by calculating the Dice score³⁹ between GM segmentation predicted by the U-Net from real autoradiographs versus manually drawn GM labels for these sections. For comparison, the Dice scores were also calculated for segmentation produced with Otsu^{39,40} thresholding. The results showed an average Dice score of 0.93 ± 0.04 for 40 human sections and 0.91 ± 0.03 for 20 macaque sections, indicating that the network generalized reliably from human to macaque brain sections. The Dice score obtained with Otsu thresholding was only 0.85 ± 0.08 in the human sections and 0.8 ± 0.14 in macaque sections, showing a significantly more accurate and precise GM segmentation with the U-Net trained on synthetic data.

4.2.3.1 U-Net 2D GM Segmentation

(1) Binary GM sections are generated from the sections using a deep neural network. A multi-tissue segmentation of BigBrain at 200 micron isotropic resolution was used to generate 10,000 2D training images (Figure 10). To better synthesise intact *in vivo* MRIs, a skull segmentation of the MNI152 atlas was coregistered to the BigBrain and used in 50% of training examples. To generate synthetic training examples, random affine transformations (with random

scaling \sim uniform(0.9, 1.1) and rotations \sim uniform(0, 15°) were first used to create randomly transformed versions of the segmented BigBrain volume. From each of these transformed volumes, 500 2D sections were extracted and used as a training example. To synthesise heterogeneous laminar structure, the cortex was segmented into a random number of cortical layers between 0 and 11, with random proportional thicknesses⁴⁸. The considered tissue classes included: WM, subcortical GM, cortical GM layers, cerebellar GM, within- and outside- brain background, and skull. For each training example, randomly generated values were assigned to each tissue class in the 2D section from Gaussian distributions. To introduce additional variation in the synthetic images, random smoothing, noise, filtering, scaling, and cropping were applied to training examples (see *Supplementary Information 5.1*).

(2) The nnUNet (v.1) software package⁴⁹ was used to automatically identify optimal hyperparameters and train a U-Net³⁸. The U-Net was trained to segment synthetic sections into cortex, background and white matter, with a second task to identify the pixels at the boundaries between these image classes.

Figure 10. Example of generated synthetic images for training the segmentation U-Net. The left column shows examples of the synthetic images with different random values assigned to cortical layers, WM, cerebrospinal fluid, skull, and background. As can be seen in the synthetic images on the left, data augmentation was used to add artefacts to the synthetic images, including: a) affine transformation of the BigBrain volume, b) gaussian smoothing and noise, c) zoom factors, and d) occluded rectangular regions. Two sets of labels are used in the training: a) GM and WM regions (middle column), b) WM-GM and GM-pial surface boundaries (right column).

4.3.1 U-NET Cortex Segmentation

(1) The prediction accuracy of the U-Net was validated using manually segmented GM images from randomly selected autoradiographs from the human data (n=39) and macaque data (n=10). The manual GM segmentations were drawn on the raw images using the GNU Image Processing (GIMP) software. The raw and segmented images were downsampled to images of 415 x 558 pixels to reflect the image size used to train the network. For comparison, segmented GM images were also created using Otsu thresholding⁴⁰. The Dice score³⁹ between the predicted GM, both using the U-Net and Otsu thresholding, and the manually segmented GM was used to validate the accuracy of the segmentation.

Comment #1.5. *While reviewing the validity of the BrainBuilder pipeline, I couldn't help but ponder the absence of comparison with positron emission tomography (PET), another method used to visualize neuroreceptors, albeit with lower resolution. Surprisingly, PET was only briefly mentioned in the introduction and completely overlooked in the discussion. Would it not have been appropriate for the authors to compare their reconstructed brain atlas with a PET scanner atlas depicting the same neuroreceptors? PET imaging is widely recognized for its ability to visualize and quantify the distribution of specific neuroreceptors in the brain. By juxtaposing the results obtained from BrainBuilder's reconstructed 3D brain atlas with those from a PET scanner atlas for the same neuroreceptors. Such a comparison could give valuable insights into the consistency and reliability of BrainBuilder's reconstructions in capturing the spatial distribution of neuroreceptors in the brain. Integrating PET data into the evaluation process would undoubtedly enrich the discussion surrounding the validity and applicability of the BrainBuilder pipeline.*

Reply: Thank you for your excellent suggestion. As someone with a keen interest in PET imaging, we share your enthusiasm for exploring the relationship between autoradiography and PET. In fact, the original motivation for this project stemmed from our desire to create a ground truth for Monte-Carlo PET simulations using 3D autoradiography data. We have conducted some preliminary comparisons between our 3D flumazenil volume and flumazenil PET images, but the results ($r^2 \approx 0.73$) were somewhat limited in value. The necessary smoothing to compare the autoradiography data with the PET atlas resolution combined with inter-subject variability and inherent differences between the modalities raises questions about the source of any observed differences.

While we fully agree that a comparison with PET would provide valuable insights and intend to explore this avenue in future studies, we feel that such a comparison falls outside the specific aims of the present manuscript, i.e., the spatial alignment of the sections. However, this is indeed an area of ongoing investigation for us.

We greatly appreciate your suggestion and look forward to incorporating this perspective in subsequent work.

Comment #1.6. *I am pretty old school when I read an article. I follow the classic sequence of reading an article: introduction, materials and methods, results, discussion, and conclusion. Hence, when I initially read the article, I was under the impression that it pertained to only one human brain and one macaque brain. The materials and methods section merely referenced "human" or "macaque" brain without explicitly indicating multiple specimens. Therefore, upon reaching the conclusion, I found myself perplexed by the sudden mention of 3 humans and 4 hemispheres from macaque brains. Additionally, I am curious about the images showing the neuroreceptors of the brains featured in the article. Are these images derived from a single brain or are they a combination representing an average across the multiple brains examined? This ambiguity further fueled my confusion and left me seeking clarification regarding the scope and context of the data presented.*

Reply: We apologize for any confusion caused by the ambiguity in the Materials and Methods section. We have now added clarification in Section 4.1, paragraph 1, to explicitly describe the data used in this work as well as the data available to us. For this proof-of-principle demonstration of BrainBuilder, we focused on a single hemisphere from both a human and a macaque brain. We have added the text in the revision as follows.

The present work uses two post-mortem brains, one acquired from a human and the second from an adult male *Macaca fascicularis*... While these brains are part of a larger data set consisting of 3 complete human brains and 4 macaque brain hemispheres, only a single human and macaque hemisphere was used for proof-of-principle of the proposed reconstruction method.

Comment #1.7. *Deformed tissue presents challenges for alignment with structural reference volumes from MRI due to potential skewing and compression. Moreover, tissue placed on glass slides may not lay completely flat, further contributing to deformation. My concern arises from the statement in the article that the alignment process involves rigid 2D inter-section alignment. Rigid alignment typically involves translation and rotation without scaling or shearing the image. However, without scaling or shearing, achieving a Dice score of 91% seems impossible. It's essential to clarify that rigid alignment alone may not adequately address the complexity of tissue deformation.*

Reply: We're very sorry for the confusion and grateful for the opportunity to clarify that BrainBuilder does indeed make heavy use of non-linear alignment (both 2D and 3D). It is only the initial alignment that is rigid because at this stage BrainBuilder only needs a very rough and low resolution 3D reconstruction to begin the multi-resolution alignment process. We have clarified the text to make this clearer (e.g., Figure 3, Methods 4.2.3.2 par. 3)

Comment #1.8. *To address these concerns, visual evidence of the aligned images should be provided in the supplementary materials. This would allow readers to assess the effectiveness of the alignment process and evaluate the extent of deformation correction achieved by the*

pipeline. It would also be beneficial to talk about deformation and alignment in the discussion section as well.

Reply: In response to Reviewer 2 we have added examples of the best and worst aligned sections in Figure 6 as follows.

Figure 6: A) Dice scores for two different validations are shown. Reference: Dice score between aligned sections versus corresponding coronal section from reference volume, Inter-section: Average Dice score between aligned section versus neighbors in posterior and anterior direction along coronal axis. Overall Dice score were high 0.89-0.95 but with some sections that appear to be poorly aligned. B) Examples of sections that were either very poorly (left) or well (right) aligned. The grey images indicate the non-linearly aligned 2D segmentations and the hot (orange-red) lines are the edges of the target 2D section from the structural reference volume. The poorly aligned sections show that the source of misalignment is frequently 2D sections that are either very small (Slab 1,2,6), poorly segmented (Slab 4), or incomplete sections (Slab 2,3). The sections that have been well aligned consistently demonstrate a Dice score of 0.97, indicating an upper ceiling for BrainBuilder.

Comment #1.9. *I suggest that the BrainBuilder pipeline could greatly benefit from the creation of a comprehensive tutorial accessible through Python notebooks or a similar platform. A tutorial like this would walk people through the process of implementing the BrainBuilder pipeline in their own research projects. Furthermore, if the dataset used in the study is public available, it is preferable to store it on a platform such as Zenodo and assign a DOI that can be linked directly to the manuscript. To familiarize users with the features and capabilities of the pipeline, a demo dataset should be provided if the data cannot be made public. Ultimately, the primary goal is to encourage people to use the BrainBuilder tool, and tutorials and easily accessible datasets play a pivotal role in achieving this goal.*

Reply: This is an excellent suggestion and we have added a tutorial that illustrates how to use BrainBuilder using a synthetic dataset that is freely available to use.

Comment #1.10. *All things considered, BrainBuilder is a very interesting concept, but require more explanations and clarifications. Addressing these concerns could enhance the utility and reliability of the pipeline while also improving our understanding of brain architecture and function.*

Specifics

Illustrations

Comment #1.11. Please make sure all illustrations use the same font size, and that values on image axes are large enough to be readable. Figure 1C, why not just zoom into the individual brain section instead of including part of the nearby section?

Reply: We have updated Figure 1 to only show the cropped version of the images as follows.

Figure 1. A) Anterior and posterior view of donor brains. The brains were not chemically fixed and hence exhibited severe deformations when compared to Freesurfer⁸ cortical surface representations derived from the donor MRI. B) Tissue slabs (~1.7-2.9cm in thickness) from the human brains were not cut perfectly parallel to one another (see the lines for an example of cuts) and, as illustrated in the depicted slab, each slab of fresh tissue was subject to its own deformation prior to shock freezing. Sections sliced from ends of slabs frequently contained small amounts of brain tissue when sliced by the flat interface of the microtome and were lost or discarded, creating gaps between the slabs. C) Exemplary autoradiographs from the human

data set showing each of the 20 neurotransmitter binding sites, and illustrating the substantial heterogeneity in pixel intensities between the autoradiographs.

Comment #1.12. Move Figure 3 into supplementary

Reply: Figure 3 has been moved to the Supplementary Material 5.1.

Comment #1.13. In Figure 6, the image is visually beautiful. However, accurately assessing the correctness of the receptor map presents a challenge, as the corresponding sagittal section's appearance of the GM is unknown. To address this, I suggest including the MRI image in the supplementary without the receptor map overlay.

Reply: We have updated Figure 4 to more clearly show the aligned sections over the cortical GM of MRI image from a sagittal and axial perspective as follows.

Human

Sagittal

Slab 6

Slab 5

Slab 4

Slab 3

Slab 2

Slab 1

Axial

Macaque

Sagittal

Axial

Figure 4: 2D coronal sections were aligned to 3D reference volumes are are shown along exemplary sagittal and axial cut for the human and macaque reconstructions, respectively. Coronal sections are almost all correctly aligned to the cortical surfaces. Some less well aligned regions are visible in the dorsal portion of slab 4 and in slab 2 in the human, indicating that additional refinements may further improve the alignment.

Comment #1.14. Text in Figure 6 Axial, sagittal, and coronal sections showing receptor binding sites for a A) → Isn't it only sagittal sections?

Reply: Yes, that is correct. The text has been corrected in Figure 5.

Comment #1.15. Table 1. Make it horizontal instead of vertical.

Reply: The table has been reformatted and additional validation metrics added as per Reviewer 2's recommendation as follows.

Slab	Human					Macaque	
	1 (Anterior Pole)	2	3	4	5	6 (Posterior Pole)	Whole-brain
Inter-section Dice Score	0.94 ± 0.04	0.93 ± 0.05	0.89 ± 0.07	0.92 ± 0.04	0.93 ± 0.06	0.94 ± 0.06	0.95 ± 0.6
Section to Reference Dice Score	0.96 ± 0.06	0.94 ± 0.03	0.93 ± 0.04	0.93 ± 0.04	0.95 ± 0.05	0.95 ± 0.02	0.93 ± 0.08

Table 1: The accuracy of the alignment was measured using two different Dice scores. The Dice score of the non-linearly aligned section versus the corresponding reference section from the structural volume ("Reference") and the Dice score between a given section and its immediate neighbors ("Inter-section"). Both Dice scores were consistently high across all tissue slabs and in both human and macaque reconstructions. In the multi-slab human reconstruction, the middle slabs had slightly lower Dice scores (0.89-0.92) versus at the poles. These lower scores likely reflecting both the added difficulty of identifying the anterior and posterior bounds of these slabs on the reference volume and the more varied set of morphological features in these middle slabs (e.g., basal ganglia and temporal lobe)

Comment #1.16. Figure 7. I think the standard deviations of the different slabs should be greater than 0.03-0.07, as indicated in table 1. Perhaps it is right, however the variation in the data appears to be more widespread than that.

Reply: We have updated Figure 7. We've used a low alpha level for the scatter points to better illustrate that most points are indeed well aligned with a few low outliers, and hence that most points do fall within 3-8% of the mean.

Comment #1.17. In Figure 8, the True Pixel Intensity value appears somewhat ambiguous. As far as I could discern, it seems to be an 8-bit image, which would typically limit the values to a range of 0 to 255. However, if it were a 16-bit image, the values could extend beyond this range. Yet, the precise definition of Pixel Intensity in this context is not clearly specified unless I have missed that information.

Reply: The raw autoradiographs are indeed 8-bit images. However raw autoradiographs only encode the radioactivity concentration and must be converted to receptor binding densities (fmol/mg protein). The receptor binding densities are stored as 32bit floating point values.

Introduction

Comment #1.18. Page 2: “Mesoscale atlases(1–5) play a pivotal role in elucidating the intricate architecture of the brain..” → The link is not working for the different mesoscale atlases.

Reply: The link is only used internally by the Google Docs Paperpile application within the document to keep the reference numbers organized. External links to the manuscripts should be added upon eventual publication.

Comment #1.19. Page 2: “This is because numerous individual challenges involved in the reconstruction of 2D autoradiography are, respectively, frequently encountered in other post-mortem 2D imaging data sets.” →The introduction might benefit from specifying some of the specific challenges involved in the reconstruction of 2D autoradiography to provide more clarity to the reader. While it mentions that there are numerous individual challenges, it doesn't elaborate on what these challenges are.

Reply: A sentence enumerating the main challenges involved in the reconstruction has been added in Section 1.1 par. 4 as follows.

These include sparse sampling of sections, severe non-linear deformations of the brain prior to sectioning, and multi-modal serial data acquisition.

Discussion

Comment #1.20. Page 18: “ Our approach therefore only assumes that the GM can be accurately segmented within the sections and does not assume an intensity-based relationship between the acquired sections and the structural reference volume.”

Do you occasionally rely on Otsu thresholding? If so, it raises concerns about the accuracy of GM segmentation, as the algorithm may overlook certain sections. Have you quantified the percentage of such occurrences? This could potentially impact the alignment between sections and, consequently, the Dice score.

Reply: In the previous submission there was a problem in the implementation of the U-Net network that caused it to fail on the macaque data. This has been fixed and in the current resubmission Otsu thresholding is not used. It's included in BrainBuilder as a backup so that the whole alignment process doesn't halt in case a single section is not segmented.

Conclusion (Page 19)

Comment #1.21. *The results here serve as a proof-of-principle that BrainBuilder can reconstruct images of 2D sections processed for the visualization of receptor autoradiographs, cell bodies or myelin accurately at high resolution in both the human and the macaque brain."*

It's uncertain whether the resolution is sufficient to capture cell bodies effectively, as many cells are smaller than the in-plane spatial resolution of 20 μm for both human and macaque sections, where the inter-section distance is 400 μm (not stated in the manuscript).

Generally, cells in the cortex have diameters below 20 μm , and given that one pixel alone cannot adequately represent a cell, there may be limitations in accurately capturing cell bodies with this resolution.

Reply: Indeed, we did not mean to imply that we could capture cell body intensities as such, but rather that we can reconstruct 2D images whose pixel intensity distribution reflects the density as opposed to pixel intensity distributions reflecting receptor distributions. We've amended the text to make this clearer.

The reconstruction is now at 250 μm resolution, though this only helps to capture laminar distributions, not microscale features like cell bodies. Also the gap between sections has now been added to the introduction (see Comment 1.3)

General comments for conclusion

Comment #1.22. *The conclusion mentions that the data are from 3 human brains and 4 hemispheres from macaque brains. Surprisingly, this information is not mentioned elsewhere in the article, which can be confusing for readers. Typically, one would expect such details to be clearly outlined in the Materials and Methods section for easy reference.*

Reply: See Comment 1.6.

Comment #1.23. *Another aspect that raised questions was the section thickness of the cryosections, which appears to be around 20 μm but is not explicitly stated in the article beside in the conclusion. This omission leaves readers guessing about the specifics of the experimental setup again.*

Reply: The 20 μm section thickness is stated in section 4.1 (line 525), but has also been made explicit in the introduction, Section 1.4 par. 2 as follows.

The data are acquired for both species and at high spatial resolution: ~50 μm in-plane, ~20 μm through-plane full-width at half maximum (FWHM) resolution.

Comment #1.24. *This is not stated in the manuscript, but I assume that the spacing between each slide of a particular receptor would be roughly 400 μm (20 * 20 μm). Moreover, completing*

the entire slab (2-3) would require at least 1000 slides (20,000 μ m/20 μ m), indicating a significant investment of time and resources for any laboratory looking to replicate this method. This lack of clarity regarding section thickness and slide count could hinder the reproducibility and scalability of the technique.

Reply: Apologies for the lack of clarity. We detailed these information in our revision. See Comment 1.4 for information on section spacing. Additionally, we have added information about the number of acquired sections in Section 4.2 par. 3 as follows.

Approximately 1402 sections were acquired for the macaque hemisphere and 3142 sections in the human hemisphere.

Materials and method

Comment #1.25. *“ Binary GM sections are generated from the sections using a deep neural network.”* → *What deep learning architecture are you using? Are there any references?*

Reply: We used a U-Net model in the present study. Please see Comment #1.4. for more details.

Comment #1.26. *Page 21: “ In cases where the network fails to provide a segmentation, e.g. returns an empty image, a segmentation method based on Otsu(44) histogram thresholding. In practice, the human sections were segmented using the neural network and the macaque sections were segmented using the histogram thresholding approach.”* → *If the neural network cannot threshold the GM, how good a job would Otsu do? maybe show some examples in supplementary of these situations?*

Reply: We compared the Dice score of the U-Net segmentations versus Otsu on manual segmentations and found that Otsu thresholding results in a score of $\sim 0.85 \pm 0.08$ in humans. So overall Otsu works fairly well but is less reliable than the U-Net segmentation. See Comment #1.4 for a more detailed description.

Comment #1.27. *“ We used the nearest neighbour interpolation between aligned 2D GM sections to estimate the morphology of the cortex where no sections could be acquired.”* → *What is the reason for using nearest neighbour? It is because these are the binary images of 0-1? If so it makes sense.*

Reply: We have updated this portion of the reconstruction and now use linear interpolation between GM segmentations. The values are between 0 and 1, but are not binary. The original binary segmentation is done on the raw images at $\sim 20\mu$ m, but then are downsampled to the resolution of the reconstruction and hence do not have binary values.

Comment #1.26. “ The missing pixel intensities for each cortical surface mesh are interpolated by applying linear interpolation over each of the corresponding upsampled inflated spherical meshes.”→

Why choose linear interpolation and not cubic? Would be good to hear if they have tried cubic and show in supplementary the pros and cons of the different interpolation methods. Could also add some text about this in the discussion section.

Reply: This is a great question. We did not use cubic interpolation because we wanted to avoid the possibility of over- or under-shoot of the interpolated values. Linear interpolation guarantees that interpolated values are strictly within the range of the actually observed values.

We have added the following lines to section 3.3 par. 2 to clarify our reasoning and acknowledge that more work is required to find the optimal interpolation scheme.

Linear interpolation was chosen because it will only produce values within the range of those observed in the acquired sections and because it provides easily interpretable results. However, this approach...

More work is required to establish which interpolation scheme, e.g., linear, spline, AI-based in-painting, gaussian process regression, produces the most reliable estimation of missing pixel intensities.

Reviewer 2

Page 2

Comment #2.2.1. Cellular architecture, maybe? A quick Google search aligns with my natural interpretation of mesoscale: "length scales ranging from that of an individual cell, down to the size of the molecular machines". Based on this, it is not the "molecular organisation of the brain" that I would emphasise here. Otherwise, e.g., if you refer to neurotransmitters, please clarify what you mean by "molecular".

Reply: We've switched to the terms “chemo-” and “cytoarchitectonic” because they mean essentially the same thing but hopefully avoid any confusion.

Mesoscale maps of brain architecture are important tools for characterizing the chemo- and cytoarchitectonic organization of the brain.

Comment #2.2.2:

Reply: See Comment 2.2.3.

Comment #2.2.3: British-American spelling inconsistency: previously you established British spelling with "organisation".

Reply: We've changed to American spelling.

Comment #2.2.4. Please be more specific about the methodology, if you are talking about 2D sections.

Reply: While the method we developed aims to be modality agnostic, we've now specified the specific modalities used in this study in Section 1.1 par. 1

Post-mortem brain imaging of 2-dimensional (2D) brain sections, using modalities such as autoradiography and histology,

Comment #2.2.5. Confusing grammar: are these examples of mesoscale structures, or examples by which 2D imaging can surpass mesoscopic visualisation?

Reply: We've changed the end of the sentence in Section 1.1 par. 1 to highlight that 2D post-mortem imaging can visualize mesoscale structures. In fact these methods can in principle go down to the microscale, but we removed reference to "at or exceeding" the mesoscale as this may have caused unnecessary confusion.

Post-mortem brain imaging of 2-dimensional (2D) brain sections, using modalities such as autoradiography and histology, enables the resolution of features at the scale of "mesoscale" brain structures such as architectonic cortical layers, cortical columns, and sub-nuclei.

Page 3

Comment 2.3.1. *Missing definition. How do the authors differentiate a mesoscale atlas from a "macroscopic" one? Is it about the cortical laminar resolution, which is mentioned later in this paragraph, or there are other important differences? Where would a standard T1-weighted 1mm isotropic MRI fall in this binary classification?*

Reply: We've added a definition of macroscopic to Section 1.1 par. 3

Existing mesoscale atlases have focused on mapping cell density or myelination. Though macroscopic atlases capable of resolving cortical areas on the scale $\geq 2.4\text{mm}$ have been created for neurotransmitter receptor distributions

Comment 2.3.2. *Even an MRI scan shows "regional differences". As long as you don't define the scale, this is meaningless.*

Reply: We replaced the sentence to try and make clearer the limitations of existing macroscopic receptor atlases. Section 1.1 par. 3.

These macroscopic receptor atlases suffer from partial-volume effects that limit their quantitative accuracy and cannot resolve laminar receptor distribution patterns.

Comment 2.3.3. *It would be great if you could mention a few examples here, so that readers could get an initial idea what kind of problems BrainBuilder can solve.*

Reply: A sentence enumerating the main challenges involved in the reconstruction has been added in Section 1.1 par. 4 as follows.

These include sparse sampling of sections, severe non-linear deformations of the brain prior to sectioning, and multi-modal serial data acquisition.

Comment 2.3.4. *These seem to be unsubstantiated claims at this point, which would warrant an experiment showing reconstructions of more than two datasets of different modalities.*

Reply: The justification for this claim is that BrainBuilder produces consistent results across the 20+ different receptors and cell- and myelin stained sections and in two different species. Many of these distributions are as different from one another as might be encountered from images from completely different imaging modalities, e.g., consider receptors that are mostly represented in the primary sensory areas versus receptors almost exclusive to the basal ganglia.

Moreover, the segmentation neural network used (now more thoroughly described) is trained purely on modality agnostic, synthetic data. Hence the network is as likely to be able to segment autoradiographs as any other modality that has contrast in the cortical GM. To emphasize this for the reader, we have added a short sentence in Section 1.1 par. 4 making this point as follows.

This flexibility is thanks in part to a deep learning network that is trained exclusively on synthetic data to segment the cortical GM.

Comment 2.3.5. *Please be more specific. For example, radiologists can report MRI scans that are corrupted by artefacts, and have more than 5 mm slice spacing, which is perfectly useable if the clinical question is about a macroscopic tumour.*

Reply: We've added text to specify what the 3D reconstruction is needed for. Section 1.2 par. 1 as follows:

Several challenges must be overcome before 2D brain sections can be reconstructed into 3D brain atlases that can be used for automated quantitative analyses across the acquired sections and in conjunction with existing 3D brain templates and annotations.

Comment 2.3.6. *Do you refer to intra- or inter-modality variations? I.e., pixel intensity differences due to inhomogeneous illumination in scanned 2D histology slides, or contrast differences between brightfield and fluorescent microscopy?*

Reply: We have clarified what is meant by “acquisition” in Section 1.2 par. 2 as follows. Specifically we mean for a given modality, the different methods for measuring a given biological feature.

Acquisition is defined as a within-modality method for measuring a given biological feature, e.g., autoradiographs acquired with different ligands measure different receptor binding sites and hence are referred to different acquisitions.

Comment 2.3.7. *This sentence provides no clarification to the use of the word "section". It is naturally understood to be a 2D sample of something. All the sentence says that the something can be anything, without providing examples what it may actually refer to in the context of this work, e.g., histology, immunofluorescence, autoradiograph, MRI slice, blockface photograph, etc.*

In fact, more confusion is created by "biological component", as it is not clear what it refers to: an organ, or part of an organ, or a species, or solid tissue vs blood smear?

Reply: We've clarified the definition of “section” and added a second definition for “2D image”. Section 1.2 par. 2 as follows.

“Section” refers generically to the 2D image produced by slicing the brain across a given plane and using an imaging modality to visualize some biological feature in the sliced tissue, such as cell density or receptor binding site.

Page 4

Comment 2.4.1. *Be more descriptive and specific. For example: anterior and posterior views of the same unfixed post-mortem human brain. A coarse comparison with digital surface reconstructions (where did this come from?) shows major bulk deformations in the axial and coronal planes.*

We clarified Figure 1.A in accordance with the reviewers suggestions and specified the source of the cortical surfaces. The image doesn't allow specific inferences of the most deformed planes and is only meant to highlight that, in contrast to the fixed brains that are normally used in histology, the unfixed brains we used were severely deformed.

Figure 1. A) Anterior and posterior view of donor brains. The brains were not chemically fixed and hence exhibited severe deformations when compared to Freesurfer⁸ cortical surface representations derived from the donor MRI. B) Tissue slabs (~1.7-2.9cm in thickness) from the human brains were not cut perfectly parallel to one another (see the lines for an example of cuts) and, as illustrated in the depicted slab, each slab of fresh tissue was subject to its own deformation prior to shock freezing. Sections sliced from ends of slabs frequently contained small amounts of brain tissue when sliced by the flat interface of the microtome and were lost or discarded, creating gaps between the slabs. C) Exemplary autoradiographs from the human

data set showing each of the 20 neurotransmitter binding sites, and illustrating the substantial heterogeneity in pixel intensities between the autoradiographs.

Comment 2.4.2. *Are the lines on the brain surfaces reconstructed from actual measurements, or they are hand-drawn examples for illustration? What am I supposed to see in the two images on the right? If those are meant to illustrate thickness, a scale bar would be necessary.*

Reply: The lines indicate the position of how the tissue was actually cut based on a comparison with what the brain looked like after the cut was made (the central panel in the image) and with the corresponding frozen slab. The exact thickness of the slabs was not recorded when the data was acquired ~25 years ago, but we have added an approximate size of the tissue chunks (1.7-2.9cm) of the reconstructed data in the figure description.

See Comment 2.4.1 for updated figure.

Comment 2.4.3. *What is in the gaps? Are you referring to fragmented tissue? Or you mean that the interslice distance was large?*

Reply: We have clarified the issues produced from cutting the tissue slabs from the fresh human brain and how these produce gaps between the tissue.

See Comment 2.4.1 for updated figure.

Comment 2.4.4. *Please crop the irrelevant parts of the images. The figure looks very busy, and it is hard to tell, how the 20 images refer to different "sites", because they all seem to be reasonably close sagittal sections.*

Reply: Figure 1 has been updated and the colormap changed to better highlight the variability of receptor density between the different acquisitions.

See Comment 2.4.1 for updated figure.

Page 5

Comment 2.5.1. *I disagree that it's a necessity. If you have a good enough optimisation method, you may directly infer the 3D surface where the section inserts into the original volume. It may be *easier* to approach the reconstruction in the sequence that you describe, but it is not a necessity.*

In fact, it may even lead to further inaccuracies. Consider this example: if one assumes that consecutive 2D sections are parallel, that is only true for the embedded state of the slab. However, the slab might have undergone further deformations during embedding, which are unknown and uncompensated.

Reply: We changed the wording to emphasize that first reconstructing the slabs is the easiest, though not necessarily the only approach. See Section 1.2 par. 4 as highlighted below.

The sections acquired across these tissue slabs cannot be naively concatenated into a single stack that could then be reconstructed into 3D. Instead, **the simplest approach is to first reconstruct the sections to the slabs of tissue from which they were** cut and only then can sections from each slab be combined into a single 3D reconstruction.

Comment 2.5.2. *Please specify the differences between acquisitions.*

Reply: Added a clause in Section 1.2 par. 5 to illustrate the different types of acquisitions used in this work, in line with the above definition of acquisition as highlighted below.

Another important obstacle to reconstruction is the use of multiple different types of image acquisitions, **e.g., different radioligands or histological staining methods,** from different sections within the same brain.

Comment 2.5.3. *Yes, but if the differences arise from accurate measurements, they should be preserved. If they arise from artefacts, they should be named here, and compensated for.*

Reply: Apologies, this comment was not entirely clear to me. The purpose of the sentence is to highlight that creating an aligned image stack from 2D images from images with heterogeneous pixel intensity distributions is not necessarily trivial and that the produced stack will also have heterogeneous pixel intensities. Although not stated in this section, the aligned image stack is difficult to align to a target reference volume because of its variability in pixel intensities (e.g. information theoretic alignment metrics will completely fail in this situation). I would be happy to provide further clarification and amend the text to address any concerns.

Comment 2.5.4. *Does a simple trilinear interpolation not solve this problem? It's a trivial concept in image registration that images are defined on a grid, and treated as continuous functions, where values between grid points are inferred via interpolation. If you mean "irregular" by sparse, please emphasise this.*

This is a great question. Just interpolating between adjacent sections has two problems.

Reply: The significant size of the gap between two acquired sections of the same type means that the morphology of the brain in general, and cortical folding in particular, can vary significantly between these two sections. Trilinear (or higher order) interpolation methods will incompletely address this issue. It is possible to account for this morphological change and indeed we had initially designed an approach that did just this. However, even when accounting for the morphology there is a problem (originally pointed

out to us by D. Van Essen at a poster presentation) that arises from the non-perpendicular sectioning of the cortex and propagation of this error between acquired sections. Figure 1 initially had an image describing this problem, i.e. interpolating the dotted line as the interpolation through euclidean space between the bold sections will mix signals from different cortical lamina. We had removed the figure to streamline the manuscript, but would be happy to add it back in if you think it would be of benefit to the reader. Having said all of this, we have used BrainBuilder to reconstruct sections from a rat brain (not included in the present manuscript) and used linear interpolation between the acquired sections. In this case it works well because the brain morphology is very simple and the sections are much more densely sampled (approx. 50um between adjacent sections).

Comment 2.5.5. *It seems pretty much impossible to me. It's an innate consequence of slicing a brain by 2D planes.*

Reply: This is probably true, but We've hedged because there may be a species with a very simple brain structure where this might be possible.

Page 6

Comment 2.6.1. *Sites are still not well-defined in the paper. Do you mean different molecular targets, i.e., neurotransmitter receptors, which produce different imaging contrast on directly adjacent sections of the brain?*

Reply: We've added a definition of "binding site" earlier on in the text in Section 1.1 par. 2.

A binding site refers to the molecular target to which a ligand binds. Different ligands can bind to the same receptor, e.g., GABA_A, but on different binding sites, e.g., benzodiazepine versus GABA binding sites, and hence map different spatial distributions.

Page 8

Comment 2.8.1. *Do binary GM maps work better than probability maps, e.g. derived from a Gaussian Mixture Model segmentation algorithm? Using binaries is an understandable choice, for the simplicity of their representation and the strong contrast to drive the registration, however, segmentation errors may get amplified in the resultant registration.*

Reply: The use of "binary" here isn't totally accurate and has been replaced by "segmentation" in Section 2 par. 3. The raw 2D images are indeed binarized, but after transformation and resampling into a lower resolution reconstructed volume, they are continuously valued floating point images that in effect look similar to a probabilistic segmentation. The suggestion of using a probabilistic segmentation is a good idea for future work. We revised the description as highlighted below.

The alignment between the reconstructed volume and the reference structural volume is performed using segmented GM volumes derived from each of these data sets, respectively.

Comment 2.8.2. *segmentation volumes*

Reply: Changed text to “GM segmentation volume” in Section 2 par. 3.

The problem of aligning a volume composed of heterogeneous pixel intensities to a reference volume with an entirely different pixel intensity distribution is thus simplified to mono-modal alignment between GM **segmentation** volumes.

Page 9

Comment 2.9.1. *Please correct the typo, which made this link inaccessible. I did not review the code, because I only noticed the availability of the code when I reached the correct link at the very end of the document, at which point it was too late in the review process.*

Reply: Typo is fixed in Section 2.1 par 1.

The code is openly available on github: <https://github.com/tfunck/brainbuilder>.

Comment 2.9.2. *The figure is a great representation of the input data structures, and provides clarity about what types of information are needed for a reconstruction. While such figures would normally belong to a documentation or appendix, I would encourage keeping it in the main text, given the lack of specific information about the reconstruction problem in the introduction.*

I appreciate that this is meant to be an illustration of the input data, but I would recommend adding a narrative description of the data in each table to the figure legend. It would help clarify what "conversion factor" stands for, for example.

A few more critical observations about the generalisability of the current input framework: 1) are the numerical fields agnostic to physical units as long as the inputs are consistent? 2) Do the available direction specifications cover all possible cases, e.g. oblique sectioning? 3) Why are pixel sizes anisotropic, and so variable across the records shown? 4) What if certain sections are flipped or rotated, while being mounted on a glass slide? 5) The file name convention seems to be oddly specific. Is it a convenience choice or is it a necessity? 6) How tolerant is the framework for errors in the input tables? Would the processing halt if there was a typo in the table?

Reply: A narrative description has been added. The figure has been moved to Section 5.1 upon the suggestion of the first reviewer.

1. The numerical fields are used to specify pixel sizes in mm because this is the unit that the Nifti file format uses by default. We added a specification of the unit.
2. Unfortunately at the moment the direction specifications only cover sections sliced more or less along the coronal axis, but would not cover sections cut along the sagittal or axial planes. However, we do have autoradiographs acquired in the cerebellum cut along the axial plane, so a long term goal is to extend BrainBuilder to sagittal and axial planes. In

practice the only limitation here is updating the way brain volumes are created and used within BrainBuilder.

3. Unfortunately the variability in the data is just the way the data were acquired circa 2000.
4. BrainBuilder assumes that the data have been preprocessed by the user to fix orientation issues and such. Our experience is that this sort of preprocessing is very idiosyncratic to the specific data set at hand and is beyond the scope of the reconstruction problem we are trying to solve.
5. The file names provided are just placeholder examples. The user could name them however they choose and have them located anywhere on a filesystem they have accessed to. We added a statement to this effect.
6. The code will automatically check the .csv files and report back any missing data to the users. This will indeed halt execution but we could add an option to override this.

A) BrainBuilder python command

```
from brainbuilder.reconstruct import reconstruct

# launch reconstruction with br
reconstruct(
    'hemisphere_info.csv',
    'slab_info.csv',
    'section_info.csv',
    resolution_list=[4,3,2,1,0.5],
    '/path/to/output/'
)
```

B) User .csv inputs:

Hemispheric information in 'hemisphere_info.csv'

sub	hemisphere	struct_ref_vol	gm_surf	wm_surf
MR1	R	MR1_R_gm_srv.nii.gz	MR1_gray_surface_R_81920.surf.gii	MR1_white_surface_R_81920.surf.gii

Slab-level information in 'slab_info.csv'

sub	hemisphere	chunk	pixel_size_0	pixel_size_1	section_thickness	direction
MR1	R	1	0.027	0.020	0.02	rostral_to_caudal
MR1	R	2	0.038	0.029	0.02	rostral_to_caudal
MR1	R	3	0.040	0.030	0.02	rostral_to_caudal
MR1	R	4	0.039	0.029	0.02	caudal_to_rostral
MR1	R	5	0.039	0.029	0.02	caudal_to_rostral
MR1	R	6	0.030	0.023	0.02	caudal_to_rostral

Section-level information in 'section_info.csv'

raw	acquisition	hemisphere	sub	chunk	sample	conversion_factor
RG#hg#MR1s6#R#ampa#5679#04#L.TIF	ampa	R	MR1	6	1353	29.91
RG#hg#MR1s6#R#rx82#5700#21#L.TIF	rx82	R	MR1	6	467	41.15
RG#hg#MR1s6#R#sr95#5723#09#L.TIF	sr95	R	MR1	6	1091	36.42
RG#hg#MR1s6#R#ly34#5721#01#L.TIF	ly34	R	MR1	6	1491	121.40

Figure 14. A) sample piece of code illustrates the usage of BrainBuilder to perform the reconstruction of the human autoradiograph sections B) The user provides the essential information in simple .csv files. **Hemispheric information:** Each row specifies the information applicable to a single hemisphere, specifically the subject, hemisphere, the reference volume, and the associated cortical surfaces. **Slab information:** the rows specify the pixel sizes (in mm) and the direction of sectioning along the coronal axis. **Section information:** Each row specifies a single 2D image, its acquisition, the hemisphere and tissue chunk to which it belongs, the subject identification. The “sample” is its positioning along the coronal axis. The “conversion_factor” is an optional parameter that scales the images to convert the pixel

intensities to a quantitative value, e.g., conversion of raw autoradiograph pixel intensities to fmol/mg protein.

Comment 2.9.1. *Is this an incorrect reference to Figure 4?*

Reply: Figure 4 is now Figure 3 so the typo is now, fortuitously, correct.

The alignment of sections acquired from a human donor and a macaque brain is shown in Figure 3.

Page 10

Comment 2.10.1. *What are the apparent vertical stripes in the 0.5mm column, top row image? Why are the stripes oblique in the same column, second row? What does the red colour indicate?*

Reply: The lines were from a small bug that caused missing sections in the 3D reconstruction volume. This has been fixed and the figure has been updated. There are still two visible lines through the reconstructed volumes because these sections had some signal in the WM or background that caused a segmentation error. The edges of the reference volume (created with a Sobel filter) are shown overlaid on the 3D segmentation volumes to better show the alignment. The text has been clarified in Figure 3.

Figure 3: A hierarchical multi-resolution scheme was used to align the slab volumes to the reference structural volume. Sagittal images of the first tissue slab for each step of the alignment scheme show progressively finer alignment between the 2D coronal sections and structural reference volume. The edges of reference volume are shown in orange. For each iteration, all 2D GM segmentations are downsampled to the current resolution in the hierarchy and hence only contain morphological information at that resolution. 1) 3D segmentation volumes are created by applying 2D transformations to the segmented 2D images. In the first iteration, 4mm, only rigid transformations are used to create an initial 3D GM segmentation volume, but on subsequent iterations the non-linear transformations calculated from the previous iteration are used to create the 3D segmentation volumes. 2) The reference volume (the outline of which is shown in the second and third row in orange) is then 3D non-linearly aligned to the 3D segmentation volume. 3) The alignment is then refined using 2D non-linear

Comment 2.10.2. *It is said that the 2D alignment takes place after the 3D alignment. Does this mean that one of the following is true?*

Option 1) the warped image data is resliced in the warped space -> in this configuration, multiple original slices' data may contribute to a single new slice, hence the rationale of a 2D alignment of the new slices is not trivially understood, and should be explained.

Option 2) the 2D transformations precede the 3D transformation -> in this configuration, the warp field is optimised first, and the 2D parameters later. However, this makes the calculation of the cost function expensive, because a non-linear transformation must be applied at every optimisation step. What justifies this expense? Furthermore, how does the precomputed 3D warp field hold up after the 2D parameters are changed? Does this not lead to a degradation of the accuracy?

Reply: The previous text was indeed confusing and communicated the opposite of what was intended. In effect what happens in BrainBuilder is that the reference volume is transformed to the space of the aligned 2D images, so each of the original images is aligned on its own to a corresponding coronal section of the aligned reference volume. We've rewritten the entire Figure 3 description to give a clearer overview of the pipeline and illustrative results.

See Comment 2.10.1.

Comment 2.10.3. *I appreciate the simplicity, however this is inexact: is it a rigid, a Procrustes, or an affine transformation in 2D?*

Reply: We clarified that these are non-linear transformations in Figure 3.

See Comment 2.10.1.

Comment 2.10.4. *Were these coronal, or sagittal sections? Figure 5 suggests coronal, but the previous dataset was sagittal. If so, this difference should be pointed out, as it exemplifies the generalisability of the method.*

Reply: All sections were coronal. We've updated the text in Figures 3 and 4 to make this clearer. In principle BrainBuilder should work with sagittal or axial sections. The main challenge, as mentioned above) is finding an elegant solution in Python/Numpy for slicing arrays in a dimension determined at runtime.

See Comment 2.10.1 for Figure 3.

Comment 2.10.5. *Even if this is from previously published material, the fact that the individual was consented for the present use case, as well as minimal demographic details should be shared here as well. Maybe more appropriately in the methods section.*

You are absolutely correct and we only realized that we hadn't included the ethics approval / consent until just after submission. We sent an updated copy to the editor but after it had been sent to you. Please find the requested information below (Methods Section 4.1 par. 1).

The macaque specimen was obtained from Covance (now Labcorp Drug Development), Munster, where they were used as control animals for pharmaceutical studies performed in compliance with legal requirements. All experimental protocols were in accordance with the guidelines of the European laws for the care and use of animals for scientific purposes. The specimen was male, ~6 years old. The human subject had given written consent before death and/or had been included in the body donor program of the Department of Anatomy, University of Dusseldorf, Germany. The human donor was a 78 year old male; who died from non-neurological causes.

Comment 2.10.6. *Same as for the human subject, except for the consent, which should be a reference to the ethical approval of the study which led to the generation of the data that is used in the current paper.*

Reply: See Comment 2.10.5 above.

Page 11

Comment 2.11.1. *Inexact description. Please state what is meant by "almost all" (what percentage out of how many slices?) and "correctly" (what criteria were used to ascertain correct alignment?). Furthermore, please show a representative example of alignment from a success case as well as a failure case as part of this figure.*

Why are there no non-linear distortions in the case of the macaque brain?

Reply: The human sections were shown on the reference volume (and hence were warped from their native space) so that all the sections could be shown together. To keep things uniform and help better visualize the results, We've updated the figure to show the sections in the native coordinate space of each slab. The human sections, like the macaque, are now all parallel. The quantitative assessment of the alignment follows in 2.3 where we also include example images of well and poorly aligned sections. This figure is only meant to give a visual, qualitative overview of the alignment.

Human

Sagittal

Slab 6

Slab 5

Slab 4

Slab 3

Slab 2

Slab 1

Axial

Macaque

Sagittal

Axial

Figure 4: 2D coronal sections were aligned to 3D reference volumes and are shown along exemplary sagittal and axial cut for the human and macaque reconstructions, respectively. Coronal sections are almost all correctly aligned to the cortical surfaces. Some less well aligned regions are visible in the dorsal portion of slab 4 and in slab 2 in the human, indicating that additional refinements may further improve the alignment.

Page 13

Comment 2.13.1. *If the original slices were sagittal, please provide an alternative view (orthogonal to the original slicing plane), where the accuracy of the reconstruction is more obviously visible.*

Since one of the many advantages of autoradiography is that it's quantitative, it would be informative to see a maximum-likelihood projection of these 20 reconstructions, which would show in one volume the most abundant receptor type in each region. This would also be useful to spot potential interpolation artefacts.

Reply: The original sections were all coronal hence the sagittal view is orthogonal to the original plane of sectioning.

The GABA receptors (in particular agonist and benzodiazepine) are present in the highest densities, so a map of the most abundant receptors would just be a map of these. We would be happy to implement the suggested analysis but we would need a bit more information.

Comment 2.13.2. *All sections look sagittal. Is anything missing from the figure, perhaps?*

Reply: Apologies this was a typo, only the sagittal images are represented (Figure 5).

2) Macaque

Figure 5. Reconstructed sagittal sections show the receptor binding sites for a A) human brain and B) a macaque brain. The macaque brain also includes sections indicating cell body and myelin density. Because each acquired section can only be visualized for a single biological feature, e.g., a given receptor binding site, there are necessarily gaps between any two sections for a given acquisition and thus creates missing pixel intensities for that acquisition. The problem of missing pixel intensities is further aggravated by the occasional tissue section that has been lost due to mechanical processing errors. Missing pixel intensities are estimated using surface-based linear interpolation.

Comment 2.13.3. *I assume that missing pixels are the consequence of having irregular sections. However, a short explanation would be great to see about these. Please state how abundant the missing pixels are, and whether they are filled by interpolation alone, or smoothing is applied too, in which case, please specify the smoothing radius for reproducibility. Furthermore: does the interpolation respect cortical layers? Is it possible that a misaligned slice will propagate errors to a larger area, across all cortical layers?*

Reply: We've added text to Figure 5 explaining the source of the missing errors and identified the average (and standard deviation) gap between sections with the same acquisition in the introduction

The surface-based interpolation was designed precisely to respect the cortical layers. Misaligned sections would only propagate error linearly between its neighboring sections in the anterior and posterior directions, so the impact would only be local.

See Comment 2.13.2.

Comment 2.13.4. *Abbreviation was not used in the figure.*

Reply: We removed the abbreviation.

Comment 2.13.5. *Is this a Dice overlap of two binary maps, where the background is 0 and the tissue is 1?*

If so, the Dice score is not sufficient to characterise the accuracy of the alignment. Theoretically, if the algorithm registered every coronal section to an axial one, this metric would still indicate high accuracy, as long as the outer tissue boundaries are matched.

For a more informative quantification of the registration accuracy, please calculate the Dice score between gray matter segmentations, and give representative visual examples where the alignment is deemed "good" vs "subpar", showing the white matter surface mesh as an overlay on the reconstructed sections.

Reply: We agree about the limits of the Dice score approach we used and very much appreciate the suggested approach. We've added a second validation that calculates the Dice

score between adjacent GM segmentations in the anterior and posterior direction and added some visual examples of the best and worst segmentations for each slab. The results are similar to what we already had but it's very reassuring to have additional quantitative validation. We also extended the alignment validation to the macaque reconstruction and also report these values. (Figure 6 & Table 1)

Slab	Human					Macaque	
	1 (Anterior Pole)	2	3	4	5	6 (Posterior Pole)	Whole-brain
Inter-section Dice Score	0.94 ± 0.04	0.93 ± 0.05	0.89 ± 0.07	0.92 ± 0.04	0.93 ± 0.06	0.94 ± 0.06	0.95 ± 0.6
Section to Reference Dice Score	0.96 ± 0.06	0.94 ± 0.03	0.93 ± 0.04	0.93 ± 0.04	0.95 ± 0.05	0.95 ± 0.02	0.93 ± 0.08

Table 1: The accuracy of the alignment was measured using two different Dice scores. The Dice score of the non-linearly aligned section versus the corresponding reference section from the structural volume ("Reference") and the Dice score between a given section and its immediate neighbors ("Inter-section"). Both Dice scores were consistently high across all tissue slabs and in both human and macaque reconstructions. In the multi-slab human reconstruction, the middle slabs had slightly lower Dice scores (0.89-0.92) versus at the poles. These lower scores likely reflecting both the added difficulty of identifying the anterior and posterior bounds of these slabs on the reference volume and the more varied set of morphological features in these middle slabs (e.g., basal ganglia and temporal lobe)

Comment 2.13.6. Consider "human brain reconstruction" (to clarify that it wasn't a reconstruction done by a human).

Reply: Now that we include the macaque in the quantitative validation, we have rewritten Section 2.3.2 par. 1 as follows.

The alignment accuracy of the reconstruction was quantified by using two Dice scores. The first dice score is calculated between aligned 2D sections and their corresponding reference sections from the structural reference volume. The global average accuracy of the alignment was 0.95 ± 0.03 (Table 1) in the human brain and 0.93 ± 0.08 in the macaque.

Comment 2.13.7. Please provide an interpretation for this observation, or state that it is unknown.

Reply: We added an explanation of the better alignment at the poles of the brain as follows (Section 2.3.2 par 3).

Slabs closer to the rostral and caudal poles of the brain had higher average Dice scores than those closer to the center of the brain. This is likely because the anterior and posterior poles provide clear boundaries versus which to position the tissue slabs. Additionally, slabs 2-4 contain the temporal lobe which, because the brains were unfixed, is pressed against the dorsal lobes of the brain in the 2D images and requires significant non-linear deformation to correct.

Comment 2.13.8. *Which slab index corresponds to the anterior pole?*

Reply: Added labels indicating the slabs corresponding to the anterior and posterior poles.

See Comment 2.13.5.

Page 14

Comment 2.14.1. Dice "score"

Reply: "Dice" has been changed to "Dice score" throughout the text.

Comment 2.14.2. *I don't see the point why the registration accuracy within the slabs should be superimposed across the slabs, and no interpretation of the graph is given either. Individual accuracy profiles in separate graphs would be more readily interpretable.*

Reply: *We removed the plotting of the Dice scores versus position on the slab.*

Figure 6: A) Dice scores for two different validations are shown. Reference: Dice score between aligned sections versus corresponding coronal section from reference volume, Inter-section: Average Dice score between aligned section versus neighbors in posterior and anterior direction along coronal axis. Overall Dice score were high 0.89-0.95 but with some sections that appear to be poorly aligned. B) Examples of sections that were either very poorly (left) or well (right) aligned. The grey images indicate the non-linearly aligned 2D segmentations and the hot (orange-red) lines are the edges of the target 2D section from the structural reference volume.

The poorly aligned sections show that the source of misalignment is frequently 2D sections that are either very small (Slab 1,2,6), poorly segmented (Slab 4), or incomplete sections (Slab 2,3).

The sections that have been well aligned consistently demonstrate a Dice score of 0.97, indicating an upper ceiling for BrainBuilder.

Comment 2.14.3. *Missing directionality: Is 0 or 100% closer to the anterior pole?*

Reply: See above.

Comment 2.14.4. *Please provide a forward reference to the methods section, so that readers know that a more detailed description is available.*

Reply: We added a reference to the appropriate Methods section in 2.3.3 par. 1.

A surface-based interpolation algorithm (section 4.2.4) was used to estimate missing pixel intensities between acquired autoradiographs.

Comment 2.14.5. *Please provide further details about how this measurement was carried out. Ideally, provide a supplementary figure explaining the procedure.*

Reply: I added another reference to the Methods section later in 2.3.3 par. 1.

The surface-based interpolation was validated (section 4.3.3)

Page 16

Comment 2.16.1. *A large majority of this section is a repetition of the results section. In my opinion, this is not necessary.*

Reply: We shortened the recapitulation of the results to one paragraph. The text below gives high level takeaways for the reader that has skipped the results section.

Visual and quantitative validation demonstrated that BrainBuilder effectively reconstructs the 3D anatomy of the reference volume, as shown in Figure 4. The average Dice scores ranged from 0.93 to 0.95 across two alignment metrics in both human and macaque brains. Lower Dice scores were observed in slabs located farther from the rostral or caudal poles (Table 1), suggesting that the slab's position and anatomy influence the reconstruction's accuracy, likely because the rostral and caudal regions provide more easily identifiable landmarks. Additionally, receptor densities measured from manually defined cytoarchitectonic regions in the autoradiographs showed a close correlation ($r^2 = 0.95$) with those measured in the 3D reconstructed volumes using an external 3D cytoarchitectonic atlas⁴. The surface-based interpolation also displayed high accuracy, with a correlation of $r^2 = 0.97$ between true and estimated pixel intensities, confirming reliable estimation of missing pixel intensities across the cortical surface.

Comment 2.16.2. *I would expect that the opposite is true: given the many morphological features of the central sections, even a small misalignment can be easily identified. However, closer to the poles, the complexity of the cortical ribbon decreases, allowing translational misalignments to become larger.*

Reply: The reason for the superior alignment at the poles is likely a combination of the relatively simpler cortical morphology at the poles (e.g., slab 3 is challenging because the temporal lobe is pushed against the frontal lobe in the 2D sections) and because we have to estimate, using expert neuroanatomist annotations, the start and end points of the slabs on the reference volume. The estimation of the slab end points is relatively easier at the poles because one of the slabs can be easily identified.

Comment 2.16.3. *Here the authors speculate on the accuracy of template-based reconstruction. The ability to use a template volume instead of the subject's own, seems to be one of the major strengths of the presented method. However, the authors did not take the opportunity to test the accuracy of such substitution on a dataset, even despite they had access to both. Before drawing conclusions on this capability, I would expect the authors to repeat the reconstruction of the human brain dataset, using an age-matched reference template.*

Reply: The question of the accuracy of template based reconstruction is indeed very important. A stereotaxic template was only used for the macaque data where we did not have any MRI scans for the specimens. While we do have MRI scans for the human donors, we explicitly recommend against the use of the templates for reconstructing human brains because of the high degree of variability in human cortical anatomy. Reconstructing the human brains with stereotaxic templates wouldn't be very informative because the extent to which it works well or not would only be reflective of how well that particular individual's brain resembles the template. There is no guarantee of such results generalizing because (outside of major anatomical landmarks) individuals frequently have more or less gyri/sulci with different shapes and discontinuities. So while this is a good suggestion, we would need MRI scans of the macaque to be able to perform it, which we unfortunately do not.

Page 17

Comment 2.17.1. *As far as I understand from 4.2.3.2, this is not true for the human dataset, or any other dataset, where slabs need to be registered into a whole brain volume. Please state this clearly. The current description is unilateral and potentially misleading.*

Reply: We added text to make clear that BrainBuilder is only fully automated when all the tissue comes from a single tissue slab spanning the whole brain, Section 3.1 par. 1.

It should also be noted that no manually selected points are used in the reconstruction of the macaque data because the data were acquired within a single tissue slab. Hence BrainBuilder can be used in a fully automated fashion when the sections have been acquired within a single slab of tissue spanning the entire brain.

Comment 2.17.2. *Certain cost functions, such as normalised cross-correlation (NCC) should allow this, as long as the evaluation of similarity is constrained to the individual 2D sections, and no interpolation takes place between adjacent sections with different contrast properties.*

Reply: This sounds like an interesting suggestion, but one would still have the problem that the relationship of pixel intensities between a particular receptor and the reference volume might be nonlinear (e.g. regional bright spots in the receptor maps). Do you know of software that implements something like this? It's an interesting idea for future work.

Comment 2.17.3. *What about sparsity prohibits the use of this method? As far as I understand, sparse but regular sections would correspond to very thick slices.*

Reply: The sparse sections can only be treated as thick slices if they're uniformly sampled at regular intervals. In practice we have to interpolate between the acquired sections to create a volume that can be aligned to the reference, which requires an extra step we had to add to the standard 3D-2D algorithm.

We have modified the text to make this a bit clearer:

However, to our knowledge, these methods cannot be directly applied here again due to the fact that although our dataset consists of sparse, **irregularly sampled sections with heterogeneous pixel intensities that cannot be naively stacked and aligned in 3D to a reference volume.**

Comment 2.17.4. *As far as I can tell from Figure 1C, the cortical ribbon is usually very distinct from the white matter, so even if the strict form of the assumption is violated, I would still expect a decent performance from this method. Furthermore, the same trick that was employed in BrainBuilder, which is to convert the input slices to segmentations/probabilities, would have been a reasonable preprocessing step for the method of Iglesias et al, and a comparison would have been adequate.*

From the wording of this sentence, it appears to me that a comparison with this method could have been dismissed for other reasons that are not mentioned here.

Reply: The MRI signal is to some degree a function of the cell density measured in the histology, whereas receptor distribution can be totally independent of it, and so while the borders of the cortex might be correctly aligned, the receptor layers might be distorted. This assessment could be incorrect--or could be theoretically correct but to not matter in practice--so it would be useful to try the Bayesian alignment and see what it looks like.

Page 18

Comment 2.18.1. *Great!*

Reply: We contacted Juan Eugenio a few years ago about this but never got a response. Still looking forward to giving it a go.

Comment 2.18.2. *Up until this point, there was no mention of the fact that the underlying framework is a neural network. Since the work fits into the context of mostly traditional optimisation methods, I would urge the authors to make this more apparent in the introduction and include a justification why a network architecture was used.*

Reply: To address concerns about the quality of the segmentation we have added a more detailed description of methods (section 4.2.3.1 par. 1-2) and results (section 2.3 paragraph (1)) for neural network segmentation. In our validation tests, the network achieves high dice scores when tested on real 2D brain images from human (Dice Score = ~0.95) and macaque (Dice Score = ~0.93). If for some reason the neural network fails on a given section, Otsu thresholding is used as a backup so that the whole reconstruction does not fail. Otsu thresholding yielded a Dice score of ~0.8-0.85 on the manual labels. We have added the following section in the revised manuscript.

2.3.1 U-Net GM segmentation

(1) A U-Net³⁸ was trained to segment the GM in 2D sections using exclusively synthetic 2D images generated from a segmentation of the BigBrain. The segmentation accuracy of the U-Net was evaluated by calculating the Dice score³⁹ between GM segmentation predicted by the U-Net from real autoradiographs versus manually drawn GM labels for these sections. For comparison, the Dice scores were also calculated for segmentation produced with Otsu^{39,40} thresholding. The results showed an average Dice score of 0.93 ± 0.04 for 40 human sections and 0.91 ± 0.03 for 20 macaque sections, indicating that the network generalized reliably from human to macaque brain sections. The Dice score obtained with Otsu thresholding was only 0.85 ± 0.08 in the human sections and 0.8 ± 0.14 in macaque sections, showing a significantly more accurate and precise GM segmentation with the U-Net trained on synthetic data.

4.2.3.1 U-Net 2D GM Segmentation

(1) Binary GM sections are generated from the sections using a deep neural network. A multi-tissue segmentation of BigBrain at 200 micron isotropic resolution was used to generate 10,000 2D training images (Figure 10). To better synthesise intact *in vivo* MRIs, a skull segmentation of the MNI152 atlas was coregistered to the BigBrain and used in 50% of training examples. To generate synthetic training examples, random affine transformations (with random scaling \sim uniform(0.9, 1.1) and rotations \sim uniform(0, 15°)) were first used to create randomly transformed versions of the segmented BigBrain volume. From each of these transformed volumes, 500 2D sections were extracted and used as a training example. To synthesise heterogeneous laminar structure, the cortex was segmented into a random number of cortical layers between 0 and 11, with random proportional thicknesses⁴⁸. The considered tissue classes included: WM, subcortical GM, cortical GM layers, cerebellar GM, within- and outside- brain background, and skull. For each training example, randomly generated values were assigned to each tissue class in the 2D section from Gaussian distributions. To introduce additional variation in the synthetic images, random smoothing, noise, filtering, scaling, and cropping were applied to training examples (see *Supplementary Information 5.1*).

(2) The nnUNet (v.1) software package⁴⁹ was used to automatically identify optimal hyperparameters and train a U-Net³⁸. The U-Net was trained to segment synthetic sections into cortex, background and white matter, with a second task to identify the pixels at the boundaries between these image classes.

Figure 10. Example of generated synthetic images for training the segmentation U-Net. The left column shows examples of the synthetic images with different random values assigned to cortical layers, WM, cerebrospinal fluid, skull, and background. As can be seen in the synthetic images on the left, data augmentation was used to add artefacts to the synthetic images, including: a) affine transformation of the BigBrain volume, b) gaussian smoothing and noise, c) zoom factors, and d) occluded rectangular regions. Two sets of labels are used in the training: a) GM and WM regions (middle column), b) WM-GM and GM-pial surface boundaries (right column).

4.3.1 U-NET Cortex Segmentation

(1) The prediction accuracy of the U-Net was validated using manually segmented GM images from randomly selected autoradiographs from the human data (n=39) and macaque data (n=10). The manual GM segmentations were drawn on the raw images using the GNU Image Processing (GIMP) software. The raw and segmented images were downsampled to images of 415 x 558 pixels to reflect the image size used to train the network. For comparison, segmented GM images were also created using Otsu thresholding⁴⁰. The Dice score³⁹ between the predicted GM, both using the U-Net and Otsu thresholding, and the manually segmented GM was used to validate the accuracy of the segmentation.

Comment 2.21.1. *I'm getting confused about this terminology. Does this refer to a slab, or a whole-brain volume, or either?*

Reply: It refers to either. In the case of the macaque it's the whole brain, in the human data it's a slab of tissue. We added two sentences to clarify this point (Section 4.2.3 par. 1):

If the sections were sliced from a whole brain, then the reference structural volume comprises a whole brain. If, as is the case of the human data in this work, the sections were sliced from tissue slabs, then the reference structural volume is also a slab extracted from a whole brain.

Comment 2.21.2. *Given that this is a reference to a conference proceeding, it is necessary to describe the network in more detail here.*

See Comment 2.18.2.

Comment 2.21.3. Does "in practice" imply that the network failed on all macaque sections, because it was trained on BigBrain, which is a human brain?
Please account for using different approaches, and ideally provide a figure where representative examples from the two methods are contrasted.

Reply: In this current resubmission, we have fixed a bug that was preventing the network from working well on the macaque brains. Now all sections are segmented using the neural network. We have included a validation of the network on manual labels on human and macaque autoradiographs and provide a comparison between the accuracy of the segmentation compared to Otsu thresholding.

See Comment 2.18.2.

Comment 2.21.4. As a natural consequence of optimizing 2D transformations, or is there a 3D transformation taking place here?

Reply: Yes, as a consequence of the 2D transformations. Although the alignment procedure alternates between 3D volumetric and 2D section-wise transformations, the raw images are only transformed in 2D until at the end the 3D volumetric transformation is used to bring everything into the reference structural volume coordinate space. We changed 3D GM volume to 2D sections in the test because this should make it clearer (Section 4.2.3.1. par. 4):

Therefore, after the first step of the multiresolution hierarchy, the 2D GM sections are warped so that they better correspond to the actual anatomy of the reference structural volume to which they were aligned at the previous resolution of the hierarchy.

Page 22

Comment 2.22.1. Why not linear or spline interpolation?

Reply: Both nearest neighbor and linear interpolation were implemented. The current version, updated since the first submission, uses linear interpolation. At the current resolution of reconstruction (250um) there isn't a significant difference because the interpolated volume is downsampled from the original in-plane resolution of 20um. However, as we approach higher resolution reconstruction this will become more important and would be important future work. Moreover, some users may already have data that supports reconstruction at very high resolution. We have added a discussion of this issue to the Limitations (Section 3.3 par. 1):

3D segmentation volumes

(1) Currently, BrainBuilder uses linear interpolation to fill missing segmented images between sections to create continuous 3D segmentation volumes. However at higher spatial resolution this may produce jagged 3D volumes. Hence we implement higher order interpolation algorithms to create smoother 3D segmentation volumes for alignment to the reference structural volume.

Comment 2.22.2. I would expect that the accuracy of this manual definition is crucial, and therefore warrants a more detailed description of how the selection was made.

My expectation is that any error in the delineation of the slab boundaries will have significant non-local effects during the 3D registration with ANTs, as it will aim to line up the tissue edges before anything else.

Therefore, if extra tissue is present in the slab selection, all sections will be drawn from their actual position towards the slab surface, rendering the registration inaccurate.

Reply: We added additional text (Section 4.2.3.2. par. 2) describing the procedure for defining manual points for defining tissue slabs on the structural reference volume. We very much shared this concern. However, it can be seen from Figure 4 (e.g., the posterior portions of slabs 3 and 4) that ANTs does not necessarily pull the sections to the bounds of the manually defined slab boundaries on the reference volume.

The points were identified by an expert neuroanatomist who localized structures that were easily and uniquely identifiable in both the MR images and both autoradiographs and photographs of the slabs before and after their shock freezing. These structures included the fundus of sulci, tips of gyri and the edges of dimples. We also identified the most rostral and/or caudal portions of the corpus callosum and of subcortical structures such as the caudate nucleus or the pulvinar nucleus. Importantly, we did not restrict identification of microanatomical landmarks to the most rostral and caudal portions of the slab, i.e., to the portion corresponding to the interface between slabs. Rather, we set landmarks throughout the entire lateral and medial surfaces of each slab.

Comment 2.22.3. Please clarify what type of linear registration is used here. The referenced supplementary information lists non-linear deformation too, and it is not clear if the linear alignment mentioned here would incorporate scaling.

Reply: Apologies, this was probably the worst typo we could have made. It should say “is non-linearly aligned”. Indeed the approach wouldn’t make sense with only linear alignment. It is now corrected in Section 4.2.3.2. par. 3.

For each slab, the reconstructed GM volume is non-linearly aligned with ANTs⁴¹ to the portion of the reference structural volume.

Comment 2.22.4. *Given that the slab-to-brain alignment was linear, where does the pipeline account for bulk deformations of the slab during handling?*

Reply: See Comment 2.22.3 above.

Comment 2.22.5. *Please provide further information about the software used, as well as actual mesh sizes. Was the interpolation barycentric or nearest-neighbour?*

Reply: The code is implemented in BrainBuilder. We added text in 4.2.4.1 par. 1. stating this and explaining the upsampling of the mesh.

Each cortical mesh is then supersampled in BrainBuilder such that the maximum distance between any two neighbouring vertices is less than or equal to the final resolution of the reconstruction, i.e., 500µm for the human reconstruction and 1mm for the macaque reconstruction (Figure 11.C). Specifically, for each triangle of 3 vertices (A,B,C) on the surface mesh, two vectors are defined (AB, AC). Points inside the surface triangle, P, are defined by the linear combination of the two vectors by scaling factors α and β, where α and β are defined as increments of one half the reconstruction resolution, r.

$$P = A + \alpha \mathbf{AB} + \beta \mathbf{AC}; i = \mathbf{AB}/(0.5r); \alpha \in [0,1], \beta \in [0,1], \alpha + \beta \in [0,1],$$

This upsampling step produces a set of points such that there is at least one vertex per voxel in sections where sections have been acquired. For a reconstruction of 0.25mm the upsampled mesh in the humans was 12,129,727 vertices in the human and 1,022,442 vertices in the macaque.

Page 23

Comment 2.23.1. *Given the uniqueness of the dataset, it would be desirable to have everything computed at the highest reasonable resolution. Please justify your choice of these two resolutions.*

Reply: The data has all been reconstructed to 0.25 mm and the figures have been updated accordingly.

Comment 2.23.2. *It is not clear why this has to be done again. I assumed that you already inflated and upsampled the cortical surfaces in the previous section. If this is a different surface,*

please clarify. If the same tools were used in the previous section, they should be referenced there.

Reply: We added text to clarify that the spherical surfaces are being upsampled, and not just resampled, to support reconstruction at the desired spatial resolution.

The inflated spherical meshes are then **upsampled** to the same number of vertices as the **already** upsampled cortical surfaces (Figure 11.F) **so that the vertex values can be estimated at a sufficient spatial resolution for the desired reconstruction resolution.**

Page 24

Comment 2.24.1. *My understanding is that you are taking a reconstructed volume from sections, where you fill in the gaps with interpolation. Then you resample the cortical data onto a set of parallel surfaces, where you fill in the gaps with interpolation, and then for the third time, you resample the twice-interpolated surface data into 3D space, and fill in the gaps with interpolation. At this point, the original data have been interpolated four times, which means there is potentially a significant degradation.*

I do not see a justification for this process, hence, it would be necessary to show that the process 1) leads to no significant degradation of the data, 2) has an advantage over using the reconstruction data from the first step. Ideally, you should present a difference map between the receptor density volumes and the same volumetric information derived from the reconstruction (prior to the surface-based resampling).

I appreciate the validation that was carried out with regards to the surface-based interpolation method. However, this is just one step in the above process, and while the correlation is maintained, the dispersion of the datapoints in Figure 8 clearly illustrates the degradation of the data in just a single step of interpolation. Furthermore, this validation uses originally acquired data, and does not take into account the gaps that were filled with already interpolated data.

Reply: Indeed this point is well taken and we have adjusted the reconstruction method to limit the number of interpolation steps. There are at minimum 2 interpolation steps that cannot be easily avoided: a 2D and 3D non-linear transformation to align the sections with the reference volume. These 2 interpolation steps allow the pipeline to create a 3D volume with 2D sections warped into the 3D reference space. There is then at minimum a 3rd interpolation step that is required to fill the missing sections between the reconstructed sections. We have implemented a validation step to assess the effect of the 3D alignment on interpolation error in 4.3.4.

While we could use a straightforward trilinear, cubic, or other interpolation scheme through euclidean space between the reconstructed sections, we have opted not to do this because this approach does not respect the laminar structure of the cortex (see Comment 2.5.4).

We've also added a supplementary analysis mostly to help validate the accuracy of the alignment, but that also provides some evidence of BrainBuilder's interpolation accuracy. From

previous work, we have a set of averaged receptor densities from cytoarchitectonic areas defined manually on the raw 2D sections. Many of these areas are also defined in the Julich cytoarchitectonic brain atlas, which was created using a completely separate histological data set. Unfortunately, full cytoarchitectonic areas segmentation on our autoradiographs dataset is not available. Nevertheless, we validated the alignment by comparing the average receptor densities extracted directly from the raw autoradiographs to receptor densities extracted from the reconstructed volumes using 3D regions defined on the Julich brain atlas. We found a close correlation between these values demonstrating that the interpolation is at the very least accurate at the scale of cytoarchitectonic areas.

Nonetheless, if users are not confident in the surface-based interpolation scheme, they can use the 3D volume containing only the sections warped into 3D (without any estimation of “missing” pixel intensities between acquired sections).

4.3.4 Validation of reconstructed pixel intensities

(1) To ensure that the reconstruction pipeline did not affect the measured pixel intensities, regions in the unprocessed autoradiographs were compared to the corresponding regions in the reconstructed volumes. Regions of interest (ROIs) were generated based on similar patterns of pixel intensity distribution, with a mean area of $21 \pm 11 \text{mm}^2$ (see Figure 13 for illustration). These ROI were not intended to be biologically meaningful per se, but rather to reflect the spatial characteristics of real parcellation schemes, e.g., ROI from atlases such as the Julich Brain⁴, which we anticipate will be used to analyse the reconstructed data.

Figure 13: A random parcellation was generated for each autoradiograph. The regions of interest were transformed using the 2D non-linear transformation (B) and the 3D non-linear transformation (C) of the autoradiograph on which they were defined. After 3D transformation, the autoradiographs and corresponding parcellations are no longer defined in a 2D plane but in a 3D volume. Hence the right-most image shows the 3D reconstructed volume from a sagittal view with the red line showing the position of the warped autoradiograph and parcellation after 3D transformation.

(2) The parcellation of ROIs was produced using K-Means clustering with the “slic” function from Skimage⁵³. The autoradiograph parcellations were transformed using the same transformations

as were applied to their corresponding autoradiographs (Figure 4). That is, a non-linear 2D transformation (4.2.3.2) and a 3D non-linear transformation (4.2.3.3). The accuracy of the transformed pixel intensities was quantified by calculating the absolute error between true and interpolated values, divided by the true intensities.

2.3.3 Interpolation Accuracy

...

Figure 9: Binding densities were measured from human autoradiographs after applying 2D (left) and then 3D (middle) non-linear transformations to the autoradiographs and compared to binding densities measured from the raw autoradiographs. While the mean accuracy is similar after 2D alignment and 3D reconstruction, the variance of the accuracy increases for smaller sections after 3D reconstruction.

4.3.2 Validation of alignment of human and macaque brain sections

...

(3) Finally, to validate the alignment using independently defined cortical regions, we compared the neurotransmitter receptor densities measured in specific cytoarchitectonic areas, manually defined on the raw 2D images from previous work by expert neuroanatomists^{34,35}, to the densities in the same cytoarchitectonic areas as defined on the Julich Brain cytoarchitectonic atlas⁴. The donor brain was non-linearly aligned using ANTs⁴¹, and the Julich Brain parcellations were transformed onto the donor brain. We selected 29 regions that were defined both in the manual 2D annotations of the autoradiographs and on the Julich Brain atlas. The regions included the visual, primary sensory, motor, cingulum, parietal, and orbitofrontal regions (See Supplementary Information 5.3 for the list of regions). The receptor densities in the 2D annotations were measured in previous work, while the JuBrain regions were used to measure

receptor densities in the 3D reconstructed volumes. To assess the accuracy of the reconstruction, we performed a regression analysis of the 2D regional receptor densities onto the corresponding receptor densities measured in the 3D reconstructed volumes using the JuBrain atlas.

2.3.2 Alignment of 2D Section to Reference Volume

...

(4) The alignment of the sections was also validated against an external source. Average regional receptor densities were obtained for each of the 20 receptor binding sites from 29 paired regions (see section 5.4 for details) in a) manually defined cytoarchitectonic areas in the raw autoradiographs³⁴ and b) from the 3D reconstructed receptor volumes using the 3D Julich Brain Atlas⁴ of cytoarchitectonic regions. The manually defined regions consisted of one-dimensional profiles through the cortical depth but did not span the entire cytoarchitectonic area. Direct areal comparison of the 1D profiles to the 3D Julich Brain Atlas was therefore not feasible and regional densities between the corresponding regions were compared instead. The regional densities measured from the manually defined 2D regions were regressed onto the densities obtained in the 3D reconstruction. This produced an r^2 of 0.95 (Figure 7), indicating a close correspondence between cytoarchitectonic areas defined in the raw sections and those defined in an independent 3D atlas.

Figure 7. The strong correlation between regional receptor densities measured from areas manually defined on 2D autoradiograph versus the same regional densities defined in the same regions on the Julich Brain Atlas⁴ indicate that BrainBuilder produces accurate alignments at the scale of cytoarchitectonic areas.

Page 26

Comment 2.26.1. *Nice figure! It is very useful to understand the actual process.*

Reply: Thank you!

Comment 2.26.2. *This validation only concerns the accuracy of the registration mechanism, which was performed by ANTs, as prescribed by the authors' multi-resolution multi-step framework. The evaluation does not take into account the possible errors of the segmentation that was used to homogenise the sections, and appears to be a novel contribution above using ANTs to align sections to volumes. It is therefore insufficient on its own to underpin the conclusions about BrainBuilder's accuracy.*

Reply: We've added validation for the GM segmentation of the sections. See Comment 2.18.2.

Comment 2.26.3. *If the resolution is 500 microns, and the voxels are isotropic, the voxel volume is not 500um³. Please clarify, or indicate isotropic 500um voxels by using parentheses before the exponent.*

Reply: Changed to 250um isotropic, because the reconstruction is now at this resolution, Section 4.3.2 par. 1.

To demonstrate the efficacy of BrainBuilder, it was applied to sections acquired from human and macaque brains. The final reconstruction for the human and macaque data was generated at 250µm isotropic voxel resolution.

Comment 2.26.4. 1 mm isotropic

Reply: Removed this portion as it is no longer necessary. See next comment.

Comment 2.26.5. *While it may be sufficient, it is not a justification for diverging from the already working human protocol. Given that the monkey brain is even smaller, computational requirements should not prohibit the higher-resolution reconstruction. Please elaborate the considerations behind this decision further, as a lower resolution reconstruction has the potential to conceal inaccuracies.*

Reply: The macaque and human data are reconstructed at the same resolution of 250um.

Comment 2.26.6. Please quantify, or give an estimate.

Reply: We've quantified the amount of sampling coverage of the brain and added a paragraph specifying the number of sections that failed quality control,

Section 1.2 par. 7

To illustrate the sparsity of the data used here, only approximately 37% of the donor brain was sampled with complete sections with an average distance of 1.03mm and a maximum distance of 1.42mm between sections measuring the same biological feature. Sections were either lost due to mechanical processing errors or excluded because the quality of the acquisition was poor (e.g., binding artefacts or blurred histological staining) or because the sections contained only a fragment of brain tissue, frequently as a result of slicing the sections close to the border of the tissue slabs. In the macaque the sections covered approximately 24% of the brain with a sampling of 1.32mm between acquired sections for the same biological feature. The macaque brain was deliberately sampled sparsely in repeats of sections along the coronal axis to allow sampling through the brain while limiting the cost of data acquisition.

Section 4.1 par. 5.

Sections were quality controlled and sections that were severely damaged, had clear binding artefacts, were visualized for non-specific binding, or were missing more than 25% of the cortex were excluded. In the human data 693 sections failed quality control, most of which were at the

ends of slabs where only small pieces of cortex were sliced from the frozen tissue slabs, and 384 for the macaque.

Comment 2.26.7. *5 mm² would be equivalent to 4.5x4.5 pixels at 500um isotropic resolution in 2D. I assume this was 10x10px instead, so the Dice score was calculated on 100 pixels.*

Reply: This was a mistake, the sliding window kernel is 5 x 5 pixels, and hence consists of 25 pixels. This was chosen based on the fact that the ANTs local cross-correlation metric usually recommends 3x3 to 6x6 pixel windows.

To avoid this problem quantitative validation was performed by calculating the Dice score within a 5x5 moving window between the aligned

Comment 2.26.8. *I would strongly suggest simplifying this expression across the whole document. Would it be enough to say "receptor" or "molecular target"?*

Reply: We agree that “neurotransmitter receptor binding site” is cumbersome. “Receptor” would technically be inaccurate because receptors can have multiple binding sites. We’ve amended the manuscript to shorten the expression to just “binding site” where possible, which seems analogous to “molecular target”.

Comment 2.26.9. *It is hard to believe that vertices intersect the razor-thin autoradiographs in such numbers. Would you please clarify how intersection was defined?*

Reply: The sections are thickened to the resolution of reconstruction, i.e., 250um, and the surfaces are upsampled so that the maximum distance is ½ the reconstruction resolution. The human surfaces are upsampled to more than 12 million vertices. We've added this information in 4.2.4.1 par. 1:

For a reconstruction of 0.25mm the upsampled mesh in the humans was 12,129,727 vertices in the human and 1,022,442 vertices in the macaque.

Page 27

Comment 2.27.1. *sampling distance?*

Reply: “Sampling distance” is indeed correct and has been added in Section 4.3.3 par. 3.

In the human data, the average sampling distance between sections of the same type was approximately 570µm.

Page 33

Comment 2.33.1. *Does this imply weighting by the original intensities, or the weighting by a binary mask?*

Reply: The weighting is based on the segmented GM mask, so in practice the center of mass should consistently be the center of the cortex, Section 5.3.2. par. 1.

Both images were GM segmentations, hence the center of mass should be the center of the cortex.

Comment 2.33.2. *Did you prove that this was necessary, or was it an empirical choice? If the latter, please state that this was your choice.*

Reply: Apologies this was poorly explained. In order to get consistent levels of smoothing and downsampling at each resolution, it's necessary to define the downsampling/smoothing factors as a function of where the algorithm is in the resolution hierarchy. Put another way, if at each resolution we naively used the standard 8x4x2x1 set of downsampling factors, then during the 4mm alignment the maximum downsampling would be 32mm (8x4mm) but at the 0.25 mm resolution the maximum downsampling would be 2mm (8*1/4mm).

We've simplified the language so hopefully this should be clearer in 5.3.2

For each alignment step in the multi-resolution hierarchy, the downsample factors passed to antsRegistration were set such that the alignment would be calculated using all the previous resolutions in the resolution hierarchy.

Comment 2.33.3. *Great job! Setting the resolutions and smoothing factors consistently across every run of the program is a welcome practice supporting reproducibility and generalisability.*

Reply: Thank you! Yes it was important to get consistent results.

Page 34

Comment 2.34.1. Please provide a rationale for using this numerical constant.

Reply: This is the formula for calculating smoothing factors that is used in the ANTs package. We have added this to the text in Section 5.3.2 par. 3.

The scalar value of 0.2 is used because this is the equation used in the ANTs package to calculate smoothing factors based on downsampling factors.

Reviewer Comment: 1. The authors acknowledged the large section spacing and discussed their choice of linear interpolation. They also mentioned plans to explore more advanced interpolation methods in future work.

o My comments: The concern about interpolation accuracy across large gaps remains. Linear interpolation may not capture fine-grained structural changes, especially in receptor mapping where subtle variations are important. Additionally, the authors noted that sectioning is rarely perfectly perpendicular to the pial surface, which can misrepresent laminar distributions and exacerbate interpolation errors. Another major concern is that only approximately 37% of the human donor brain was sampled with complete sections, with an average distance of 1.03 mm and a maximum distance of 1.42 mm between sections measuring the same biological feature. For the macaque brain, only 24% of the brain was covered, with a sampling distance of 1.32 mm between sections. Given these large gaps, it becomes difficult to be sure that the reconstructed receptor maps accurately represent the real underlying biological structures. In fact, typical MRI scanners already achieve a resolution of 1 mm, which is comparable or even better than the sampling performed here.

Although MRI cannot capture receptor signals directly, the resolution mismatch raises concerns about whether the reconstructed maps genuinely provide mesoscopic-level detail. Including a comparison with receptor density measurements from PET scans could help validate the biological accuracy of the reconstructed maps.

Authors' Response:

We appreciate the reviewer's concerns regarding the impact of sparse sampling and sectioning angle on interpolation accuracy and the fidelity of reconstructed receptor maps. We fully agree that the accuracy of any reconstruction is ultimately constrained by the sampling rate of the raw data. In our case, the relatively low sampling density—approximately 37% of the human brain and 24% of the macaque brain—reflects limitations of the available autoradiography datasets, not of the BrainBuilder methodology itself.

The aim of this manuscript is to present a robust and flexible reconstruction pipeline capable of handling such challenging, sparsely sampled, multimodal datasets. Importantly, these limitations would not apply to datasets with higher sampling density, such as densely acquired histological series, which BrainBuilder can readily accommodate.

To mitigate sectioning artifacts and account for curvature-related distortions, we implemented a surface-based interpolation method that propagates receptor density values across cortical layers, which more accurately reflects the folded structure of the cortex than linear interpolation in Euclidean space. While this does not eliminate all limitations arising from sparse sampling, it is a biologically informed and scalable approach appropriate for the current data.

Following the reviewer's recommendation, we empirically assessed the accuracy of the reconstructed receptor maps by comparing the reconstructed benzodiazepine volume to in vivo [18-F]-flumazenil PET scans from ten healthy participants. This comparison is particularly

appropriate because both PET and autoradiography used the same radioligand. As described in the revised manuscript, the reconstructed volume showed no greater dissimilarity to the PET scans than the PET scans did to one another (Spearman's correlation = 0.71-72), supporting the biological plausibility of the reconstruction. We believe this is a particularly strong result and thank the reviewer for their excellent suggestion.

Lastly, while current MRI resolution may limit the spatial precision of alignment steps in BrainBuilder, this is a general constraint shared by many reconstruction methods, including those inspired by Malandain et al. (2004). In the absence of block-face imaging, we address this by generating a high-resolution GM mask (250 μm) through supersampling between white and gray matter surface meshes. As higher-resolution MRI becomes more widely available—e.g., through the use of ultra-high field scanners—BrainBuilder is fully compatible and will benefit directly from improved structural reference volumes without requiring modifications to the pipeline.

Change to manuscript:

2.3.4 Reconstructed values versus in vivo PET

The reconstructed GABA_A-benzodiazepine receptor volume exhibited a high degree of similarity to 10 PET-derived GABA_A-benzodiazepine distributions from healthy controls. Spearman's ρ correlations between the reconstructed volume and PET scans ($\rho=0.71\pm0.06$) were comparable to those observed among PET scans themselves ($\rho=0.72\pm.05$). Permutation testing ($n=10,000$) revealed that the mean difference in correlation was not statistically significant ($p=0.54$), indicating that the reconstructed receptor volume was as similar to the PET-derived receptor distributions as the PET scans were to one another. These results suggest that the reconstructed autoradiography-based receptor distribution closely aligns with in vivo PET-derived receptor mapping.

Receptor densities observed in the reconstructed GABA_A-benzodiazepine volumes were also not statistically different from the distribution of benzodiazepine receptor densities measured from in vivo PET scans in healthy participants, indicating accurate reconstruction of the autoradiographs. (lines 406-421)

4.1.3 Positron Emission Tomography Acquisition and Preprocessing

[18F]Flumazenil PET scans were acquired for 9 healthy subjects (age 61 ± 10 years, 9 males). PET scans for all subjects with an ECAT HRRT PET scanner in list mode (Siemens Medical

Solutions, Knoxville, TN, USA) (Wienhard et al., 2002), with a spatial resolution of ~2.4mm FWHM. After a transmission scan for attenuation correction (¹³⁷Cs-source), approximately 370 MBq [¹⁸F] FMZ were injected intravenously as a slow bolus over 60 s. The list mode data were acquired for 60 min after injection and were subsequently binned into 2209 sinograms (each of size 256 radial bins × 288 azimuthal bins) using span 9 compression for a total of 17 time frames (40 s, 20 s, 2 × 30 s, 360 s, 4 × 50 s, 3 × 300 s, and 3 × 600 s), resulting in images with a voxel size of 1.22 × 1.22 × 1.22 mm³. Fully 3D FBP by 3D reprojection (3D RP) was carried out with a Hamming windowed Colsher filter (alpha = 0.5, cut off at the Nyquist frequency).

For each scan, normalized standardized uptake value ratio (SUVR) images were calculated by dividing the PET images by the mean radioactivity concentration in the WM. (Lines 647-661)

4.3.4 Validation of reconstructed benzodiazepine receptor volume versus PET

To assess the validity of the reconstructed GABA_A-benzodiazepine receptor volume, we compared its regional receptor density measurements to those obtained from [¹⁸F]-flumazenil PET scans acquired from nine healthy controls. We quantified the similarity between the reconstructed receptor volume and WM normalized SUVR images derived from the PET scan and, for comparison, measured the similarity among SUVR images themselves. SUVR images were used because they partially control for the amount of non-specific ligand binding versus specific binding to the receptor. This approach tested whether reconstructed receptor volumes exhibit a level of similarity to PET-derived receptor distributions comparable to the similarity observed among PET scans themselves.

Both the benzodiazepine receptor autoradiographs and PET scans were acquired using the [¹⁸F]-flumazenil ligand, making them ideal for direct comparison. Regional receptor densities were extracted from the reconstructed flumazenil volume using the Julich Brain Atlas (Amunts et al. 2020).

Each PET scan was linearly aligned to the corresponding participant's MRI with a rigid transformation. The MRI scans of the healthy controls and the brain donor were then non-linearly aligned to the MNI152 (2009c) template (Fonov et al. 2011) using ANTs. PET scans and the reconstructed benzodiazepine receptor volumes were subsequently transformed into MNI152 space, and regional receptor densities were extracted using the Julich Brain Atlas (Amunts et al. 2020).

To quantify similarity, we calculated the Spearman's ρ correlation between regional receptor densities from the reconstructed and PET-derived volumes. To evaluate whether the reconstructed receptor volume was as similar to PET-derived receptor distributions as PET scans were to one another, we compared the mean Spearman's ρ from reconstructed vs. PET comparisons to the mean Spearman's ρ from PET vs. PET comparisons. (Lines 935-960)

Reviewer Comment 2: The authors explained that a direct quantitative comparison with existing methods was challenging due to the lack of equivalent tools. However, they did not include any baseline or qualitative comparison.

o My comments: Without any form of comparison, it is difficult to assess BrainBuilder's relative performance. Additionally, while receptor mapping is the primary focus of BrainBuilder, PET scanners, which provide receptor density measurements in vivo, could serve as a relevant comparison. Despite PET's lower spatial resolution (typically 1-5 mm), it directly measures receptor binding, and such a comparison could help validate BrainBuilder's reconstructed receptor maps. Moreover, this verification would go hand in hand with the interpolation issue: if the reconstructed receptor map resembles PET results to some degree, it would suggest that interpolation across large section gaps is performing reasonably well. Without such a comparison, it remains uncertain whether interpolation over large distances (up to 1.42 mm) is producing biologically meaningful maps.

Response: We agree that the comparison with PET imaging provides a useful check to the reconstructed values and thank the reviewer for the suggestion. Please see response 2 above where we detail the PET comparison we have added to the manuscript.

3. Handling of Tissue Artifacts Post-mortem sections often contain artifacts, such as tears or uneven staining, which can affect segmentation and alignment.

o My comments: The authors did not mention any preprocessing steps to handle tissue artifacts or provide examples of how BrainBuilder performs in their presence.

Response: Damaged sections are indeed a major challenge. This is especially true for autoradiographs sectioned from the ends of tissue slabs. As mentioned in lines 143-147, the data were quality controlled to remove tissue with major tearing or staining artefacts. There were nonetheless some tissue sections that were damaged or incomplete, i.e., do not contain a full coronal section of brain tissue, that passed quality control on the basis that they contained enough tissue to be aligned. The lower range of alignment quality is illustrated in Figure 6.B, which shows the worst case scenario of alignment in each tissue slab of the human reconstruction. Essentially this shows that BrainBuilder can grossly align even incomplete sections.

4. Segmentation Reliability The authors provided more details about their segmentation process, showing that fallback to Otsu thresholding was rare and explaining how they achieved high Dice scores (~0.95 for humans, ~0.93 for macaques). Additionally, they validated the U-Net model on manually segmented gray matter (GM) images from real data, using 39 human and 10 macaque samples.

o My comments: The segmentation results look strong, and the validation on manually segmented real data provides confidence in the U-Net model's

reliability. However, it remains unclear how often they had to rely on Otsu thresholding and whether those fallback cases caused any noticeable errors.

Response: We apologize for the confusion. Presently all of the images were successfully segmented with the U-Net. The Otsu thresholding was added because a bug, now fixed, in the code was causing some images to be segmented to 0 and we needed a backup. We are also using BrainBuilder for rodent data for which the U-Net does not yet provide accurate segmentations and hence for which we require an alternative segmentation method. Hence Otsu thresholding is provided to the user as a backup or alternative in case they wish to use BrainBuilder for a species which we have not yet evaluated, e.g., marmoset. We have modified the text to clarify this point.

Change to manuscript:

While all images in the present study were successfully segmented with the U-Net, in the eventuality that the network fails to provide a segmentation, i.e. returns an empty image, a segmentation method based on Otsu⁴⁰ histogram thresholding is used. (Lines 729-731)

5. Scalability of the Approach The current description of BrainBuilder focuses on reconstructing cortical maps from post-mortem sections with significant manual preparation and scanning effort.

o My comments: Given the time-intensive nature of acquiring and processing post-mortem sections, it is important to assess whether BrainBuilder can handle large-scale datasets efficiently. This includes considerations of computational load and memory requirements during segmentation, alignment, and interpolation. Including a brief discussion of scalability and potential automation steps for future work would be helpful.

Authors' Response: We have added text in the Discussion on the computational bottlenecks of BrainBuilder with respect to scalability.

Scalability and Computational Costs

The reconstruction process is computationally expensive due to the large number of sections and the high spatial resolution of the reconstructed volumes. The 3D nonlinear alignment step using ANTs is the primary bottleneck in terms of memory usage; however, if sufficient RAM is available for this step, then the rest of the pipeline can run without additional constraints. For the reconstruction of a human hemisphere at 250 μm resolution, RAM usage peaked between 64 and 128 GB. Extrapolating from this, a 50 μm reconstruction would require approximately 1 TB of RAM, which exceeds the capacity of standard workstations and requires access to high-performance computing (HPC) clusters.

To address these limitations, future versions of BrainBuilder will implement a piecemeal 3D volumetric alignment approach—aligning sub-volumes independently—to reduce the memory burden for very high-resolution reconstructions. This strategy will allow for more efficient scaling while maintaining the accuracy of the reconstructed volumes. (Lines 549-561)

6. Error Propagation Across Reconstruction Steps Since BrainBuilder involves multiple sequential steps, errors in early stages could propagate through the pipeline.

o My comments: There is no discussion on how errors propagate or whether any safeguards are in place to minimize their impact. Including a brief discussion on error propagation and any measures taken to mitigate its effects would improve confidence in the reliability of the final reconstructions.

Authors' response: We have added the following text to discuss the quality control measures that are implemented to prevent error propagation.

Error Propagation and quality control

BrainBuilder relies on sequential processing stages where errors in earlier stages may be propagated to the end result. While the validation tests reported in 2.3, which includes the detection and quantification of spatial and intensity distortions resulting from the deformation of sections, aim to ensure the correct functioning of each step and the accuracy of the final reconstructed values, errors remain a possibility especially with new datasets. Several measures are implemented to facilitate the identification of errors. Quality control images are generated for all downsampled, segmented images, and for the alignment between 2D sections versus corresponding sections in the target reference volume. Additionally, Dice scores are calculated for all 2D alignments to facilitate plotting and identification of outlier scores that may indicate poor alignment. (Lines 561-570)

7. Applicability to Non-Cortical Structures The manuscript focuses on cortical reconstructions, but it is unclear whether BrainBuilder can be applied to other brain regions.

o My comments: Non-cortical structures, such as the hippocampus or basal ganglia, may pose different challenges for segmentation and alignment. If applicable, please clarify whether BrainBuilder can handle non-cortical regions and provide examples demonstrating its performance

Authors' Response: At the moment BrainBuilder only supports reconstruction of the cortex and the hippocampus and does not yet provide reconstructions of the subcortex. Nonetheless, the sagittal cross-section of the human reconstruction in Fig.4 shows that the subcortex is surprisingly well aligned given that this region was not included in the reference volume used for reconstruction. The reconstruction of the subcortex is the next step in the development of BrainBuilder and something we are actively working on. Essentially the challenge with subcortical reconstruction is that we would like to use morphological priors to constrain the interpolation of missing sections. We have added a section on future work to clarify the current scope and future work with regards to the subcortex.

Reconstruction of the subcortex

BrainBuilder presently only reconstructs the cortex and the hippocampus and does not include the subcortex. Nonetheless, the sagittal cross-section of the human reconstruction in Fig.4 shows that the subcortex is surprisingly well aligned given that this region was not included in the reference volume used for reconstruction. This will be improved by the inclusion of the subcortex in the U-Net segmentation network and in the reference volume. Future work will expand BrainBuilder to also include the subcortex by implementing morphologically informed interpolation of missing sections. (Lines 571-578)

COMMSBIO-24-0971-T - BrainBuilder: A Software Pipeline for 3D Reconstruction of Cortical Maps from Multi-modal 2D Data Sets

Short summary:

This manuscript introduces BrainBuilder, a novel method and software package for reconstructing 3D cortical maps from 2D post-mortem serial brain sections. It makes it possible to create mesoscale 3D atlases without the need for a corresponding structural reference volume.

The pipeline covers data preprocessing, alignment procedures, surface-based interpolation techniques, and validation methods. BrainBuilder has been applied to data from human and macaque brains and offers new opportunities for quantitative analysis, cross-species comparisons and provides insights into reconstructing multimodal datasets.

In conclusion, they were able to reconstruct images of 2D sections with an average accuracy of 91% (based dice score) this study can potentially improve our understanding of brain structure and function across different species and modalities,

General Comments

While the BrainBuilder pipeline presents an intriguing approach to reconstructing 3D volumes from 2D multimodal datasets, there are concerns that warrant careful consideration before endorsing its widespread use. It's essential to acknowledge that the method itself isn't inherently flawed; however, certain limitations and areas of uncertainty raise questions about its applicability and reliability in practical settings.

One notable concern revolves around the assumption underlying BrainBuilder's methodology. The pipeline relies heavily on structural reference volumes and intricate alignment processes, implicitly assuming that these references accurately capture the complex anatomy of the brain. This assumption may introduce biases or inaccuracies, particularly if the reference volumes do not adequately represent the variability across different brains or species.

Furthermore, the interpolation of missing sections within the pipeline raises concerns about the accuracy of reconstructed volumes, particularly in regions where sharp boundaries between biological features exist. The linear interpolation method used between vertices may oversimplify the complexity of tissue morphology and potentially obscure crucial distinctions between different receptor architectonic areas, which may lead to inaccuracies in the reconstructed volumes.

Additionally, missing sections are referenced multiple times, but how much are we talking about (1%, 10%, or more?)

Another area of concern lies in the segmentation of cortical gray matter from 2D sections, which forms a critical step in the reconstruction process. While BrainBuilder employs neural networks for segmentation, the effectiveness of these models may vary depending on the imaging modality and tissue contrasts involved. The reliance on accurate segmentation raises questions about the pipeline's robustness and generalizability across different datasets and experimental conditions.

While reviewing the validity of the BrainBuilder pipeline, I couldn't help but ponder the absence of comparison with positron emission tomography (PET), another method used to visualize neuroreceptors, albeit with lower resolution. Surprisingly, PET was only briefly mentioned in the introduction and completely overlooked in the discussion. Would it not have been appropriate for the authors to compare their reconstructed brain atlas with a PET scanner atlas depicting the same neuroreceptors?

PET imaging is widely recognized for its ability to visualize and quantify the distribution of specific neuroreceptors in the brain. By juxtaposing the results obtained from BrainBuilder's reconstructed 3D brain atlas with those from a PET scanner atlas for the same neuroreceptors. Such a comparison could give valuable insights into the consistency and reliability of BrainBuilder's reconstructions in capturing the spatial distribution of neuroreceptors in the brain. Integrating PET data into the evaluation process would undoubtedly enrich the discussion surrounding the validity and applicability of the BrainBuilder pipeline.

I am pretty old school when I read an article. I follow the classic sequence of reading an article: introduction, materials and methods, results, discussion, and conclusion. Hence, when I initially read the article, I was under the impression that it pertained to only one human brain and one macaque brain. The materials and methods section merely referenced "human" or "macaque" brain without explicitly indicating multiple specimens. Therefore, upon reaching the conclusion, I found myself perplexed by the sudden mention of 3 humans and 4 hemispheres from macaque brains. Additionally, I am curious about the images showing the neuroreceptors of the brains featured in the article. Are these images derived from a single brain or are they a combination representing an average across the multiple brains examined? This ambiguity further fueled my confusion and left me seeking clarification regarding the scope and context of the data presented.

Deformed tissue presents challenges for alignment with structural reference volumes from MRI due to potential skewing and compression. Moreover, tissue placed on glass slides may not lay completely flat, further contributing to deformation. My concern arises from the statement in the article that the alignment process involves rigid 2D inter-section alignment. Rigid alignment typically involves translation and rotation without scaling or shearing the image. However, without scaling or shearing, achieving a Dice score of 91% seems impossible. It's essential to clarify that rigid alignment alone may not adequately address the complexity of tissue deformation.

To address these concerns, visual evidence of the aligned images should be provided in the supplementary materials. This would allow readers to assess the effectiveness of the alignment process and evaluate the extent of deformation correction achieved by the pipeline. It would also be beneficial to talk about deformation and alignment in the discussion section as well.

I suggest that the BrainBuilder pipeline could greatly benefit from the creation of a comprehensive tutorial accessible through Python notebooks or a similar platform. A tutorial like this would walk people through the process of implementing the BrainBuilder pipeline in their own research projects. Furthermore, if the dataset used in the study is public available, it is preferable to store it on a platform such as Zenodo and assign a DOI that can be linked directly to the manuscript. To familiarize users with the features and capabilities of the pipeline, a demo dataset should be provided if the data cannot be made public. Ultimately, the

primary goal is to encourage people to use the BrainBuilder tool, and tutorials and easily accessible datasets play a pivotal role in achieving this goal.

All things considered, BrainBuilder is a very interesting concept, but require more explanations and clarifications. Addressing these concerns could enhance the utility and reliability of the pipeline while also improving our understanding of brain architecture and function.

Specifics

Illustrations

- Please make sure all illustrations use the same font size, and that values on image axes are large enough to be readable.
- Figure 1C, why not just zoom into the individual brain section instead of including part of the nearby section?
- Move Figure 3 into supplementary
- In Figure 6, the image is visually beautiful. However, accurately assessing the correctness of the receptor map presents a challenge, as the corresponding sagittal section's appearance of the GM is unknown. To address this, I suggest including the MRI image in the supplementary without the receptor map overlay.
- Text in Figure 6 Axial, sagittal, and **coronal** sections showing receptor binding sites for a A)
→ Isn't it only sagittal sections?
- Table 1. Make it horizontal instead of vertical.
- Figure 7. I think the standard deviations of the different slabs should be greater than 0.03-0.07, as indicated in table 1. Perhaps it is right, however the variation in the data appears to be more widespread than that.
- In Figure 8, the True Pixel Intensity value appears somewhat ambiguous. As far as I could discern, it seems to be an 8-bit image, which would typically limit the values to a range of 0 to 255. However, if it were a 16-bit image, the values could extend beyond this range. Yet, the precise definition of Pixel Intensity in this context is not clearly specified unless I have missed that information.

Introduction

- Page 2: “Mesoscale atlases(1-5) play a pivotal role in elucidating the intricate architecture of the brain..”
→ The link is not working for the different mesoscale atlases.
- Page 2: “This is because **numerous individual challenges** involved in the reconstruction of 2D autoradiography are, respectively, frequently encountered in other post-mortem 2D imaging data sets.”
→The introduction might benefit from specifying some of the specific **challenges** involved in the reconstruction of 2D autoradiography to provide more clarity to the reader. While it mentions that there are numerous individual challenges, it doesn't elaborate on what these challenges are.

Discussion

- Page 18: “ Our approach therefore only assumes that the GM can be accurately segmented within the sections and does not assume an intensity-based relationship between the acquired sections and the structural reference volume.”
→ Do you occasionally rely on Otsu thresholding? If so, it raises concerns about the accuracy of GM segmentation, as the algorithm may overlook certain sections. Have you quantified the percentage of such occurrences? This could potentially impact the alignment between sections and, consequently, the Dice score.

Conclusion (Page 19)

The results here serve as a proof-of-principle that BrainBuilder can reconstruct images of 2D sections processed for the visualization of receptor autoradiographs, **cell bodies or myelin accurately** at high resolution in both the human and the macaque brain.”

→ It's uncertain whether the resolution is sufficient to capture cell bodies effectively, as many cells are smaller than the in-plane spatial resolution of 20 μm for both human and macaque sections, where the inter-section distance is 400 μm (not stated in the manuscript). Generally, cells in the cortex have diameters below 20 μm, and given that one pixel alone cannot adequately represent a cell, there may be limitations in accurately capturing cell bodies with this resolution.

General comments for conclusion

- The conclusion mentions that the data are from 3 human brains and 4 hemispheres from macaque brains. Surprisingly, this information is not mentioned elsewhere in the article, which can be confusing for readers. Typically, one would expect such details to be clearly outlined in the Materials and Methods section for easy reference.
- Another aspect that raised questions was the section thickness of the cryosections, which appears to be around 20μm but is not explicitly stated in the article beside in the conclusion. This omission leaves readers guessing about the specifics of the experimental setup again.
- This is not stated in the manuscript, but I assume that the spacing between each slide of a particular receptor would be roughly 400 μm (20 * 20μm). Moreover, completing the entire slab (2-3) would require at least 1000 slides (20,000μm/20μm), indicating a significant investment of time and resources for any laboratory looking to replicate this method. This lack of clarity regarding section thickness and slide count could hinder the reproducibility and scalability of the technique.

Materials and method

- Page 21: “ Binary GM sections are generated from the sections using a deep neural network.”
→ What deep learning architecture are you using? Are there any references?
- Page 21: “ In cases where the network fails to provide a segmentation, e.g. returns an empty image, a segmentation method based on Otsu(44) histogram thresholding. In practice, the human sections were segmented using the neural network and the macaque sections were segmented using the histogram thresholding approach.”
→ If the neural network cannot threshold the GM, how good a job would Otsu do? maybe show some examples in supplementary of these situations?

- Page 22: “ We used the nearest neighbour interpolation between aligned 2D GM sections to estimate the morphology of the cortex where no sections could be acquired.”
→ What is the reason for using nearest neighbour? It is because these are the binary images of 0-1? If so it makes sense.
- Page 23: “ The missing pixel intensities for each cortical surface mesh are interpolated by applying linear interpolation over each of the corresponding upsampled inflated spherical meshes.”
→ Why choose linear interpolation and not cubic? Would be good to hear if they have tried cubic and show in supplementary the pros and cons of the different interpolation methods. Could also add some text about this in the discussion section.

Dear Editor and Authors,

Thank you very much for the opportunity to review the manuscript entitled "*BrainBuilder: A Software Pipeline for 3D Reconstruction of Cortical Maps from Multi-modal 2D Data Sets*".

The paper describes a new open-source software repository (*BrainBuilder*), which is applied to two separate types of datasets: one with autoradiographs and whole-brain MRI data from a human subject, and another from a macaque, but the MRI is from a template. The authors describe a convincing strategy for reconstructing sparsely and alternately sampled 2D sections into a 3D volume that bears strict anatomical correspondence with the MRI data. Furthermore, the authors claim that their method is generalisable to other datasets besides their own.

In consideration for publication in Nature Communications Biology, I was asked to review the manuscript as part of a transparent peer review on the basis of the following criteria:

- The results are novel: **yes**
- The paper provides strong evidence for its conclusions: **further evidence required**
- The data are technically sound: **yes**
- The manuscript is important to scientists in the specific sub-field of biology: **yes**

With respect to the above, I have included detailed commentary in the attached PDF file, and hereby I just would like to reiterate my major critical observations:

- 1) The 3D receptor density volumes, which are considered the final output of the BrainBuilder pipeline, seem to be a result of four consecutive interpolations, which raises a significant concern about the integrity of the results. A quantitative comparison with the reconstructed volumetric autoradiograph data is suggested, and based on the results, the modification of the pipeline may be necessary.
- 2) The authors' claim of the registration accuracy between the autoradiographs and the MRI data are seemingly substantiated by Dice score calculations, however, the evaluation does not take segmentation errors into account, which could have propagated into the registration as well. Therefore, at the very least, representative visualisations of the registered autoradiographs with the white matter surface overlay should be provided.
- 3) The description of the methods at several points are incomplete. For example, too little information is given about the neural-network-based segmentation of cortical ribbons in autoradiographs, despite this being mentioned as one of the key technical novelties of the pipeline.

Further to the above, I would like to bring the authors' attention to a conference presentation that I personally gave at the ISMRM in 2023: "The joint registration of multiple microscopy contrasts to MRI in the BigMac dataset" (<https://cds.ismrm.org/protected/23MPresentations/videos/0079.htm>). I believe that our approach was very similar to the presented pipeline, which would warrant a citation and differentiation of the two methods.

Despite the above critical observations, I believe that the work presented in this manuscript is an important contribution to our field, and therefore it will be eligible for publication in Nature Communications Biology, conditional to the sufficient resolution of the above three points, and my detailed comments. **I would therefore recommend the manuscript for major revisions at this time.**

Yours sincerely,

Istvan N Huszar, M.D., D.Phil.

BrainBuilder: A Software Pipeline for 3D Reconstruction of Cortical Maps from Multi-modal 2D Data Sets

Short Title: BrainBuilder: 3D Brain Atlases from 2D Data Sets

Thomas Funck¹, Konrad Wagstyl², Claude Lepage³, Mona Omidyeganeh³, Paule-Joanne Toussaint³, Ting Xu¹, Katrin Amunts^{4,5}, Alexander Thiel^{6,7}, Nicola Palomero-Gallagher^{4,5} & Alan C. Evans³

1. Child Mind Institute, New York, New York, United States of America

2. Wellcome Center for Human Neuroimaging, University College London, London, United Kingdom

3. Montreal Neurological Institute, McGill University, Montreal, Quebec, Canada

4. Institute of Neuroscience and Medicine INM-1, Research Centre Jülich, Jülich, Germany

5. C. & O. Vogt Institute for Brain Research, University Hospital Düsseldorf, Heinrich-Heine-University, 40225 Düsseldorf, Germany

6. Lady Davis Institute, Jewish General Hospital, McGill University, Montreal, Quebec, Canada

7. Department of Neurology and Neurosurgery, McGill University, Montreal, Quebec, Canada

Corresponding author: Nicola Palomero-Gallagher

Institute of Neuroscience and Medicine (INM-1)

Research Centre Jülich

52425 Jülich

Germany

Phone: +49 2461 61-4790

E-mail: n.palomero-gallagher@fz-juelich.de

Abstract

Mesoscale maps of brain architecture are important tools for characterising the ¹molecular ²organisation of the brain. These maps are essential for advancing our understanding of normal and pathologic brain function because they provide a bridge between neuron-level micro-scale imaging and macro-level population brain imaging. Decades of work with mesoscale mapping has relied primarily on post-mortem sections presented in a 2D representation, thereby constraining its capacity to map the 3D columnar and laminar architecture of the cortex. Here we introduce a novel method and software package called BrainBuilder for reconstructing 3-dimensional cortical maps from data sets of 2-dimensional post-mortem serial brain sections processed for the ³visualization of multiple different biological components. This pipeline can be applied to the brains from different species, without the strict need for a corresponding structural reference volume from the brain donor. As a proof of principle, this new method has been applied to data showing the distribution of multiple neurotransmitter receptor binding sites in the human and macaque brain. Additionally images of cell body and myelin stained sections from the macaque brain are also reconstructed. The accuracy of the reconstruction of sections from the human brain was quantified by calculating the Dice score between the reconstructed volume versus its reference anatomic volume. The average Dice score was 0.91, demonstrating a high level of accuracy. We show that BrainBuilder can serve as the basis for the development of future mesoscale 3D atlases.

1. Introduction

1.1 3D mesoscale atlases from 2D postmortem data

Post-mortem brain ⁴imaging of 2-dimensional (2D) brain sections enables the resolution of features at or exceeding the scale of "mesoscale" brain structures (e.g. ⁵architectonic cortical layers, cortical columns, and sub-nuclei). This approach allows the measurement of biological features that may remain inaccessible through in vivo imaging approaches, such as RNA transcriptomics. Mesoscale atlases(1–5) play a pivotal role in elucidating the intricate architecture of the brain, shedding light on the fine-grained details of its structure that are essential for advancing our understanding of brain function and pathology.

Imaging the complex anatomy of the brain using 2D sections has inherent limitations that can only be overcome by restoring the original embedding of these sections through 3D reconstruction. Notably, the laminar structure of the cortex is poorly represented when the section's cutting angle is not perpendicular to the curvature of the cortical surface. It is also

Number: 1 Author: inhuszar Subject: Highlight Date: 4/5/24, 11:44:06

Cellular architecture, maybe? A quick Google search aligns with my natural interpretation of mesoscale: "length scales ranging from that of an individual cell, down to the size of the molecular machines". Based on this, it is not the "molecular organisation of the brain" that I would emphasise here. Otherwise, e.g., if you refer to neurotransmitters, please clarify what you mean by "molecular".

Number: 2 Author: inhuszar Subject: Highlight Date: 4/5/24, 11:40:08

Number: 3 Author: inhuszar Subject: Highlight Date: 4/5/24, 11:42:28

British-American spelling inconsistency: previously you established British spelling with "organisation".

Number: 4 Author: inhuszar Subject: Comment on Text Date: 4/5/24, 11:47:12

Please be more specific about the methodology, if you are talking about 2D sections.

Number: 5 Author: inhuszar Subject: Comment on Text Date: 4/5/24, 11:49:55

Confusing grammar: are these examples of mesoscale structures, or examples by which 2D imaging can surpass mesoscopic visualisation?

difficult to use 2D sections with other brain imaging data sets, e.g., stereotaxic 3D atlases, because it is challenging to map the 2D sections to other data. Accurate 3D reconstruction is therefore essential to creating 3D mesoscale brain atlases.

Existing mesoscale atlases have focused on mapping cell density or myelination. Though ¹macroscopic atlases of some neurotransmitter receptors have been created using in vivo positron emission tomography (6,7), mesoscale atlases of neurotransmitter receptor distributions are lacking. This is unfortunate because mesoscale atlases of neurotransmitter receptor binding sites would not only provide information concerning ²regional differences in receptor densities, but also enable visualization of their laminar distribution patterns. Since cortical layers differ in their connectivity patterns and neurotransmitter receptors underpin all information processing in the brain, both their regional and laminar distribution patterns are key to understanding how mesoscale brain structure links to brain function.

The creation of mesoscale neurotransmitter receptor atlases is particularly challenging because of the obstacles involved in reconstructing autoradiographs, i.e., images of sections processed by in vitro receptor autoradiography, the gold standard method for imaging neurotransmitter receptors. We have created a reconstruction pipeline, BrainBuilder, that, while initially designed to reconstruct mesoscale neurotransmitter receptor atlases, in fact generalizes to virtually any post-mortem 2D sections where the cortical grey matter (GM) can be visualized. This is because ³numerous individual challenges involved in the reconstruction of 2D autoradiography are, respectively, frequently encountered in other post-mortem 2D imaging data sets. Thus, by solving these problems for the particular application of 2D autoradiography, we have created a ⁴flexible, robust reconstruction pipeline that we hope will help generate mesoscale atlases from numerous other modalities.

1.2 Challenges of 3D reconstruction from 2D post-mortem brain sections

Several challenges must be overcome before 2D brain sections can be reconstructed into ⁵usable 3D brain atlases. These include sparse sampling of the sections, variable pixel intensities from ⁶different acquisitions, and non-linear 2D and 3D deformations in the brain sections. In the following, we describe some of the most frequent challenges that our pipeline aims to overcome. For simplicity, the term “section” is used in this work generically, regardless of the actual ⁷biological component visualized in the brain tissue.

A major obstacle to 3D reconstruction are 3D deformations of the donor brain prior to sectioning. These include shrinkage from chemical fixation or deformations occurring during immersion fixation. Deformations are especially severe when the acquisition method, e.g., in vitro autoradiography, requires the use of fresh tissue and where chemical fixation cannot be used (Figure 1.A). The result is that though the coronal sections may appear as 2D coronal

Number: 1 Author: inhuszar Subject: Comment on Text Date: 4/5/24, 12:00:32

Missing definition. How do the authors differentiate a mesoscale atlas from a "macroscopic" one? Is it about the cortical laminar resolution, which is mentioned later in this paragraph, or there are other important differences? Where would a standard T1-weighted 1mm isotropic MRI fall in this binary classification?

Number: 2 Author: inhuszar Subject: Comment on Text Date: 4/5/24, 12:01:11

Even an MRI scan shows "regional differences". As long as you don't define the scale, this is meaningless.

Number: 3 Author: inhuszar Subject: Comment on Text Date: 4/5/24, 12:09:43

It would be great if you could mention a few examples here, so that readers could get an initial idea what kind of problems BrainBuilder can solve.

Number: 4 Author: inhuszar Subject: Comment on Text Date: 4/5/24, 12:11:14

These seem to be unsubstantiated claims at this point, which would warrant an experiment showing reconstructions of more than two datasets of different modalities.

Number: 5 Author: inhuszar Subject: Comment on Text Date: 4/5/24, 12:14:00

Please be more specific. For example, radiologists can report MRI scans that are corrupted by artefacts, and have more than 5 mm slice spacing, which is perfectly useable if the clinical question is about a macroscopic tumour.

Number: 6 Author: inhuszar Subject: Comment on Text Date: 4/5/24, 12:17:44

Do you refer to intra- or inter-modality variations? I.e., pixel intensity differences due to inhomogeneous illumination in scanned 2D histology slides, or contrast differences between brightfield and fluorescent microscopy?

Number: 7 Author: inhuszar Subject: Comment on Text Date: 4/5/24, 12:30:03

This sentence provides no clarification to the use of the word "section". It is naturally understood to be a 2D sample of something. All the sentence says that the something can be anything, without providing examples what it may actually refer to in the context of this work, e.g., histology, immunofluorescence, autoradiograph, MRI slice, blockface photograph, etc.

In fact, more confusion is created by "biological component", as it is not clear what it refers to: an organ, or part of an organ, or a species, or solid tissue vs blood smear?

images, they in fact come from a more complex 3D spatial embedding in the brain which must be recovered during reconstruction.

Figure 1. ¹⁾ Donor brains were not chemically fixed and hence exhibited significant deformation. ²⁾ Tissue slabs were not cut parallel to one another and substantial ³⁾ gaps were present between acquired slabs. ⁴⁾ Exemplary autoradiographs from the human data set showing each of the 20 neurotransmitter binding sites, and illustrating the substantial heterogeneity in pixel intensities between the autoradiographs.

Number: 1 Author: inhuszar Subject: Comment on Text Date: 4/5/24, 12:36:11

Be more descriptive and specific. For example: anterior and posterior views of the same unfixed post-mortem human brain. A coarse comparison with digital surface reconstructions (where did this come from?) shows major bulk deformations in the axial and coronal planes.

Number: 2 Author: inhuszar Subject: Comment on Text Date: 4/5/24, 12:44:03

Are the lines on the brain surfaces reconstructed from actual measurements, or they are hand-drawn examples for illustration? What am I supposed to see in the two images on the right? If those are meant to illustrate thickness, a scale bar would be necessary.

Number: 3 Author: inhuszar Subject: Comment on Text Date: 4/5/24, 12:38:16

What is in the gaps? Are you referring to fragmented tissue? Or you mean that the interslice distance was large?

Number: 4 Author: inhuszar Subject: Comment on Text Date: 4/5/24, 12:41:28

Please crop the irrelevant parts of the images. The figure looks very busy, and it is hard to tell, how the 20 images refer to different "sites", because they all seem to be reasonably close sagittal sections.

Sections may also be acquired from slabs of brain tissue that have not been cut along a single parallel axis (Figure 1.B). This results in 2D sections with varying cutting angles and which are therefore not in the same plane. The sections acquired across these tissue slabs cannot be naively concatenated into a single stack that could then be reconstructed into 3D. ¹ Instead, the sections must be reconstructed first to the slabs of tissue from which they were cut and only then can sections from each slab be combined into a single 3D reconstruction.

Another important obstacle to reconstruction is the use of ² multiple different types of image acquisition from different sections within the same brain. While the use of several methods of acquisition is highly desirable to measure multiple biological features from within the same subject, it results in images with heterogeneous pixel intensity distributions (Figure 1.C). ³ These sections may be difficult to align to one another and will produce 3D volumes with extremely varied pixel intensity distributions when concatenated together.

The use of serial acquisition of multiple types of biological features within the same brain (e.g., receptor density, cell body and white matter (WM) staining) implies that there will be gaps between acquired sections of a given modality. This problem is compounded by the occasional loss of sections due to mechanical processing. ⁴ The sparse sampling of sections necessitates a method for estimating the distribution of pixel intensities in missing sections for a particular type of section.

The challenge posed by missing sections is further aggravated because the cutting angle of the section is ⁵ rarely completely perpendicular to the curvature of the pial surface across the entire cortical ribbon. As a consequence, the laminar distributions of certain biological features, such as cell body or receptor density, may be misrepresented across the cortical depth. Hence an interpolation method is required for estimating missing sections that will not propagate artefacts resulting from the cutting angle of the sections.

1.3 Existing 3D reconstruction strategies

Methods have been developed to attempt to account for each of the above mentioned challenges individually, including the use of fiducial markers in the brain prior to sectioning, block-face imaging, and the use of structural reference volumes from the donor, e.g., typically a T1 weighted (T1w) MRI. Furthermore, many reconstruction algorithms that leverage these techniques to perform 3D reconstruction have been developed (see Dubois(8) and Pichat et al(9) for reviews). Although these specialized pipelines are able to solve many of the challenges presented in 1.2, none of them can address and solve the *combination of all* these challenges. E.g., semi-automated 2D reconstruction methods have been proposed using fiducial markers implanted in the brain prior to sectioning(10)^(11,12) and using block-face imaging(13)^(14,15). Another semi-automated approach to 2D image reconstruction was to manually identify anatomic landmarks on adjacent sections(16,17). Automated reconstruction can be performed with only the 2D sections themselves using principal-axes transforms(18), intensity or

Number: 1 Author: inhuszar Subject: Highlight Date: 4/5/24, 12:56:54

I disagree that it's a necessity. If you have a good enough optimisation method, you may directly infer the 3D surface where the section inserts into the original volume. It may be *easier* to approach the reconstruction in the sequence that you describe, but it is not a necessity.

In fact, it may even lead to further inaccuracies. Consider this example: if one assumes that consecutive 2D sections are parallel, that is only true for the embedded state of the slab. However, the slab might have undergone further deformations during embedding, which are unknown and uncompensated.

Number: 2 Author: inhuszar Subject: Comment on Text Date: 4/5/24, 12:57:57

Please specify the differences between acquisitions.

Number: 3 Author: inhuszar Subject: Comment on Text Date: 4/5/24, 12:59:17

Yes, but if the differences arise from accurate measurements, they should be preserved. If they arise from artefacts, they should be named here, and compensated for.

Number: 4 Author: inhuszar Subject: Comment on Text Date: 4/5/24, 13:04:19

Does a simple trilinear interpolation not solve this problem? It's a trivial concept in image registration that images are defined on a grid, and treated as continuous functions, where values between grid points are inferred via interpolation. If you mean "irregular" by sparse, please emphasise this.

Number: 5 Author: inhuszar Subject: Comment on Text Date: 4/5/24, 13:06:23

It seems pretty much impossible to me. It's an innate consequence of slicing a brain by 2D planes.

frequency-based cross-correlation(19)⁽²⁰⁾, sum of squared error(21), discrepancy matching optical flow(21–23), or edge-based point matching(21–23). Methods have also been developed to perform more robust alignment between sections and to maximise the smoothness of the 3D reconstruction(24–27). A recent and particularly innovative approach to reconstruction involved using Bayesian estimation to simultaneously align the histological sections and corresponding MRI sections while also transforming the pixel intensities of the former to resemble the latter(28)⁽²⁹⁾.

Finally, an iterative strategy for reconstructing 2D *unimodal* sections using an accompanying structural brain image was proposed by Malandain et al(30) and later adapted by Yang(31) and Amunts et al(1). This scheme uses densely sampled sections for a single kind of histological acquisition, e.g., cell-density, and iterates between two steps at progressively higher spatial resolutions. First the reference structural brain image is aligned in 3D to a stack of 2D sections and then the 2D sections are linearly aligned in 2D to the structural brain image. By beginning at a coarse spatial resolution and progressively refining the resolution, these pipelines converge to an accurate alignment between the 2D sections and the 3D reference structural volume.

Summarizing, despite the existence of many reconstruction pipelines, none of them is designed to account for all of the challenges described in section 1.2, and which are associated with the 3D reconstruction of 2D autoradiographs coding for the distribution patterns of **20 different neurotransmitter binding sites**. Hence the need for the development of a pipeline that is capable of working with this highly challenging multimodal data set.

1.4 Aim of the project

Our goal is to create an automated 3D reconstruction pipeline that makes minimal assumptions about the 2D post-mortem sections such that it can be used to create mesoscale brain atlases from a wide variety of 2D acquisitions. In particular, our pipeline works: 1) with multimodal data sets imaged with multiple serial acquisitions such as multi-receptor autoradiographs, 2) with sparsely sampled sections, 3) when sections are acquired from slabs of tissue within the whole brain, 4) without the strict requirement of a corresponding structural reference volume from the brain donor and 5) across multiple species, specifically humans and macaques.

We here apply it to two autoradiography data sets collected by the Zilles labs at Heinrich-Heine-University Düsseldorf and the Research Centre Jülich over a period of more than two decades: one covering the entire human brain, the other that of the macaque monkey (for overviews see, Zilles et al.(32); Palomero-Gallagher and Zilles(32,33); Palomero-Gallagher et al.(34)). This is a unique resource, because it samples, for both species at a high spatial resolution, the distribution patterns of different receptors for the classical neurotransmitters glutamate, GABA, acetylcholine, noradrenaline, serotonin and dopamine and for the neuromodulator adenosine (20 binding site types in the human and 14 in the macaque brain), as well as cell bodies and of myelinated fibers. The challenges present in these two data sets have prevented them from being reconstructed until now.

 Number: 1 Author: inhuszar Subject: Comment on Text Date: 4/5/24, 13:17:55

Sites are still not well-defined in the paper. Do you mean different molecular targets, i.e., neurotransmitter receptors, which produce different imaging contrast on directly adjacent sections of the brain?

Specifically, as proof-of-principle we have demonstrated the application of BrainBuilder on the human brain for the entire receptor autoradiographic dataset of 20 neurotransmitter receptors from one hemisphere and on the macaque brain for a selection of 14 receptors for the classical neurotransmitter systems from one hemisphere. Sections stained for cell body and myelin density were also reconstructed from the macaque data. Furthermore, a reference structural MRI dataset was available for the human but not for the macaque donor brain. We show that BrainBuilder can be used on a wide variety of 2D images and can therefore be used as the basis for the creation of 3D mesoscale atlases of the brain across multiple species and acquisition modalities.

2. Results

We created a pipeline enabling the 3D reconstruction of 2D sections coding for the distribution of 20 different neurotransmitter binding sites in the human brain, and for which a volumetric reference was available. We also used the pipeline for the 3D reconstruction of histologically processed sections, and implemented the option to use a template brain, and not a structural volume of the donor brain, as the reference structural template. As a proof of concept, we reconstructed a dataset from the macaque monkey brain encompassing sections visualizing 14 receptor types as well as cell bodies and myelinated fibres, and for which no structural template was available. The pipeline consists of 3 major processing stages (Figure 2): 1) an inter-section 2D alignment, 2) an iterative multi-resolution 3D volumetric registration followed by 2D section-wise alignment of section to the reference structural brain image, 3) and a surface-based interpolation of receptor binding densities.

Figure 2. Overview of BrainBuilder. The pipeline contains 3 major processing stages: 1) inter-section 2D alignment, 2) iterative multi-resolution 3D volumetric registration followed by 2D section-wise alignment of section to the reference structural brain image, 3) surface-based interpolation of receptor binding densities.

Briefly, BrainBuilder (Figure 2) is composed of 3 major processing stages:

- 1) An initial volume is created by rigid 2D inter-section alignment of acquired sections (Figure 2.1).
- 2) An iterative multi-resolution alignment scheme that alternates between 3D volumetric followed by 2D section-wise alignment of the sections to the reference structural brain image (e.g., donor's T1w MRI; Figure 2.2). The alignment between the reconstructed volume and the reference structural volume is performed using ¹binary GM volumes derived from each of these data sets, respectively. The problem of aligning a volume composed of heterogeneous pixel intensities to a reference volume with an entirely different pixel intensity distribution is thus simplified to mono-modal alignment between GM ²volumes.
- 3) Morphologically informed surface-based interpolation is used to estimate missing pixel intensities for locations where a type of section was not acquired (Figure 2.3).

2.1 BrainBuilder Usage

BrainBuilder is designed to be flexible and simple to use. It can be run through a python script or on the command line (Figure 3.A). The essential information required for the reconstruction is

 Number: 1 Author: inhuszar Subject: Comment on Text Date: 4/5/24, 13:32:11

Do binary GM maps work better than probability maps, e.g. derived from a Gaussian Mixture Model segmentation algorithm? Using binaries is an understandable choice, for the simplicity of their representation and the strong contrast to drive the registration, however, segmentation errors may get amplified in the resultant registration.

 Number: 2 Author: inhuszar Subject: Comment on Text Date: 4/5/24, 13:33:03

segmentation volumes

stored in .csv files which can easily be generated by users without experience in programming (Figure 3.B). The code is openly available on github: <https://github.com/tfunk/brainbuilder>. Additionally, BrainBuilder can be run through a Docker container: `tffunck/brainbuilder:latest`.

A) BrainBuilder python command

```
from brainbuilder.reconstruct import reconstruct

# launch reconstruction with br
reconstruct(
    'hemisphere_info.csv',
    'slab_info.csv',
    'section_info.csv',
    resolution_list=[4,3,2,1,0.5],
    '/path/to/output/'
)
```

B) User .csv inputs:

Hemispheric information in 'hemisphere_info.csv'

sub	hemisphere	struct_ref_vol	gm_surf	wm_surf
MR1	R	MR1_R_gm_srv.nii.gz	MR1_gray_surface_R_81920.surf.gii	MR1_white_surface_R_81920.surf.gii

Slab-level information in 'slab_info.csv'

sub	hemisphere	chunk	pixel_size_0	pixel_size_1	section_thickness	direction
MR1	R	1	0.027	0.020	0.02	rostral_to_caudal
MR1	R	2	0.038	0.029	0.02	rostral_to_caudal
MR1	R	3	0.040	0.030	0.02	rostral_to_caudal
MR1	R	4	0.039	0.029	0.02	caudal_to_rostral
MR1	R	5	0.039	0.029	0.02	caudal_to_rostral
MR1	R	6	0.030	0.023	0.02	caudal_to_rostral

Section-level information in 'section_info.csv'

raw	acquisition	hemisphere	sub	chunk	sample	conversion_factor
RG#hg#MR1s6#R#ampa#5679#04#L.TIF	ampa	R	MR1	6	1353	29.91
RG#hg#MR1s6#R#rx82#5700#21#L.TIF	rx82	R	MR1	6	467	41.15
RG#hg#MR1s6#R#sr95#5723#09#L.TIF	sr95	R	MR1	6	1091	36.42
RG#hg#MR1s6#R#ly34#5721#01#L.TIF	ly34	R	MR1	6	1491	121.40

Figure 3. A) sample piece of code illustrates the usage of BrainBuilder to perform the reconstruction of the human autoradiograph sections B) The user provides the essential information (structural reference volume, cortical surfaces, pixel sizes, etc.) in simple .csv files.

2.2 3D reconstruction

The multi-resolution algorithm for aligning sections to the structural reference volume produced increasingly accurate alignments, as shown exemplarily in Figure 3 for the reconstruction of the most rostral tissue slab from a human hemisphere, where mismatches are clearly visible when the alignment was performed at 1mm resolution, versus that at 0.5mm Figure 4.

 Number: 1 Author: inhuszar Subject: Comment on Text Date: 4/6/24, 11:50:58

Please correct the typo, which made this link inaccessible. I did not review the code, because I only noticed the availability of the code when I reached the correct link at the very end of the document, at which point it was too late in the review process.

 Number: 2 Author: inhuszar Subject: Highlight Date: 4/5/24, 16:11:45

The figure is a great representation of the input data structures, and provides clarity about what types of information are needed for a reconstruction. While such figures would normally belong to a documentation or appendix, I would encourage keeping it in the main text, given the lack of specific information about the reconstruction problem in the introduction.

I appreciate that this is meant to be an illustration of the input data, but I would recommend adding a narrative description of the data in each table to the figure legend. It would help clarify what "conversion factor" stands for, for example.

A few more critical observations about the generalisability of the current input framework: 1) are the numerical fields agnostic to physical units as long as the inputs are consistent? 2) Do the available direction specifications cover all possible cases, e.g. oblique sectioning? 3) Why are pixel sizes anisotropic, and so variable across the records shown? 4) What if certain sections are flipped or rotated, while being mounted on a glass slide? 5) The file name convention seems to be oddly specific. Is it a convenience choice or is it a necessity? 6) How tolerant is the framework for errors in the input tables? Would the processing halt if there was a typo in the table?

 Number: 3 Author: inhuszar Subject: Sticky Note Date: 4/5/24, 16:30:34

Is this an incorrect reference to Figure 4?

 Number: 4 Author: inhuszar Subject: Highlight Date: 4/5/24, 16:23:52

Figure 4: A multi-resolution scheme was used to align the slab volumes to the reference structural volume. At a given resolution, starting at 4mm, the sections are segmented into binary ¹GM volumes. The binary GM volumes are non-linearly aligned in 3D to the reference structural volume. The individual sections are ²then aligned in ³2D to the corresponding sections in the aligned reference structural volume.

To enable a greater flexibility of the pipeline, we implemented the option to reconstruct 2D images for which no structural reference volume was available. The alignment of ⁴sections acquired from a ⁵human donor and a ⁶macaque brain is shown in Figure 5. Whereas the reconstruction of the human sections was performed with a T1w MRI as reference structural volume, the macaque sections were reconstructed with the MEBRAINS(35) stereotaxic template. In Figure 4 the non-linearly aligned 2D sections are shown in a sagittal and axial view with WM and GM cortical surfaces overlaid on top. Visual inspection shows that the reconstruction performed similarly between the human reconstruction with a donor MRI and the macaque reconstruction using a reference template brain.

Number: 1 Author: inhuszar Subject: Highlight Date: 4/5/24, 16:42:09

What are the apparent vertical stripes in the 0.5mm column, top row image? Why are the stripes oblique in the same column, second row? What does the red colour indicate?

Number: 2 Author: inhuszar Subject: Highlight Date: 4/5/24, 16:50:07

It is said that the 2D alignment takes place after the 3D alignment. Does this mean that one of the following is true?

Option 1) the warped image data is resliced in the warped space -> in this configuration, multiple original slices' data may contribute to a single new slice, hence the rationale of a 2D alignment of the new slices is not trivially understood, and should be explained.

Option 2) the 2D transformations precede the 3D transformation -> in this configuration, the warp field is optimised first, and the 2D parameters later. However, this makes the calculation of the cost function expensive, because a non-linear transformation must be applied at every optimisation step. What justifies this expense? Furthermore, how does the precomputed 3D warp field hold up after the 2D parameters are changed? Does this not lead to a degradation of the accuracy?

Number: 3 Author: inhuszar Subject: Highlight Date: 4/5/24, 16:39:52

I appreciate the simplicity, however this is inexact: is it a rigid, a Procrustes, or an affine transformation in 2D?

Number: 4 Author: inhuszar Subject: Highlight Date: 4/5/24, 17:24:39

Were these coronal, or sagittal sections? Figure 5 suggests coronal, but the previous dataset was sagittal. If so, this difference should be pointed out, as it exemplifies the generalisability of the method.

Number: 5 Author: inhuszar Subject: Highlight Date: 4/5/24, 17:03:29

Even if this is from previously published material, the fact that the individual was consented for the present use case, as well as minimal demographic details should be shared here as well. Maybe more appropriately in the methods section.

Number: 6 Author: inhuszar Subject: Highlight Date: 4/5/24, 17:04:42

Same as for the human subject, except for the consent, which should be a reference to the ethical approval of the study which led to the generation of the data that is used in the current paper.

Figure 5: GM (green) and WM (red) cortical surface meshes superimposed on the 3D human (left) and macaque (right) autoradiograph reconstructions. 3D volumes are shown in the receptor coordinate space of each tissue slab. ¹ Autoradiograph sections are almost all correctly aligned to the cortical surfaces.

BrainBuilder was not only able to reconstruct volumes for multiple neurotransmitter receptor binding sites from the human (Figure 6.A) and macaque brains, but also for cell body and myelin stained sections from the latter species (Figure 6.B). BrainBuilder was able to accurately align human sections coding for receptor distribution patterns to their corresponding MRI volume, as well as macaque sections coding for receptor, cell body and myelin distribution patterns to a reference template brain MRI.

Number: 1 Author: inhuszar Subject: Highlight Date: 4/5/24, 17:11:39

Inexact description. Please state what is meant by "almost all" (what percentage out of how many slices?) and "correctly" (what criteria were used to ascertain correct alignment?). Furthermore, please show a representative example of alignment from a success case as well as a failure case as part of this figure.

Why are there no non-linear distortions in the case of the macaque brain?

1) Human

2) Macaque

Figure 6.) Axial, sagittal, and coronal sections showing receptor binding sites for a A) human brain and B) a macaque brain. The macaque brain also includes sections indicating cell body and myelin density. Missing pixel intensities are estimated using surface-based linear interpolation. s. binding site

2.3 Validation

The accuracy of the human reconstruction was quantified by calculating the Dice score between aligned sections and corresponding sections from the structural reference volume. The global average accuracy of the alignment was 0.91 ± 0.06 (Figure 7). Slabs closer to the rostral and caudal poles of the brain had higher average Dice scores than those closer to the center of the brain.

Slab	Dice
1	0.94 ± 0.06
2	0.88 ± 0.06
3	0.88 ± 0.06
4	0.87 ± 0.07
5	0.91 ± 0.05
6	0.93 ± 0.03

Table 1: The mean and standard deviation Dice score of each reconstructed slab.

T Number: 1 Author: inhuszar Subject: Highlight Date: 4/5/24, 18:47:58

If the original slices were sagittal, please provide an alternative view (orthogonal to the original slicing plane), where the accuracy of the reconstruction is more obviously visible.

Since one of the many advantages of autoradiography is that it's quantitative, it would be informative to see a maximum-likelihood projection of these 20 reconstructions, which would show in one volume the most abundant receptor type in each region. This would also be useful to spot potential interpolation artefacts.

T Number: 2 Author: inhuszar Subject: Highlight Date: 4/5/24, 17:15:14

All sections look sagittal. Is anything missing from the figure, perhaps?

T Number: 3 Author: inhuszar Subject: Highlight Date: 4/5/24, 17:21:02

I assume that missing pixels are the consequence of having irregular sections. However, a short explanation would be great to see about these. Please state how abundant the missing pixels are, and whether they are filled by interpolation alone, or smoothing is applied too, in which case, please specify the smoothing radius for reproducibility. Furthermore: does the interpolation respect cortical layers? Is it possible that a misaligned slice will propagate errors to a larger area, across all cortical layers?

T Number: 4 Author: inhuszar Subject: Highlight Date: 4/5/24, 17:21:42

Abbreviation was not used in the figure.

T Number: 5 Author: inhuszar Subject: Highlight Date: 4/5/24, 17:38:30

Is this a Dice overlap of two binary maps, where the background is 0 and the tissue is 1?

If so, the Dice score is not sufficient to characterise the accuracy of the alignment. Theoretically, if the algorithm registered every coronal section to an axial one, this metric would still indicate high accuracy, as long as the outer tissue boundaries are matched.

For a more informative quantification of the registration accuracy, please calculate the Dice score between gray matter segmentations, and give representative visual examples where the alignment is deemed "good" vs "subpar", showing the white matter surface mesh as an overlay on the reconstructed sections.

T Number: 6 Author: inhuszar Subject: Highlight Date: 4/5/24, 17:31:16

Consider "human brain reconstruction" (to clarify that it wasn't a reconstruction done by a human).

T Number: 7 Author: inhuszar Subject: Comment on Text Date: 4/5/24, 17:41:24

Please provide an interpretation for this observation, or state that it is unknown.

T Number: 8 Author: inhuszar Subject: Highlight Date: 4/5/24, 17:44:06

Which slab index corresponds to the anterior pole?

Figure 7: The accuracy (measured as the Dice score) of the alignment between the sections and the reference volume MRI was plotted versus the relative position (0-100% along the coronal axis of the the tissue slab) of the section within the tissue slab.

A surface-based interpolation algorithm was used to estimate missing pixel intensities between acquired autoradiographs. The surface-based interpolation was validated by applying it within randomly selected patches of vertices within acquired autoradiograph sections. For each ligand, the overall correlation between true and interpolated pixel intensities was $r^2=0.97$ ($p<0.001$) (Figure 8). The distances between the vertices with known pixel intensities and the vertices to be estimated spanned 0.05-1.2mm.

Number: 1 Author: inhuszar Subject: Sticky Note Date: 4/5/24, 17:42:02
Dice "score"

Number: 2 Author: inhuszar Subject: Highlight Date: 4/5/24, 17:47:51
I don't see the point why the registration accuracy within the slabs should be superimposed across the slabs, and no interpretation of the graph is given either. Individual accuracy profiles in separate graphs would be more readily interpretable.

Number: 3 Author: inhuszar Subject: Highlight Date: 4/5/24, 17:45:17
Missing directionality: Is 0 or 100% closer to the anterior pole?

Number: 4 Author: inhuszar Subject: Comment on Text Date: 4/5/24, 18:45:06
Please provide a forward reference to the methods section, so that readers know that a more detailed description is available.

Number: 5 Author: inhuszar Subject: Comment on Text Date: 4/5/24, 18:02:04
Please provide further details about how this measurement was carried out. Ideally, provide a supplementary figure explaining the procedure.

Figure 8. The surface-based interpolation algorithm was evaluated within the human autoradiograph sections and demonstrated a high correlation between interpolated and true pixel intensities.

3. Discussion

3.1 ¹Summary

We have created and validated BrainBuilder, a versatile pipeline that can successfully reconstruct 3D volumes from 2D multimodal datasets from different species, and without the constraint of requiring a structural volume of the donor brain. As a proof of principle, we have demonstrated this flexibility by reconstructing a set of receptor autoradiographs coding for the distribution of 20 neurotransmitter receptor binding sites in a human brain for which a T1w volume was available, and a dataset from a macaque monkey brain encompassing the distribution of 14 receptors as well as classical histological cell body and myelin stains, and using the MEBRAINS(35) stereotaxic template as a reference volume. In brief, all sections were aligned to one another using rigid transformations to create an initial 3D volume. An iterative multiresolution scheme was then used to align the volume of 2D sections to the reference structural brain image. Finally, for each kind of section, a surface-based interpolation algorithm was used to create volumetric maps that represent the distribution of the biological feature measured by each respective 2D acquisition method throughout the cortical ribbon.

Visual and quantitative validation shows that BrainBuilder accurately recovers the 3D anatomy of the reference volume (see Figure 5), with an average Dice score of 0.91 ± 0.06 in the case of the human reconstruction. The average Dice score was lower for slabs farther from the rostral or caudal poles of the brain (Table 1), indicating that the position and anatomy of the reconstructed slab impacts the accuracy of the reconstruction. This is likely impacted by the fact that the rostral- and caudal-most slabs are the ²easiest to identify on the structural reference volume. Surface-based interpolation validation demonstrates high correlation ($r^2=0.97$) between true and estimated pixel intensities, indicating accurate estimation of missing pixel intensities over the surface of the cortex.

The results also demonstrate that the BrainBuilder is not limited to human brains, but can be applied to macaque brains as well. Moreover, the reconstruction of the macaque brain was performed without an individual MRI, using the MEBRAINS(35) stereotaxic atlas instead. Our method provides, therefore, the flexibility to reconstruct data sets of 2D sections where no corresponding 3D structural image has been acquired. An important caveat is that, while still presenting a conspicuous gyrification pattern, macaque brain morphology is much simpler than that of humans. ³Therefore, while it may not be necessary to have a corresponding 3D structural image when reconstructing sections from animals with a lower degree of gyrification, it may be required to accurately reconstruct sections acquired from human brains.

3.2 Comparison With Existing Methods

We chose not to apply existing semi-automated reconstruction methods based on manual identification of anatomic landmarks(16,17) because they are dependent on rater subjectivity and not easily reproducible. For the human data the bounds of the tissue slabs on the reference structural image were manually identified, but these points serve only to constrain the 3D alignment and do not drive the reconstruction process. It should also be noted that no manually

Number: 1 Author: inhuszar Subject: Comment on Text Date: 4/5/24, 18:10:11

A large majority of this section is a repetition of the results section. In my opinion, this is not necessary.

Number: 2 Author: inhuszar Subject: Comment on Text Date: 4/5/24, 18:08:27

I would expect that the opposite is true: given the many morphological features of the central sections, even a small misalignment can be easily identified. However, closer to the poles, the complexity of the cortical ribbon decreases, allowing translational misalignments to become larger.

Number: 3 Author: inhuszar Subject: Comment on Text Date: 4/5/24, 18:21:52

Here the authors speculate on the accuracy of template-based reconstruction.

The ability to use a template volume instead of the subject's own, seems to be one of the major strengths of the presented method.

However, the authors did not take the opportunity to test the accuracy of such substitution on a dataset, even despite they had access to both. Before drawing conclusions on this capability, I would expect the authors to repeat the reconstruction of the human brain dataset, using an age-matched reference template.

selected points are used in the reconstruction of the macaque data and hence that BrainBuilder can be used in a ¹fully automated fashion. We were not able to apply 2D reconstruction methods using fiducial markers(10)^(11,12) or relying on block-face imaging(13)^(14,15), because neither fiducial markers nor block-face images were used in the acquisition of our data.

A similar approach to ours was used by both Malandain et al.(30) and Amunts et al.(1) to reconstruct histological volumes in 3D that iterates between two steps: first the donor MRI is aligned in 3D to a stack of sections and then, in the second step, the sections are linearly aligned in 2D to the transformed MRI. Amunts et al.(1) improved on the 3D-2D reconstruction approach by using a multi-resolution non-linear warping schema. However, this 3D-2D approach cannot be used for multimodal datasets due to their ²heterogeneous pixel intensity distributions, nor with ³sparsely sampled sections with. To address these problems we transformed all acquired sections to binary GM masks and recalculated a continuous 3D GM volume between acquired sections at each resolution. The progressively smoother 3D representation of the GM improves alignment to the reference structural volume and allows for the reconstruction of even very sparsely sampled sections.

Our approach differs fundamentally from existing methods through the use of morphologically informed estimation of missing pixel intensities using cortical surface-based interpolation. The advantage of this approach is that cortical surfaces and geodesic distances better represent the actual morphology of the cortex than Euclidean distances because the cortex is organized into layers over a folded manifold.

Many methods have been proposed to create smooth and continuous 3D volumes from 2D sections without the use of external references(25–27,29), such as fiducial markers, block face images, or structural reference volumes. In principle these methods would be helpful for creating the initial reconstruction in Section 4.2.2. However, to our knowledge, these methods cannot be directly applied here again due to the fact that although our dataset consists of serially sectioned brain tissue, alternating sections were used for the visualization of different biological components, thus resulting in a relatively sparse sampling of sections for each modality with heterogeneous pixel intensities. Fortunately, because the multiresolution schema (described in Section 4.2.3) begins at a very low spatial resolution of 4mm^3 , it only requires that the sections be grossly aligned to start with.

Another promising approach has been proposed by Iglesias et al(28) to align sections to corresponding sections from MRI volumes. Their method uses Bayesian inference to estimate both physical transformations to the sections and the transformation of pixel intensities to align sections acquired with different intensity distributions. A key assumption of this approach is that the pixel intensities in sections are a warped version of the corresponding MRI. This assumption is sensible given that both MRI signal and histological signal derive directly, in the case of histology, or indirectly, in the case of MRI, from the density of cells and myelin in the brain tissue. ⁴It is not clear if this assumption would hold if it were to be applied to modalities whose

Number: 1 Author: inhuszar Subject: Comment on Text Date: 4/6/24, 14:03:12

As far as I understand from 4.2.3.2, this is not true for the human dataset, or any other dataset, where slabs need to be registered into a whole brain volume. Please state this clearly. The current description is unilateral and potentially misleading.

Number: 2 Author: inhuszar Subject: Comment on Text Date: 4/5/24, 18:25:45

Certain cost functions, such as normalised cross-correlation (NCC) should allow this, as long as the evaluation of similarity is constrained to the individual 2D sections, and no interpolation takes place between adjacent sections with different contrast properties.

Number: 3 Author: inhuszar Subject: Comment on Text Date: 4/5/24, 18:28:05

What about sparsity prohibits the use of this method? As far as I understand, sparse but regular sections would correspond to very thick slices.

Number: 4 Author: inhuszar Subject: Comment on Text Date: 4/5/24, 18:39:20

As far as I can tell from Figure 1C, the cortical ribbon is usually very distinct from the white matter, so even if the strict form of the assumption is violated, I would still expect a decent performance from this method.

Furthermore, the same trick that was employed in BrainBuilder, which is to convert the input slices to segmentations/probabilities, would have been a reasonable preprocessing step for the method of Iglesias et al, and a comparison would have been adequate.

From the wording of this sentence, it appears to me that a comparison with this method could have been dismissed for other reasons that are not mentioned here.

distribution of pixel intensities are independent of cyto- and myeloarchitecture, as are the laminar distributions of neurotransmitter receptor binding sites (33).

Our method, by contrast, converts all sections to binary GM segmentations and thereby transforms the problem of multi-modal alignment to one uni-modal alignment that can be solved with traditional intensity-based alignment algorithms, such as ANTS(36). Our approach therefore only assumes that the GM can be accurately segmented within the sections and does not assume an intensity-based relationship between the acquired sections and the structural reference volume. This not only means that BrainBuilder can be used with a wide range of 2D ex vivo imaging modalities but also that our approach can in principle be applied with non-MR based structural reference volumes, e.g., with computed tomography, provided the latter have sufficient resolution to support accurate reconstruction.

¹Future work will apply the Iglesias, et al(28) joint registration and synthesis approach to evaluate whether it provides enhanced 2D alignment to the methods presented in 4.2.3.1 and 4.2.3.3. If so, their approach can be integrated into the overall BrainBuilder framework to provide even more precise 3D reconstruction.

3.3 Limitations

Interpolation of Missing Sections

The surface-based interpolation scheme used here performs linear interpolation between vertices and therefore assumes that pixel intensities measured from sections change linearly between the acquired sections and the missing section. This is not strictly biologically valid because there are sharp boundaries between receptor architectonic areas (32,33) which would be obscured by this interpolation method. It does not appear possible to devise an interpolation method that could reproduce such sharp regional boundaries without additional anatomic information.

Segmentation of sections

A potential limitation of our reconstruction method is that it relies on the segmentation of accurate GM images from 2D sections, whether of receptor density or cell and myelin stained sections. It is possible that there are imaging modalities where this estimation of cortical GM may be more challenging. However, ²the network used in this work is trained using only synthetic data which can, in principle, be extended to include different tissue contrasts, more tissue classes, and additional imaging artefacts(37). This means that if there is a particular type of section that is not well segmented by the current network, the synthetic data set can be augmented to reflect the particularities of the problematic sections and hence improve the segmentation of these sections.

 Number: 1 Author: inhuszar Subject: Comment on Text Date: 4/5/24, 18:41:10

Great!

 Number: 2 Author: inhuszar Subject: Comment on Text Date: 4/5/24, 18:55:53

Up until this point, there was no mention of the fact that the underlying framework is a neural network. Since the work fits into the context of mostly traditional optimisation methods, I would urge the authors to make this more apparent in the introduction and include a justification why a network architecture was used.

Conclusion

We have created an image processing pipeline for reconstructing 2D sections into 3D volumes. The results here serve as a proof-of-principle that BrainBuilder can reconstruct images of 2D sections processed for the visualization of receptor autoradiographs, cell bodies or myelin accurately at high resolution in both the human and the macaque brain. We have also demonstrated that this can be done even when no MRI volume is available for the 2D data to be reconstructed. The work presented here will allow for the creation of an unparalleled data set of 20 receptor binding site volumes at 20 μ m for 3 human brains and 4 hemispheres from macaque brains. Future work will focus on extending BrainBuilder to also reconstruct subcortical structures and to evaluate its use on other species.

4. Material and Methods

4.1 Data Preprocessing

The pipeline was initially developed for the 3D reconstruction of 2D serial sections through the human brain, which had been processed by in vitro receptor autoradiography for the visualization of multiple neurotransmitter binding site densities, and for which a structural reference volume of the donor brain was available(32,33). The pipeline was then adapted to enable reconstruction of a comparable 2D multimodal dataset obtained from the macaque brain, but which also included sections stained for cell bodies and myelin fibers, and for which no structural reference volume of the donor brain was available.

For both humans and macaques, brains were removed from the skull at autopsy, the hemispheres were separated and the cerebellum removed. Human hemispheres were cut into 6 coronal slabs, each approximately 2-3 cm thick, macaque hemispheres were cut in the coronal plane into a rostral and a caudal block. Brain slabs were shock frozen and serially sectioned in the coronal plane. Alternating sections were processed by in vitro receptor autoradiography for the visualization of multiple types of receptor binding sites (20 in the human and 14 in the macaque brain), or histologically for the staining of cell bodies and of myelin(32,33). The radioactively labelled sections were exposed against tritium-sensitive films and the ensuing autoradiographs digitized as 8 bit images with an in-plane spatial resolution of 20 μ m resolution for the human and macaque sections, respectively by means of a CCD camera attached to the image acquisition and processing software Axiovision (Zeiss, Germany). Images of the cell-body and myelin stained sections were acquired with a TISSUEScope™ Huron Scanner (Huron, Canada) as 8 bit images with an in-plane spatial resolution of 1 μ m per pixel.

For simplicity, in the following steps we will use the term “section” to refer to the digitized images of both the receptor autoradiographs and the cell-body or myelin stained sections.

4.1.1 Preprocessing receptor autoradiographic sections

For both species, sections were preprocessed to isolate the target piece of brain tissue from each image and remove extraneous tissue and visual cues (for details, see Funck, et al.(38).

4.1.2 Preprocessing MRI

For the human brain, a binary MRI grey matter (GM) volume was derived from the donor's T1w MRI using a mesh representation of the cortical surface. The contrast between the GM and surrounding tissue in the sections facilitates alignment to the GM in the structural reference volume. Cortical surface meshes were obtained from the MRI using the CIVET pipeline(39). A super-resolution cortical GM mask at 250 μ m was obtained from these cortical surface meshes by sampling points between the inner white-matter and outer GM surface meshes(40).

For the macaque reconstruction, the MEBRAINS template was used as reference structural template(35). Hence for the macaque reconstruction, the reference structural volume did not come from the donor brain. The MEBRAINS data release provides cortical surfaces derived with Freesurfer(41,42) and these were used for the reconstruction.

4.2 BrainBuilder Pipeline

4.2.2 Initial inter-section alignment

The initial step of 3D reconstruction involves aligning the sections from all available modalities to one another using 2D rigid body transformations. For simplicity, the ensuing volume is designated as the “initial volume”. Low contrast sections are more likely to be misaligned due to the difficulty in resolving all anatomic structures. It is therefore desirable to first align sections with high contrast before proceeding to lower contrast sections. To this end, sections are first ranked automatically by the Michelson contrast(43) of pixel intensity produced by each acquisition method.

Within each tissue slab, the central section is designated as “fixed” and serves as the reference to which all subsequent sections are aligned. Moving outwards from the central section, sections visualizing the same biological structure (e.g. a given receptor type) are aligned to their nearest “fixed” neighbour towards the center of the slab. Once aligned, each newly added section becomes “fixed” in place.

After aligning the highest contrast sections, they collectively serve as a reference against which to align subsequent lower contrast sections. This process iterates with progressively lower contrast sections until all sections have been aligned.

For the reconstruction of the human brain, 6 initial volumes (one for each of the slabs into which the hemisphere was cut) were produced by rigid alignment of the section. For the macaque brain, the sections were treated as belonging to only a single slab and hence a single initial volume was produced for the acquired hemisphere.

All alignments were calculated with ANTs. For details, see *Supplementary Information 5.1.1*.

4.2.3 Alignment of initial volume to ¹reference structural volume

The alignment of the initial volume to the reference structural volume was done within a multiresolution hierarchical framework. Hence, the steps of the BrainBuilder pipeline described in this section are repeated for each resolution in the hierarchy. The resolutions in the hierarchy were 4.0mm, 3.0mm, 2.0mm, 1.0mm, and finally, at 0.5mm. This resolution schedule is specified by the user at run-time and can be modified to suit the user's particular dataset.

3D reconstruction of a multimodal 2D dataset results in a single volume composed of extremely heterogeneous pixel intensities that are discontinuous between neighboring sections. This makes it impossible to perform volumetric alignment using either cross-correlation or an information theoretic cost function. We simplified the problem of aligning the heterogeneous initial volume to the structural reference volume by creating binary masks representing cortical pixels in both the initial and the reference volumes.

4.2.3.1 Extracting GM mask from initial volume

Binary GM sections are generated from the sections using a ²deep neural network. The network is trained to segment brain GM from 2D images using synthetically generated sections derived from the BigBrain(37). In cases where the network fails to provide a segmentation, e.g. returns an empty image, a segmentation method based on Otsu(44) histogram thresholding. ³In practice, the human sections were segmented using the neural network and the macaque sections were segmented using the histogram thresholding approach.

Then, at each resolution in the multiresolution hierarchy, these 2D GM sections are transformed using the best available transformations to align them together into a single 3D GM volume. At the initial resolution, the rigid body transformations calculated in the initial inter-section alignment (Section 4.2.2) serve as the best available transformations. At subsequent resolutions, the best alignments are the non-linear 2D transformations calculated for the previous resolution in the hierarchy (4.2.3.3). Therefore, after the first step of the multiresolution hierarchy, the 3D GM volumes are ⁴transformed so that they better correspond to the actual

 Number: 1 Author: inhuszar Subject: Comment on Text Date: 4/6/24, 12:10:08

I'm getting confused about this terminology. Does this refer to a slab, or a whole-brain volume, or either?

 Number: 2 Author: inhuszar Subject: Comment on Text Date: 4/6/24, 11:59:27

Given that this is a reference to a conference proceeding, it is necessary to describe the network in more detail here.

 Number: 3 Author: inhuszar Subject: Comment on Text Date: 4/6/24, 12:07:34

Does "in practice" imply that the network failed on all macaque sections, because it was trained on BigBrain, which is a human brain?

Please account for using different approaches, and ideally provide a figure where representative examples from the two methods are contrasted.

 Number: 4 Author: inhuszar Subject: Comment on Text Date: 4/6/24, 12:08:55

As a natural consequence of optimising 2D transformations, or is there a 3D transformation taking place here?

anatomy of the reference structural volume to which they were aligned at the previous resolution of the hierarchy.

The aligned 2D GM sections contain gaps along the coronal axis where 1) no sections were acquired for a particular modality and 2) due to sections lost during acquisition. When reconstructing complete volumes for a particular modality, the missing pixel intensities produced by these gaps must be filled to enable a continuous representation of the cortical GM. ⁴We used the nearest neighbour interpolation between aligned 2D GM sections to estimate the morphology of the cortex where no sections could be acquired.

4.2.3.2 3D alignment of reconstructed volumes to reference structural volume

When more than one slab of tissue is reconstructed, as in the human data set, a significant challenge is to identify which portion of the reference structural volume corresponds to each slab of brain tissue. Due to the deformation of the tissue slabs prior to freezing and loss of sections between slabs, the total width of the brain slabs along the coronal axis was less than that of the brain in the MRI volume, hence the slabs could not simply be placed adjacent to one another. We ²manually identified the anterior and posterior most points of each slab on the reference structural volume and extracted a corresponding slab of tissue from the reference volume. Therefore, the alignment of the receptor slab was limited to a manually defined portion of the corresponding reference structural volume. This step may be omitted if sections sampled from a single tissue slab spanning a whole hemisphere are being reconstructed, as was effectively the case for the macaque data set.

For each slab, the reconstructed GM volume is ³nearly aligned with ANTs(36) to the portion of the reference structural volume. For details, see *Supplementary Information 5.1.2*.

⁴4.2.3.3 2D refinement of section alignment to reference structural volume

After the initial 3D alignment of the reconstructed GM volume to the reference structural volume, the alignment is refined by aligning the sections in 2D to their corresponding coronal sections in the reference structural volume. This 2D alignment between corresponding coronal sections is possible because the 3D alignment in the previous stage produces a reference structural volume that has been transformed into the coordinate space of the reconstructed GM volumes for each tissue slab. The alignment is performed with ANTs(36), for details see *Supplementary Information 5.1.2*.

4.2.4 Surface-based interpolation of missing binding site densities

4.2.4.1 ⁵Upsampling surface meshes

Number: 1 Author: inhuszar Subject: Comment on Text Date: 4/6/24, 12:14:44

Why not linear or spline interpolation?

Number: 2 Author: inhuszar Subject: Comment on Text Date: 4/6/24, 14:09:22

I would expect that the accuracy of this manual definition is crucial, and therefore warrants a more detailed description of how the selection was made.

My expectation is that any error in the delineation of the slab boundaries will have significant non-local effects during the 3D registration with ANTs, as it will aim to line up the tissue edges before anything else.

Therefore, if extra tissue is present in the slab selection, all sections will be drawn from their actual position towards the slab surface, rendering the registration inaccurate.

Number: 3 Author: inhuszar Subject: Comment on Text Date: 4/6/24, 14:13:44

Please clarify what type of linear registration is used here. The referenced supplementary information lists non-linear deformation too, and it is not clear if the linear alignment mentioned here would incorporate scaling.

Number: 4 Author: inhuszar Subject: Comment on Text Date: 4/6/24, 14:14:57

Given that the slab-to-brain alignment was linear, where does the pipeline account for bulk deformations of the slab during handling?

Number: 5 Author: inhuszar Subject: Comment on Text Date: 4/6/24, 14:26:29

Please provide further information about the software used, as well as actual mesh sizes. Was the interpolation barycentric or nearest-neighbour?

Intermediate cortical surfaces are generated by evenly subdividing the WM-GM and pial-GM border (Figure 9.A). Each cortical mesh is then supersampled such that the maximum distance between any two neighbouring vertices is less than or equal to the final resolution of the reconstruction, i.e., 100 μ m for the human reconstruction and 1mm for the macaque reconstruction (Figure 9.C). This upsampling step is done to ensure that there is at least one vertex per voxel in sections where sections have been acquired.

For the human brain, 18 intermediate cortical surfaces were generated, yielding a total of 20 cortical meshes spanning the depth of the cortex between the WM-GM and GM-pial border (Figure 9.B). In the macaque brain, 8 intermediate cortical surface meshes are generated between the WM-GM and GM-pial border, yielding a total of 10 cortical surface meshes. The number of surfaces would be increased for higher resolution reconstructions such that at least one intermediate surface would intersect every pixel between the WM-GM and GM-pial border.

4.2.4.2 Projecting binding densities onto a cortical surface mesh

The surface meshes are transformed with ANTs(36) from the coordinate space of the reference structural volume to the coordinate space of each of the slab volumes, respectively, by applying the inverse linear transformations and non-linear deformation fields calculated in section 4.2.3.2. The pixel intensities in sections are projected onto the surfaces in the native coordinate space of the slab with nearest neighbour interpolation (Figure 9.D).

4.2.4.3 Interpolating missing pixel intensities

When pixel intensities from acquired sections are projected onto the cortical mesh, the gaps between acquired sections from a given modality result in vertices with missing pixel intensities. These missing vertex values are estimated for vertices at which no pixel intensities were measured using a surface-based approach. All of the meshes along the depth of the native cortical surfaces are inflated to spheres (Figure 9.E) using the Freesurfer's `mris_sphere` (iterations = 100) and `mris_inflate`(41). The inflated spherical meshes are then resampled to the same number of vertices as the upsampled surfaces (Figure 9.F).

The missing pixel intensities for each cortical surface mesh are interpolated by applying linear interpolation over each of the corresponding upsampled inflated spherical meshes (Figure 9.G) using the SSRFPACK algorithm(45) implemented in the `stripy`(46) Python package. The interpolation was performed on the inflated sphere instead of directly on the cortical surfaces, because it is computationally simpler to calculate distances between vertices on a simple geometric object like a sphere than on the complex surface of the cortex. Furthermore, given that the surface inflation process approximately preserves relative distances between vertices(41), the interpolated densities on the inflated sphere are equivalent to those which would have been calculated directly on the cortical surface.

 Number: 1 Author: inhuszar Subject: Comment on Text Date: 4/6/24, 14:23:17

Given the uniqueness of the dataset, it would be desirable to have everything computed at the highest reasonable resolution. Please justify your choice of these two resolutions.

 Number: 2 Author: inhuszar Subject: Comment on Text Date: 4/6/24, 14:36:32

It is not clear why this has to be done again. I assumed that you already inflated and upsampled the cortical surfaces in the previous section. If this is a different surface, please clarify. If the same tools were used in the previous section, they should be referenced there.

4.2.4.4 Interpolating surface pixel intensities to a ¹volume

Finally, we implemented an algorithm to project intensity values from the surfaces spanning the cortex into a volume (Figure 9.H). This was done for each voxel within the cortical ribbon by averaging the intensities of the vertices located within the volume of each respective voxel. This method may leave gaps in the reconstructed cortex where no surface vertices are located within the volume of a voxel. The voxel intensities of these empty voxels are estimated by linear interpolation based on the values of the neighbouring voxels with pixel intensities.

Number: 1 Author: inhuszar Subject: Comment on Text Date: 4/6/24, 17:08:33

My understanding is that you are taking a reconstructed volume from sections, where you fill in the gaps with interpolation. Then you resample the cortical data onto a set of parallel surfaces, where you fill in the gaps with interpolation, and then for the third time, you resample the twice-interpolated surface data into 3D space, and fill in the gaps with interpolation. At this point, the original data have been interpolated four times, which means there is potentially a significant degradation.

I do not see a justification for this process, hence, it would be necessary to show that the process 1) leads to no significant degradation of the data, 2) has an advantage over using the reconstruction data from the first step. Ideally, you should present a difference map between the receptor density volumes and the same volumetric information derived from the reconstruction (prior to the surface-based resampling).

I appreciate the validation that was carried out with regards to the surface-based interpolation method. However, this is just one step in the above process, and while the correlation is maintained, the dispersion of the datapoints in Figure 8 clearly illustrates the degradation of the data in just a single step of interpolation. Furthermore, this validation uses originally acquired data, and does not take into account the gaps that were filled with already interpolated data.

Figure 9. Illustrative schema of surface-based interpolation algorithm. Intermediate meshes are defined between the WM and GM surfaces. The cortical surfaces are upsampled and pixel intensities of autoradiographs are projected onto the surface (green circles). Each of the surfaces is inflated to a sphere. Missing pixel intensities are interpolated over the spherical surfaces (red circles). The acquired and interpolated pixel intensities are interpolated into a volume to produce a volumetric atlas of receptor density for a specific receptor.

4.3 Validation

4.3.1 Validation of alignment of human and macaque brain sections

To demonstrate the efficacy of BrainBuilder, it was applied to sections acquired from human and macaque brains.

The final reconstruction for the human data was generated at $300\mu\text{m}^3$. This resolution was chosen because it was sufficient to demonstrate the accuracy of the reconstruction.

The macaque data was only reconstructed up to 4mm as this was sufficient to demonstrate as a proof-of-principle that BrainBuilder can be applied a) to non-human data, b) using a reference template brain as reference structural volume, and c) using cell body and myelin stained sections in addition to receptor autoradiographs.

A challenge in quantifying the accuracy of the alignment is that several of the tissue sections were damaged during acquisition and are missing pieces of tissue. Hence, even if the remaining tissue from a damaged section is perfectly aligned to the corresponding section from the structural reference volume, the resulting Dice(47)⁽⁴⁸⁾ score will be low and not reflect the accuracy of the alignment with the available tissue. To avoid this problem quantitative validation was performed by calculating the Dice score within an 3mm^2 moving window between the aligned GM segmentation of the sections and the corresponding sections in the GM structural reference volume. These local Dice scores were only calculated where tissue was available in the GM tissue segmentations. The local Dice scores were then averaged to produce a global Dice score.

4.3.2 Validation of surface-based interpolation

For each receptor binding site type, respectively, 10,000 vertices that intersected acquired autoradiographs were selected at random. For each of these “seed” vertices, neighbours were identified within n steps along the surface mesh within the same plane as the acquired autoradiograph (all vertices in Figure 10), where n follows a uniform probability distribution $n \sim U(2,6)$. For the purpose of validation, the vertices from the seed vertex to the $n-1$ neighbour (purple, green, and blue vertices in Figure 10) were treated as though the pixel intensities at these locations were missing, though in fact all pixel intensities were known because the

Number: 1 Author: inhuszar Subject: Comment on Text Date: 4/6/24, 14:21:44

Nice figure! It is very useful to understand the actual process.

Number: 2 Author: inhuszar Subject: Comment on Text Date: 4/6/24, 16:54:10

This validation only concerns the accuracy of the registration mechanism, which was performed by ANTs, as prescribed by the authors' multi-resolution multi-step framework. The evaluation does not take into account the possible errors of the segmentation that was used to homogenise the sections, and appears to be a novel contribution above using ANTs to align sections to volumes. It is therefore insufficient on its own to underpin the conclusions about BrainBuilder's accuracy.

Number: 3 Author: inhuszar Subject: Comment on Text Date: 4/6/24, 16:29:22

If the resolution is 500 microns, and the voxels are isotropic, the voxel volume is not 500um³. Please clarify, or indicate isotropic 500um voxels by using parentheses before the exponent.

Number: 4 Author: inhuszar Subject: Comment on Text Date: 4/6/24, 16:31:13

1 mm isotropic

Number: 5 Author: inhuszar Subject: Comment on Text Date: 4/6/24, 16:38:17

While it may be sufficient, it is not a justification for diverging from the already working human protocol. Given that the monkey brain is even smaller, computational requirements should not prohibit the higher-resolution reconstruction. Please elaborate the considerations behind this decision further, as a lower resolution reconstruction has the potential to conceal inaccuracies.

Number: 6 Author: inhuszar Subject: Comment on Text Date: 4/6/24, 16:38:41

Please quantify, or give an estimate.

Number: 7 Author: inhuszar Subject: Comment on Text Date: 4/6/24, 16:42:55

5 mm² would be equivalent to 4.5x4.5 pixels at 500um isotropic resolution in 2D. I assume this was 10x10px instead, so the Dice score was calculated on 100 pixels.

Number: 8 Author: inhuszar Subject: Highlight Date: 4/6/24, 16:55:39

I would strongly suggest simplifying this expression across the whole document. Would it be enough to say "receptor" or "molecular target"?

Number: 9 Author: inhuszar Subject: Comment on Text Date: 4/6/24, 16:57:55

It is hard to believe that vertices intersect the razor-thin autoradiographs in such numbers. Would you please clarify how intersection was defined?

vertices all intersect an acquired autoradiograph. A subset of m neighbours around the seed vertex, where $m \sim U(1,5)$, were then identified (purple and green vertices in Figure 10). These form a core patch of vertices whose average pixel intensity was estimated.

The vertices that were n edges away from the seed vertex (red vertices in Figure 10) were considered to have known pixel intensities. The surface-based linear interpolation algorithm was then used to estimate pixel intensities for vertices within the seed and core patch of vertices given the vertices with “known” intensities. Pixel intensities were estimated for each vertex individually and then averaged together.

In the human data, the average ¹sampling between sections of the same type was approximately $570\mu\text{m}$. The value of n was chosen such that the interpolated vertices were $0.05\text{-}1.2\text{mm}$ away from vertices with known pixel intensities.

Figure 10. A toy example of a mesh patch used to validate the surface-based linear interpolation for estimating missing pixel intensities. Here a seed vertex, purple, with a neighbourhood of core vertices, $m=1$, is estimated using known border vertices, red, $n=3$.

Acknowledgements

Founder

Grant reference number

Author

 Number: 1 Author: inhuszar Subject: Comment on Text Date: 4/6/24, 17:04:45
sampling distance?

European Union's Horizon 2020	945539 (Human Brain Project SGA3)	Katrin Amunts, Nicola Palomero-Gallagher
Federal Ministry of Education and Research (BMBF)	01GQ1902	Nicola Palomero-Gallagher
Helmholtz Association's Initiative and Networking Fund	InterLabs-0015	Katrin Amunts, Alan Evans

*The founders had no role in study design, data collection and interpretation, or the decision to submit the work for publication.

Open Access publication costs are funded by the Deutsche Forschungsgemeinschaft (DFG, German Research Foundation) – 491111487

References

1. Amunts K, Lepage C, Borgeat L, Mohlberg H, Dickscheid T, Rousseau MÉ, et al. BigBrain: an ultrahigh-resolution 3D human brain model. *Science*. 2013 Jun 21;340(6139):1472–5.
2. Howard AFD, Huszar IN, Smart A, Cottaar M, Daubney G, Hanayik T, et al. An open resource combining multi-contrast MRI and microscopy in the macaque brain. *Nat Commun*. 2023 Jul 19;14(1):4320.
3. Alkemade A, Bazin PL, Balesar R, Pine K, Kirilina E, Möller HE, et al. A unified 3D map of microscopic architecture and MRI of the human brain. *Sci Adv*. 2022 Apr 29;8(17):eabj7892.
4. Amunts K, Mohlberg H, Bludau S, Zilles K. Julich-Brain: A 3D probabilistic atlas of the human brain's cytoarchitecture. *Science*. 2020 Aug 21;369(6506):988–92.
5. Woodward A, Hashikawa T, Maeda M, Kaneko T, Hikishima K, Iriki A, et al. The Brain/MINDS 3D digital marmoset brain atlas. *Sci Data*. 2018 Feb 13;5:180009.
6. Nørgaard M, Beliveau V, Ganz M, Svarer C, Pinborg LH, Keller SH, et al. A high-resolution in vivo atlas of the human brain's benzodiazepine binding site of GABA receptors. *Neuroimage*. 2021 May 15;232:117878.
7. Beliveau V, Ganz M, Feng L, Ozenne B, Højgaard L, Fisher PM, et al. A High-Resolution In Vivo Atlas of the Human Brain's Serotonin System. *J Neurosci*. 2017 Jan 4;37(1):120–8.
8. Dubois A, Dauguet J, Delzescaux T. Ex Vivo and In Vitro Cross Calibration Methods [Internet]. *Small Animal Imaging*. 2011. p. 317–46. Available from: http://dx.doi.org/10.1007/978-3-642-12945-2_23
9. Pichat J, Iglesias JE, Yousry T, Ourselin S, Modat M. A Survey of Methods for 3D Histology Reconstruction [Internet]. Vol. 46, *Medical Image Analysis*. 2018. p. 73–105. Available from: <http://dx.doi.org/10.1016/j.media.2018.02.004>
10. Toga A, Arnica T. Image Analysis of Brain Physiology [Internet]. Vol. 5, *IEEE Computer Graphics and Applications*. 1985. p. 20–5. Available from: <http://dx.doi.org/10.1109/mcg.1985.276259>
11. Toga AW, Arnica-Sulze TL. Digital image reconstruction for the study of brain structure and function [Internet]. Vol. 20, *Journal of Neuroscience Methods*. 1987. p. 7–21. Available from: [http://dx.doi.org/10.1016/0165-0270\(87\)90035-5](http://dx.doi.org/10.1016/0165-0270(87)90035-5)
12. Toga AW, Santori EM, Samaie M. Regional distribution of flunitrazepam binding constants: visualizing K_d and B_{max} by digital image analysis. *J Neurosci*. 1986 Sep;6(9):2747–56.
13. Schormann T, Dabringhaus A, Zilles K. Statistics of deformations in histology and application to improved alignment with MRI. *IEEE Trans Med Imaging*. 1995;14(1):25–35.

14. Dauguet J, Delzescaux T, Condé F, Mangin JF, Ayache N, Hantraye P, et al. Three-dimensional reconstruction of stained histological slices and 3D non-linear registration with in-vivo MRI for whole baboon brain. *J Neurosci Methods*. 2007 Aug 15;164(1):191–204.
15. Schubert N, Axer M, Schober M, Huynh AM, Huysegoms M, Palomero-Gallagher N, et al. 3D Reconstructed Cyto-, Muscarinic M2 Receptor, and Fiber Architecture of the Rat Brain Registered to the Waxholm Space Atlas [Internet]. Vol. 10, *Frontiers in Neuroanatomy*. 2016. Available from: <http://dx.doi.org/10.3389/fnana.2016.00051>
16. Kim B, Boes JL, Frey KA, Meyer CR. Mutual information for automated unwarping of rat brain autoradiographs. *Neuroimage*. 1997 Jan;5(1):31–40.
17. Gangolli M, Holleran L, Kim JH, Stein TD, Alvarez V, McKee AC, et al. Quantitative validation of a nonlinear histology-MRI coregistration method using generalized Q-sampling imaging in complex human cortical white matter [Internet]. Vol. 153, *NeuroImage*. 2017. p. 152–67. Available from: <http://dx.doi.org/10.1016/j.neuroimage.2017.03.059>
18. Hibbard LS, Hawkins RA. Three-dimensional reconstruction of metabolic data from quantitative autoradiography of rat brain. *Am J Physiol*. 1984 Sep;247(3 Pt 1):E412–9.
19. Hibbard LS, Hawkins RA. Objective image alignment for three-dimensional reconstruction of digital autoradiograms [Internet]. Vol. 26, *Journal of Neuroscience Methods*. 1988. p. 55–74. Available from: [http://dx.doi.org/10.1016/0165-0270\(88\)90129-x](http://dx.doi.org/10.1016/0165-0270(88)90129-x)
20. Toga AW, Banerjee PK. Registration revisited. *J Neurosci Methods*. 1993 Jun;48(1-2):1–13.
21. Andreasen A, Drewes AM, Assentoft JE, Larsen NE. Computer-assisted alignment of standard serial sections without use of artificial landmarks. A practical approach to the utilization of incomplete information in 3-D reconstruction of the hippocampal region [Internet]. Vol. 45, *Journal of Neuroscience Methods*. 1992. p. 199–207. Available from: [http://dx.doi.org/10.1016/0165-0270\(92\)90077-q](http://dx.doi.org/10.1016/0165-0270(92)90077-q)
22. Rangarajan A, Chui H, Mjolsness E, Pappu S, Davachi L, Goldman-Rakic P, et al. A robust point-matching algorithm for autoradiograph alignment. *Med Image Anal*. 1997 Sep;1(4):379–98.
23. Zhao W, Young TY, Ginsberg MD. Registration and three-dimensional reconstruction of autoradiographic images by the disparity analysis method [Internet]. Vol. 12, *IEEE Transactions on Medical Imaging*. 1993. p. 782–91. Available from: <http://dx.doi.org/10.1109/42.251130>
24. Ourselin S, Roche A, Subsol G, Pennec X, Ayache N. Reconstructing a 3D structure from serial histological sections [Internet]. Vol. 19, *Image and Vision Computing*. 2001. p. 25–31. Available from: [http://dx.doi.org/10.1016/s0262-8856\(00\)00052-4](http://dx.doi.org/10.1016/s0262-8856(00)00052-4)
25. Chakravarty MM, Mallar Chakravarty M, Bertrand G, Hodge CP, Sadikot AF, Louis Collins D. The creation of a brain atlas for image guided neurosurgery using serial histological data [Internet]. Vol. 30, *NeuroImage*. 2006. p. 359–76. Available from:

<http://dx.doi.org/10.1016/j.neuroimage.2005.09.041>

26. Cifor A, Bai L, Pitiot A. Smoothness-guided 3-D reconstruction of 2-D histological images. *Neuroimage*. 2011 May 1;56(1):197–211.
27. Pichat J, Modat M, Yousry T, Ourselin S. A multi-path approach to histology volume reconstruction [Internet]. 2015 IEEE 12th International Symposium on Biomedical Imaging (ISBI). 2015. Available from: <http://dx.doi.org/10.1109/isbi.2015.7164108>
28. Iglesias JE, Modat M, Peter L, Stevens A, Annunziata R, Vercauteren T, et al. Joint registration and synthesis using a probabilistic model for alignment of MRI and histological sections. *Med Image Anal*. 2018 Dec;50:127–44.
29. Mancini M, Casamitjana A, Peter L, Robinson E, Crampsie S, Thomas DL, et al. A multimodal computational pipeline for 3D histology of the human brain [Internet]. Available from: <http://dx.doi.org/10.1101/2020.02.10.941948>
30. Malandain G, Bardinet É, Nelissen K, Vanduffel W. Fusion of autoradiographs with an MR volume using 2-D and 3-D linear transformations [Internet]. Vol. 23, *NeuroImage*. 2004. p. 111–27. Available from: <http://dx.doi.org/10.1016/j.neuroimage.2004.04.038>
31. Yang Z, Richards K, Kurniawan ND, Petrou S, Reutens DC. MRI-guided volume reconstruction of mouse brain from histological sections. *J Neurosci Methods*. 2012 Nov 15;211(2):210–7.
32. Zilles K, Palomero-Gallagher N, Grefkes C, Scheperjans F, Boy C, Amunts K, et al. Architectonics of the human cerebral cortex and transmitter receptor fingerprints: reconciling functional neuroanatomy and neurochemistry. *Eur Neuropsychopharmacol*. 2002 Dec;12(6):587–99.
33. Palomero-Gallagher N, Zilles K. Cyto- and receptor architectonic mapping of the human brain [Internet]. *Handbook of Clinical Neurology*. 2018. p. 355–87. Available from: <http://dx.doi.org/10.1016/b978-0-444-63639-3.00024-4>
34. Palomero-Gallagher N, Zilles K, Schleicher A, Vogt BA. Cyto- and receptor architecture of area 32 in human and macaque brains. *J Comp Neurol*. 2013 Oct 1;521(14):3272–86.
35. Balan PF, Zhu Q, Li X, Bakker R, Palomero-Gallagher N, Vanduffel W. MEBRAINS: a new population-based monkey template (v1.0) [Internet]. EBRAINS; 2023. Available from: <https://search.kg.ebrains.eu/instances/9414c255-ba26-4b4b-ae0b-7e0a48140b0c>
36. Tustison NJ, Cook PA, Holbrook AJ, Johnson HJ, Muschelli J, Devenyi GA, et al. The ANTsX ecosystem for quantitative biological and medical imaging. *Sci Rep*. 2021 Apr 27;11(1):9068.
37. Konrad Wagstyl, Thomas Funck, Joseph Paul Cohen, Katrin Amunts, Alan C Evans, Nicola Palomero-Gallagher. Ultra-high resolution, modality-agnostic segmentation of the cerebral cortex from 2D images. In: Annual Meeting of the Organization for Human Brain Mapping. 2022.

38. Funck T, Wagstyl K, Lepage C, Omidyeganeh M, Toussaint PJ, Amunts K, et al. 3D reconstruction of ultra-high resolution neurotransmitter receptor atlases in human and non-human primate brains [Internet]. bioRxiv. 2022 [cited 2023 Dec 29]. p. 2022.11.18.517039. Available from: <https://www.biorxiv.org/content/10.1101/2022.11.18.517039v1.abstract>
39. Ad-Dab'bagh, Y., Einarson, D., Lyttelton, O., Muehlboeck, J.-S., Mok, K., Ivanov, O., Vincent, R.D., Lepage, C., Lerch, J., Fombonne, E., and Evans, A.C. The CIVET Image-Processing Environment: A Fully Automated Comprehensive Pipeline for Anatomical Neuroimaging Research. In: Corbetta M, editor. NeuroImage; 2006. Available from: <http://www.bic.mni.mcgill.ca/users/yaddab/Yasser-HBM2006-Poster.pdf>
40. Funck T, Paquette C, Evans A, Thiel A. Surface-based partial-volume correction for high-resolution PET. Neuroimage. 2014 Nov 15;102 Pt 2:674–87.
41. Fischl B, Sereno MI, Dale AM. Cortical surface-based analysis. II: Inflation, flattening, and a surface-based coordinate system. Neuroimage. 1999 Feb;9(2):195–207.
42. Dale AM, Fischl B, Sereno MI. Cortical surface-based analysis. I. Segmentation and surface reconstruction. Neuroimage. 1999 Feb;9(2):179–94.
43. Michelson AA. Studies in Optics. Courier Corporation; 1995. 176 p.
44. Otsu N. A threshold selection method from gray-level histograms. IEEE Trans Syst Man Cybern. 1979 Jan;9(1):62–6.
45. Renka RJ. Algorithm 773 [Internet]. Vol. 23, ACM Transactions on Mathematical Software. 1997. p. 435–42. Available from: <http://dx.doi.org/10.1145/275323.275330>
46. Moresi L, Mather B. Stripy: A Python module for (constrained) triangulation in Cartesian coordinates and on a sphere [Internet]. Vol. 4, Journal of Open Source Software. 2019. p. 1410. Available from: <http://dx.doi.org/10.21105/joss.01410>
47. Sørensen T. A Method of Establishing Groups of Equal Amplitude in Plant Sociology Based on Similarity of Species Content and Its Application to Analyses of the Vegetation on Danish Commons. 1948. 34 p.
48. Dice LR. Measures of the amount of ecologic association between species. Ecology. 1945 Jul;26(3):297–302.
49. Mattes D, Haynor DR, Vesselle H, Lewellyn TK, Eubank W. Nonrigid multimodality image registration. In: Sonka M, Hanson KM, editors. Medical Imaging 2001: Image Processing [Internet]. SPIE; 2001. Available from: <http://proceedings.spiedigitallibrary.org/proceeding.aspx?articleid=906829>

5. Supplementary Information

5.1 Parameters for antsRegistration

Within the ANTs software package, “antsRegistration” is the software that calculates the optimal transformation between two images for a given kind of transformation.

5.1.1 Parameters for the initial inter-section alignment

The rigid alignment described in Section 4.2.2 is performed hierarchically with ANTs at voxel sizes of 4.096mm, 1.024mm, and 0.256mm with smoothing kernels of 2mm, 0.5mm, and 0.125mm full width at half-maximum (FWHM), respectively, and 100, 50, and 25 iterations(36).

5.1.2 Parameters for alignment of the reconstructed volume to the structural reference volume

The ANTs alignment parameters used in the 3D alignment of the reconstructed volume (4.2.3.2) and 2D alignment of the individual sections (4.2.3.3) were set as follows. First, the transformations were calculated in the following order: rigid (parameters: 3 translations + 3 rotations), similarity transform (parameters: rigid + 1 global scaling), affine (parameters: rigid parameters + 3 scaling + 3 sheering parameters), SyN non-linear transformation with the Mattes mutual information (MI)(49) metric, and SyN non-linear transformation with local cross-correlation (LCC). The rigid transformation was initialized by antsRegistration using the ¹center of mass of the fixed and moving moving images. Subsequent transformations were initialized using the previous transformation in the series.

While BrainBuilder implements a multiresolution framework within step Section 4.2.3, it is also ²necessary to specify a sub-multiresolution framework within each use of the antsRegistration program. For each transformation, the downsample factors passed to antsRegistration were set such that the alignment would be calculated using all the previous resolutions in the ³resolution hierarchy. That is, if in step Section 4.2.3, the pipeline is currently at resolution 1.0mm out of a hierarchy consisting of $r = (4.0mm, 3.0mm, 2.0mm, 1.0mm, 0.5mm)$, then antsRegistration will be set so that it calculates the optimal transformation between the specified images at 4.0mm, 3.0mm, 2.0mm, and finally the current resolution of 1.0mm. Hence the downsample factors, d_i , are calculated as

$$d_i = r_i / r_c$$

where i is an integer $i \in \{0, 1, \dots, c\}$ that indexes the resolution hierarchy, r , from the lowest level resolution in the hierarchy, $r_0=4.0mm$, to the current level, $r_c=1.0mm$.

The smoothing factors, s_i , were calculated by:

1 Number: 1 Author: inhuszar Subject: Comment on Text Date: 4/6/24, 11:28:14

Does this imply weighting by the original intensities, or the weighting by a binary mask?

1 Number: 2 Author: inhuszar Subject: Comment on Text Date: 4/6/24, 11:34:52

Did you prove that this was necessary, or was it an empirical choice? If the latter, please state that this was your choice.

1 Number: 3 Author: inhuszar Subject: Comment on Text Date: 4/6/24, 11:42:21

Great job! Setting the resolutions and smoothing factors consistently across every run of the program is a welcome practice supporting reproducibility and generalisability.

$$s_i = 0.2 \times d_i \text{ if } i < c, \text{ otherwise } 0$$

The smoothing factor s_c is equal to 0 because no smoothing needs to be performed for a downsample factor of 1 where images are at the current resolution, r_c .

The number of iterations for each use of antsRegistration was calculated by multiplying a given number of iterations:

$$t_i = B_k \times (N - i),$$

where B_k is the base number of iterations for the kind of transformation $k \in \{\text{linear, SyN with Mattes(49) MI, SyN with LCC}\}$, N is the the number of levels in multiresolution hierarchy r , i.e., here $N=6$, and i is, as above, an integer that indexes r . B_k equals 500 for linear transformation, 200 for SyN with Mattes(49) MI, and 100 for SyN with LCC.

Code & Data Availability

The code used in this manuscript is available at <https://github.com/tfunck/brainbuilder>

Please provide a rationale for using this numerical constant.

Review COMMSBIO-24-0971A - BrainBuilder: A Software Pipeline for 3D Reconstruction of Cortical Maps from Multi-modal 2D Data Sets

The paper is well-structured, and the revisions have improved clarity, especially regarding how segmentation and alignment work. While the concept is interesting, there are still some concerns about how well the method works in practice and how useful it will be for others. Given the substantial time investment required for preparing, scanning, and processing post-mortem sections, the practical utility of BrainBuilder depends heavily on whether it can deliver reliable reconstructions despite the large gaps between sections.

1. The authors acknowledged the large section spacing and discussed their choice of linear interpolation. They also mentioned plans to explore more advanced interpolation methods in future work.
 - **My comments:** The concern about interpolation accuracy across large gaps remains. Linear interpolation may not capture fine-grained structural changes, especially in receptor mapping where subtle variations are important. Additionally, the authors noted that sectioning is rarely perfectly perpendicular to the pial surface, which can misrepresent laminar distributions and exacerbate interpolation errors. Another major concern is that only approximately 37% of the human donor brain was sampled with complete sections, with an average distance of 1.03 mm and a maximum distance of 1.42 mm between sections measuring the same biological feature. For the macaque brain, only 24% of the brain was covered, with a sampling distance of 1.32 mm between sections. Given these large gaps, it becomes difficult to be sure that the reconstructed receptor maps accurately represent the real underlying biological structures. In fact, typical MRI scanners already achieve a resolution of 1 mm, which is comparable or even better than the sampling performed here. Although MRI cannot capture receptor signals directly, the resolution mismatch raises concerns about whether the reconstructed maps genuinely provide mesoscopic-level detail. Including a comparison with receptor density measurements from PET scans could help validate the biological accuracy of the reconstructed maps.
2. The authors explained that a direct quantitative comparison with existing methods was challenging due to the lack of equivalent tools. However, they did not include any baseline or qualitative comparison.
 - **My comments:** Without any form of comparison, it is difficult to assess BrainBuilder's relative performance. Additionally, while receptor mapping is the primary focus of BrainBuilder, PET scanners, which provide receptor density measurements in vivo, could serve as a relevant comparison. Despite PET's lower spatial resolution (typically 1-5 mm), it directly measures receptor binding, and such a comparison could help validate BrainBuilder's reconstructed receptor maps. Moreover, this verification would go hand in hand with the interpolation issue: if the reconstructed receptor map resembles PET results to some degree, it would suggest that interpolation across large section gaps is performing reasonably well. Without such a comparison, it remains uncertain whether interpolation over large distances (up to 1.42 mm) is producing biologically meaningful maps.

3. **Handling of Tissue Artifacts** Post-mortem sections often contain artifacts, such as tears or uneven staining, which can affect segmentation and alignment.
 - **My comments:** The authors did not mention any preprocessing steps to handle tissue artifacts or provide examples of how BrainBuilder performs in their presence.
4. **Segmentation Reliability** The authors provided more details about their segmentation process, showing that fallback to Otsu thresholding was rare and explaining how they achieved high Dice scores (~ 0.95 for humans, ~ 0.93 for macaques). Additionally, they validated the U-Net model on manually segmented gray matter (GM) images from real data, using 39 human and 10 macaque samples.
 - **My comments:** The segmentation results look strong, and the validation on manually segmented real data provides confidence in the U-Net model's reliability. However, it remains unclear how often they had to rely on Otsu thresholding and whether those fallback cases caused any noticeable errors.
5. **Scalability of the Approach** The current description of BrainBuilder focuses on reconstructing cortical maps from post-mortem sections with significant manual preparation and scanning effort.
 - **My comments:** Given the time-intensive nature of acquiring and processing post-mortem sections, it is important to assess whether BrainBuilder can handle large-scale datasets efficiently. This includes considerations of computational load and memory requirements during segmentation, alignment, and interpolation. Including a brief discussion of scalability and potential automation steps for future work would be helpful.
6. **Error Propagation Across Reconstruction Steps** Since BrainBuilder involves multiple sequential steps, errors in early stages could propagate through the pipeline.
 - **My comments:** There is no discussion on how errors propagate or whether any safeguards are in place to minimize their impact. Including a brief discussion on error propagation and any measures taken to mitigate its effects would improve confidence in the reliability of the final reconstructions.
7. **Applicability to Non-Cortical Structures** The manuscript focuses on cortical reconstructions, but it is unclear whether BrainBuilder can be applied to other brain regions.
 - **My comments:** Non-cortical structures, such as the hippocampus or basal ganglia, may pose different challenges for segmentation and alignment. If applicable, please clarify whether BrainBuilder can handle non-cortical regions and provide examples demonstrating its performance.

Overall Comment

Overall, the authors have made substantial progress in improving the manuscript, and BrainBuilder has the potential to be a valuable tool for neuroscience research. While this work represents a significant investment of time and effort, ensuring that these remaining issues are clarified will increase its impact and usability for the broader scientific community.